# CausalGame: Benchmarking Causal Thinking of LLM Agents in Games

Zhenhao Chen [* 1]   Yongqiang Chen [* 1 2]   Chenxi Liu [* 3]   Junchi Yu [4]   Xiangchen Song [2]   Zijian Li [1 2]   Jialin Li [5]
Philip Torr [4]   Bo Han [3]   Kun Zhang [1 2]

## Abstract

Recently, it has received growing attention in building AI Scientist agents with Large Language Models (LLMs). Since scientific discovery fundamentally relies on uncovering causal relationships from observations, the capability of *causal thinking* that distinguish causation from correlation and hidden biases, is essential to LLM agents. Despite a number of existing benchmarks for AI scientists, they do not explicitly incorporate challenges from hidden confounders, selection bias, and noisy measurements that widely exist in real-world scientific discovery. To this end, we present `CausalGame`, a benchmark that evaluates the causal thinking capabilities of LLM agents through interactive games. More specifically, we ask LLM agents to actively design experimental protocols, collect observation data and derive a final solution with an explanation report. To emulate realistic scientific discovery challenges, we design 14 game settings with the incorporation of selection bias, noisy measurements, and hidden confounders. The results with 29 frontier LLM agents show that they consistently fail to reason about and recover the underlying causal relationships required to solve the games. `CausalGame` provides a controlled testbed for evaluating causal thinking of AI Scientist agents. The project is available at `causalgame.github.io`.

## 1. Introduction

Recently, as the large language models (LLMs) demonstrate increasing capabilities in reasoning and resolving complex tasks (Guo et al., 2025; Li et al., 2025c; Plaat et al., 2025), it

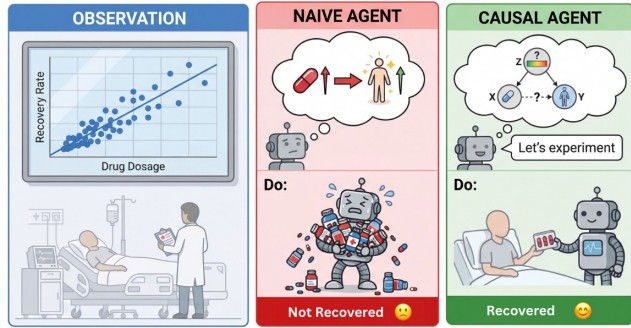

*Figure 1.* Why causal thinking matters for AI scientist agents. Observational correlations (left) can be misleading due to hidden confounders. A naive agent that treats correlation as causation arrives at a suboptimal solution (middle), while a causal agent identifies the underlying mechanism through active experimentation and achieves a substantially better outcome (right). `CausalGame` instantiates such challenges in interactive game scenarios to evaluate whether LLM agents can reason beyond statistical patterns.

has sparked growing curiosity and discussion in the community on building LLM-based AI Scientist agents (ZHENG et al., 2025; Zhou et al., 2025). In fact, there are increasing evidence showing the promise of LLMs in automating research tasks of scientific discovery, such as conducting literature surveys (Lu et al., 2024), proposing useful hypotheses (Mitchener et al., 2025), writing papers Yamada et al. (2025), running machine learning training tasks (Toledo et al., 2025; Hambardzumyan et al., 2026), and discovering novel algorithms or mathematical proofs that surpasses decades of human efforts (Novikov et al., 2025; Hubert et al., 2025; Lange et al., 2025; OpenAI, 2026).

In the history, nevertheless, discovery by human scientists originates from identifying critical variables and revealing the underlying causal mechanisms (Hanson, 1958; Kuhn & Hawkins, 1963; Wallace, 1981; Spirtes et al., 2000). Causal thinking that distinguishes statistical correlations from the causal relations is essential to establishing rigorous scientific conclusions (Glymour; Pearl, 2009). Otherwise, confusing causation with correlation can lead to misleading conclusions or cause severe consequence (Rossouw et al., 2002). For example, the existence of hidden confounders and selection bias can mislead the conclusions driven by statistics (Doll & Hill, 1950; Simpson, 1951). Shown as in Fig. 1, if we were using an AI Scientist agent in medicine

*Equal contribution and core contributors. [1]MBZUAI [2]Carnegie Mellon University [3]TMLR Group, Hong Kong Baptist University [4]University of Oxford [5]New York University, Abu Dhabi. Correspondence to: Bo Han <bhanml@comp.hkbu.edu.hk>, Kun Zhang <kunz1@cmu.edu>.

*Proceedings of the 43rd International Conference on Machine Learning*, Seoul, South Korea. PMLR 306, 2026. Copyright 2026 by the author(s).

*Table 1.* Comparison to the representative existing AI Scientist benchmarks. `CausalGame` features in automated evaluation, imitating real-world multi-turn interaction with designing experiments and receiving feedback, fine-grained evaluation of the explanation provided by the agent, and the challenges underlying observational pitfalls where selection bias, noisy measurement, and hidden confounders will make statistical analysis systematically misleading.

| Paradigm | Dataset | Automated evaluation | Experiment design | Multi-turn interaction | Causal relations | Explanation evaluation | Observational pitfalls |
|---|---|---|---|---|---|---|---|
| Research Workflow | Zhang et al. (2025b) | ✓ | ✗ | ✗ | ✗ | ✓ | ✗ |
| | Liu et al. (2025) | ✓ | ✗ | ✗ | ✗ | ✓ | ✗ |
| | Starace et al. (2025) | ✓ | ✗ | ✗ | ✗ | ✓ | ✗ |
| | Mandal et al. (2025) | ✗ | ✓ | ✗ | ✗ | ✗ | ✗ |
| Data Analysis | Chan et al. (2025) | ✓ | ✗ | ✓ | ✗ | ✗ | ✗ |
| | Jing et al. (2024) | ✓ | ✗ | ✓ | ✗ | ✗ | ✗ |
| | Majumder et al. (2024) | ✓ | ✗ | ✗ | ✗ | ✗ | ✗ |
| | Verma et al. (2025) | ✓ | ✗ | ✗ | ✓ | ✗ | ✗ |
| | Wang et al. (2025b) | ✓ | ✗ | ✓ | ✗ | ✓ | ✗ |
| Interactive Discovery | Jansen et al. (2024) | ✗ | ✓ | ✓ | ✓ | ✗ | ✗ |
| | Roohani et al. (2024) | ✗ | ✗ | ✓ | ✓ | ✗ | ✗ |
| | Zheng et al. (2025a) | ✓ | ✓ | ✓ | ✓ | ✗ | ✗ |
| | Swanson et al. (2025) | ✗ | ✓ | ✓ | ✓ | ✗ | ✗ |
| | Gandhi et al. (2025) | ✓ | ✓ | ✓ | ✗ | ✓ | ✗ |
| Causal Reasoning | Acharya et al. (2025) | ✓ | ✗ | ✗ | ✓ | ✗ | ✗ |
| | Keshmirian et al. (2024) | ✗ | ✗ | ✗ | ✓ | ✗ | ✗ |
| | `CausalGame` | ✓ | ✓ | ✓ | ✓ | ✓ | ✓ |

and the agent is incapable of causal thinking, then the agent can give incorrect treatment that may cause a severe issue. Despite the necessity of causal thinking in scientific discovery, it has been surprisingly neglected in developing and measuring the capabilities of AI Scientists. In contrast, most of the existing AI Scientist frameworks largely rely on LLMs' capabilities to derive scientific hypotheses and conclusions, therefore, in this work, we ask an intriguing research question:

*Are existing LLM agents capable of causal thinking?*

Although there exists a number of benchmarks specifically designed for AI Scientists, they mostly focus on execution of the scientific research pipeline (Wan et al., 2026; Liu et al., 2025), statistical-driven data analysis (Chan et al., 2025; Jing et al., 2024) from *observed* variables (Shojaee et al., 2025; Zheng et al., 2025a), none of them considers the challenges imposed by *hidden* mechanisms beyond observational signals. Nevertheless, the discovery and awareness of the hidden mechanisms are critical to scientific breakthroughs (Glymour; Wallace, 1981).

Therefore, we present a benchmark, `CausalGame`, that simulates the real-world scientific discovery into interactive games, where the agent is required to interact with the environment, collect and analyze observational data, design and perform experiments, and draw hypotheses and conclusions. More specifically, `CausalGame` ask the agent to determining the design of drones, e.g., attributes of the different components. Those drones will be dispatched to execute tasks where different weather conditions and enemy attack can affect the survival rate of drones. The relations between the vulnerability of drone components, the weather conditions, and the enemy attacks are characterized by an underlying *structural causal model* (SCM). The agent will have some budgets to send small patches of the drones to collect the data and gain an understanding of the underlying *causal process*. The understanding will be reported and used to come up with the final design of the drones. Like the real-world scientific discovery, we will evaluate both the quality of the drone design and the report through rubrics.

More importantly, the flexible design in `CausalGame` allows us to incorporate realistic challenges in real-world scientific discovery. Specifically, we construct several game scenarios to incorporate the selection bias, noisy measurements, and hidden confounders (Spirtes et al., 2000). For example, the agent can only observe *survived* drones throughout the turns. Agents that are limited in the capability of causal thinking can easily be biased and suffer from spurious statistical clues. Even when the agent can obtain high ratings by luck, the evaluation design of `CausalGame` can easily distinguish those incapable of causal thinking.

We construct 14 game scenarios in `CausalGame`. Through a comprehensive evaluation of 29 frontier LLMs, we find that LLM agents consistently fail to identify and reason about hidden causal mechanisms under selection bias, noisy measurement, and hidden confounders, and are easily misled by statistical clues. Those results highlight the intrinsic limitations of LLMs in building AI Scientist agents that require causal thinking.

## 2. Related Work

In this section, we briefly review the related work and defer a detailed review to Appendix B.

**AI Scientist agents and benchmarks.** Recent advances in LLM-based agents have drawn increasing attention to the

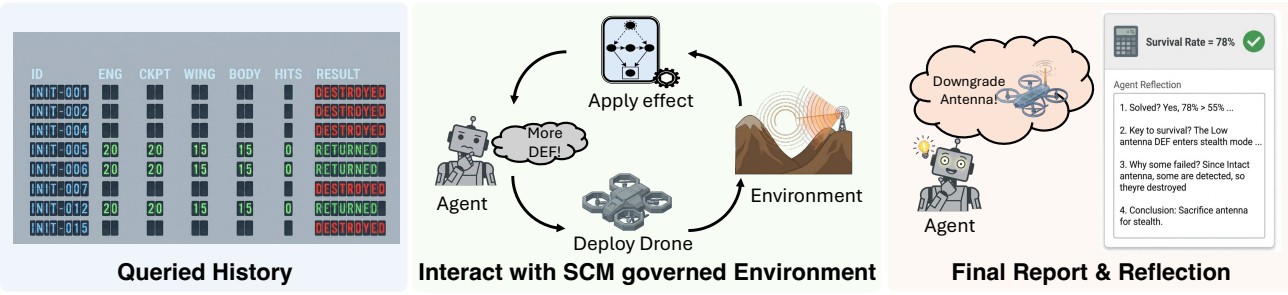

*Figure 2.* Illustration of `CausalGame` pipeline. The agent is given some historical records of surviving drones and required to interact with the environment to gain an understanding of the underlying causal mechanism.

concept of AI scientists, which has great potential for accelerating scientific discovery (Lu et al., 2024; Yamada et al., 2025; Gottweis et al., 2025; Novikov et al., 2025). Faithfully benchmarking the scientific capability of LLMs and LLM-based agents is becoming imperative as they are the foundation for AI scientists. Early studies focused on benchmarking scientific knowledge via multi-disciplinary QA (Yue et al., 2024; Rein et al., 2024; Phan et al., 2025b), while recent works evaluate agentic capabilities across different stages of scientific discovery, including the ideation (Liu et al., 2025), data analysis (Wang et al., 2025b; Shojaee et al., 2025), coding (Starace et al., 2025), interactive scientific discovery (Jansen et al., 2024; Zheng et al., 2025a; Gandhi et al., 2025), and experiment design (Mandal et al., 2025). Beyond AI benchmarks, cognitive science has also studied causal thinking in both humans and LLMs to see whether LLMs possess the same capability (Steyvers et al., 2003; Keshmirian et al., 2024; Geng et al., 2025).

The key differences between `CausalGame` and the existing benchmarks are given in Table 1. Although existing benchmarks provide a holistic evaluation of LLM-based AI Scientist agents, they place less emphasis on replicating the iterative, data-driven nature of real-world scientific discovery. Recent interactive discovery benchmarks such as BoxingGym (Gandhi et al., 2025), DiscoveryWorld (Jansen et al., 2024), and NewtonBench (Zheng et al., 2025a) evaluate whether agents can design experiments and discover underlying relationships in simulated environments, but none explicitly incorporate observational pitfalls such as selection bias and hidden confounders, where naive statistical analysis yields systematically misleading conclusions. The closest benchmarks related to `CausalGame` are Acharya et al. (2025); Verma et al. (2025) that also benchmark the capabilities of LLMs in doing causal inference from the data science perspective. Nevertheless, they lack in the replication of the real-world scientific discovery and the consideration of challenges raised by the observational pitfalls in causality, such as selection bias and hidden confounders.

**Causality and scientific discovery.** Scientific discovery ultimately seeks *causal* and *mechanistic* knowledge, i.e.,

claims about how a system would change under interventions and why, rather than correlations that hold only under a fixed data-generating process (Pearl, 2009; Spirtes et al., 2000). In practice, causal discovery is complicated by latent confounding, selection effects, and measurement noise, all of which can make observational regularities misleading (Spirtes & Glymour, 1991). These challenges have motivated a substantial literature on *active* causal discovery, which asks which interventions most efficiently identify causal structure (Hyttinen et al., 2013; Li et al., 2025a). Yet most active methods assume that all relevant causal variables are observed and that interventions yield clean outcomes, which rarely holds in realistic scientific settings (Liu et al., 2024). Causal representation learning addresses the hidden-variable problem by recovering latent causal processes from observations (Schölkopf et al., 2021; Yao et al., 2024), but these methods typically operate on passively collected data. `CausalGame` bridges these gaps by testing whether LLM agents can actively design experiments and reason causally under hidden confounding, selection bias, and measurement noises to identify hidden causal mechanisms.

## 3. CausalGame Benchmark

In this section, we introduce the key design of `CausalGame` in replicating the setting and challenges in real-world scientific discovery.

### 3.1. Basic game setting

Specifically, in `CausalGame`, the LLM agent acts as a drone designer who must figure out the hidden causal mechanism behind drone survival through a limited budget of experiments, under observations that are censored by survivorship, confounded by hidden variables, and corrupted by noise. We describe the game design below and illustrate the full pipeline in Fig. 2.

**Game objective.** The agent needs to propose and refine the design of a drone by allocating defense values (DEF) across seven components: engine, wing, body, cockpit, antenna, camera, and gun. What the agent can observe is the damaged

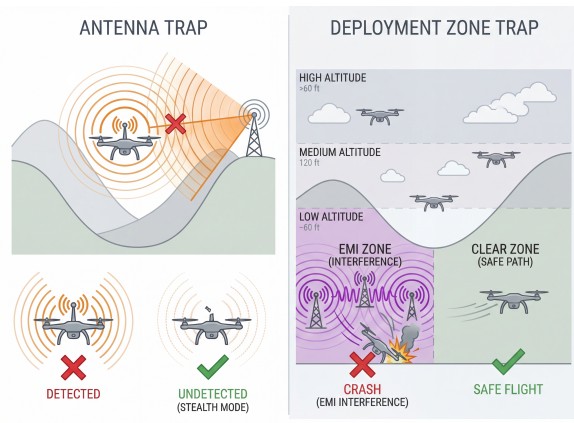

*Figure 3.* Illustration of game scenarios in `CausalGame`.

conditions across different components of the surviving drones from the task execution. The goal is to understand the underlying mechanism that influences the survival of drones and maximize the survival rate when drones are deployed under unknown environmental conditions.

**SCM as the game engine.** The environmental factors that affect drone survival, including weather conditions, enemy detection, and component damage, are governed by an underlying structural causal model (SCM). Our core design choice is to treat the SCM as the scenario "engine": scientific discovery aims to uncover the hidden data-generating mechanism. Concretely, each scenario corresponds to an SCM that specifies structural equations over variables $X_1, \ldots, X_d$ with exogenous noise $U_i$,

$$X_i := f_i(\mathrm{Pa}(X_i), U_i), \quad i = 1, \ldots, d. \tag{1}$$

The agent can only win by recovering the true causal mechanism rather than fitting surface-level correlations. Fully specified structural equations, noise distributions, and parameter values for all 14 scenarios in `CausalGame` are provided in Appendix C.3.

**Example game scenarios.** `CausalGame` contains a suite of scenarios instantiated from a common simulator. As shown in Fig. 3, we start from two base scenarios: `Antenna Trap`, where a latent weather pattern affects both antenna damage and detection risk, and a surviving antenna can increase radar detection via hidden signal emission (thus "protecting the antenna" can be suboptimal); and `Deployment Zone Trap`, where an unobserved mission zone jointly determines the deployment corridor (e.g., altitude band) and the true failure driver (e.g., EMI), inducing a strong but spurious correlation that can mislead correlational strategies.

**Game protocol.** The game proceeds in two stages:

- **Stage 1 (Exploration):** The agent has a budget of 200 drones and up to 10 deployment calls. In each call, the agent chooses a drone design, deploys a small batch, and

receives partial feedback, including survival outcomes and observable attributes of the fleet. Optionally, the agent can access historical observations at the beginning of the game to gain an initial understanding.
- **Stage 2 (Evaluation):** The agent submits a single final design, which is evaluated on a fleet of $1,000$ drones. We report the fleet survival rate. A *win* is defined as exceeding a scenario-specific threshold, set approximately $5\%$–$8\%$ below the theoretical optimal survival rate.

The optimal design for each scenario is derived analytically from the SCM and verified empirically, with theoretical and empirical survival rates agreeing within $\pm 2 - 3$ percentage points (Appendix C.4).

**Evaluation.** In the end, the agent submits both a final design and a short natural-language *report* that explains the design choice based on the evidence collected during interaction. `CausalGame` also evaluate whether the explanations in the report is aligned to the underlying SCM to assess whether the agent truly understands the mechanism.

### 3.2. Causal thinking challenges

With the established game setting, we now elaborate the details of how to incorporate causal thinking challenges in `CausalGame` through SCM. Causal thinking is to infer how and why an outcome would change under hypothetical interventions, rather than merely describing associations observed in data (Spirtes et al., 2000; Pearl, 2009). Across the history of science, many influential empirical findings were initially obscured or misinterpreted due to systematic biases arising from data collection, measurement processes, or unobserved common causes (Wallace, 1981; Glymour). These challenges motivated the development of explicit causal concepts and methodological tools that go beyond correlational analysis. Table 2 lists some representative historical scientific discovery cases that illustrate three recurring obstacles in causal inference and their underlying mechanisms.

**Selection bias** arises when the process by which data are selected depends on variables related to both variables, inducing spurious dependencies in the observed data. This phenomenon is exemplified by Berkson's demonstration of spurious correlations in hospital-based populations (Berkson, 1946) and Sackett's empirical analysis of admission rate bias in case–control studies (Sackett, 1979).

In this benchmark, we introduce controlled selection biases: the agent can only observe *survived* drones. The survival of drones is determined by the underlying SCMs. For example, in `Antenna Trap`, the antenna can be destroyed by bad weather, while drones with a destroyed antenna will be less likely to be detected by enemies and have more chance to survive. Hence, the agent will observe a majority of *surviving drones with damaged antennas*. A natural mitigation is

*Table 2.* A summary of causal mechanisms and representative historical cases considered in our benchmark.

| Challenge | Causal Mechanism | Historical Cases |
|---|---|---|
| Selection bias | X → S ← Y | • Spurious correlation induced by hospital-based selection (Berkson, 1946) 
 • Admission rate bias in hospital case–control studies (Sackett, 1979) |
| Measurement error | X — Z → Y; $X_{obs}$, $Z_{obs}$, $Y_{obs}$ | • Severe attenuation of disease risk estimates due to dietary measurement error (Kipnis et al., 2003) 
 • Regression calibration to correct dietary measurement error in postmenopausal breast cancer studies (Prentice et al., 2013) |
| Latent confounders | U → X, U → Y | • Common-cause explanation for the smoking–lung cancer association (Doll & Hill, 1950) 
 • Unmeasured smoking as a latent confounder in radiation–lung cancer cohort studies (Richardson et al., 2014) |

to strengthen the antenna, which will be trapped by the spurious statistical clues. In addition, we can also strengthen the survival biases in historical data and evaluate whether LLMs can identify and correct selection-induced biases through experimental reasoning.

**Measurement error** arises when latent variables of scientific interest are imperfectly observed through noisy proxies, distorting the conditional independence among observed variables. Dietary epidemiology provides clear evidence of this issue, where the OPEN biomarker study showed severe attenuation of disease risk estimates due to dietary measurement error (Kipnis et al., 2003), and regression calibration was shown to partially recover associations in postmenopausal breast cancer studies (Prentice et al., 2013).

In this benchmark, we inject noise into measurements with varying magnitudes to evaluate whether LLMs can reason robustly under realistic observational imperfections.

**Latent confounder** arises when unobserved common causes jointly influence two variables, generating non-causal associations. Classic and modern examples include the common-cause hypothesis in the smoking–lung cancer debate (Doll & Hill, 1950) and the demonstration that unmeasured smoking substantially confounds radiation–lung cancer associations in occupational cohorts (Richardson et al., 2014). Determining the causal relations requires revealing the latent confounders underlying the observed variables.

In `CausalGame`, by initially withholding critical variables, we test whether LLMs can be aware of the potentially existing latent confounders and actively propose what additional variables should be observed through interacting with environments. Moreover, one could also inject the spurious correlations caused by latent confounders in the history.

### 3.3. Rubric-based evaluation

While survival rate provides a quantitative measure of task performance, it alone cannot distinguish *why* agents fail or how they reason about the causal structure. To address this, we develop a fine-grained rubric evaluation framework that assesses agent behavior along four complementary di-

mensions. The explicit criteria are summarized in Table 18 in the Appendix. Each agent session is evaluated by an LLM-based judge using this rubric, yielding dimension-wise scores and an overall rubric score.

**Causal reasoning (11 points)** evaluates whether the agent correctly identifies the core causal mechanisms specified in the Task Report, avoids known traps or spurious correlations, and provides mechanistic explanations with sufficient depth. High-scoring responses must articulate explicit causal chains (rather than correlations), include intermediate variables or processes, and propose testable predictions or validation strategies.

**Experimental design (2 points)** assesses whether the agent supports its conclusions with concrete experimental evidence. Agents are expected to cite specific numerical results (e.g., survival percentages, controlled comparisons, or threshold conditions) and clearly connect these results to the claims being made.

**Reflection quality (2 points)** evaluates the agent's ability to reflect on its own reasoning. High-quality reflections should identify concrete errors, blind spots, or unverified assumptions that are directly traceable to the proposed approach, rather than offering vague or generic caveats.

**Data usage (1 point)** examines whether the agent explicitly links observed data to conclusions. Agents must state which specific data source or measurements support each claim, avoiding unsupported or purely speculative reasoning.

## 4. Experiments

We conduct extensive experiments to fully understand the causal thinking capability of LLM agents.

### 4.1. Experimental setting

**Models.** We evaluate 29 frontier LLMs spanning major model families: the OpenAI GPT series (GPT-5.5, GPT-5.5-High, GPT-5.5-XHigh, GPT-5.2, GPT-5.2-High, GPT-5-Mini, GPT-OSS-120B), Anthropic Claude (Claude-Opus-4.5, Claude-Opus-4.7, Claude-Sonnet-4.5), Google Gem-

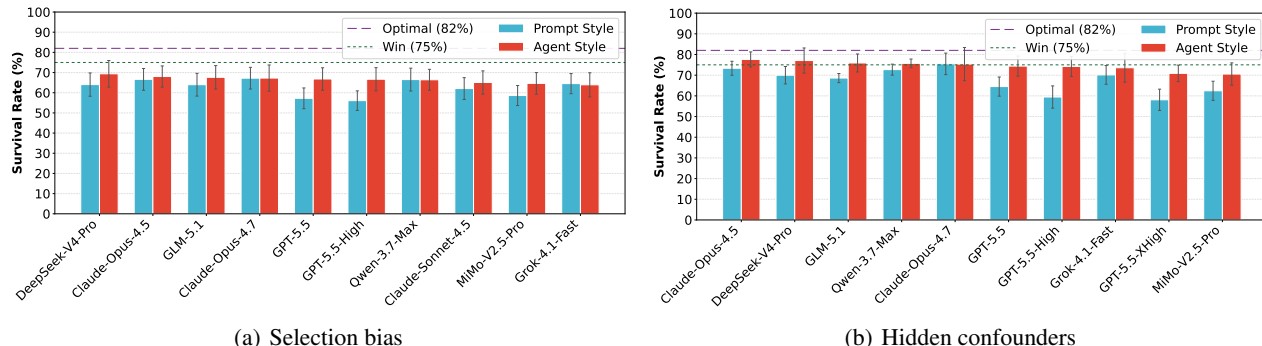

(a) Selection bias
(b) Hidden confounders

*Figure 4.* Main results of different LLM agents in `CausalGame`. The results of the top 10 performing LLMs are reported. We report mean survival rates, and error bars that denote the standard deviation of LLM performance across the corresponding scenarios.

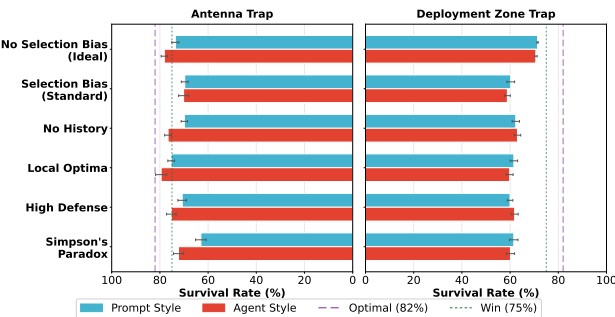

*Figure 5.* Performance comparison under different environments. The error bars denote the standard deviation of LLM performance across the corresponding scenarios.

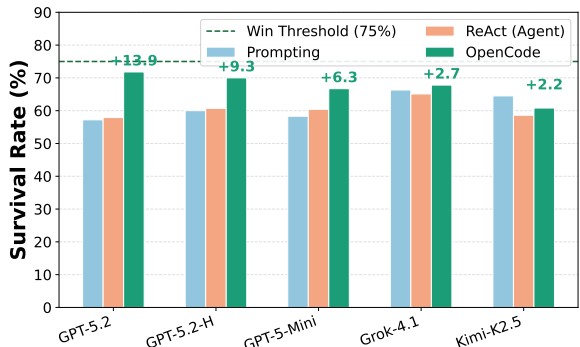

*Figure 6.* Comparison of three execution modes across five models. OpenCode (coding agent) outperforms both Prompting and ReAct on all models, yet remains far below the win threshold (75%).

ini (Gemini-3.5-Flash, Gemini-3.1-Flash-Lite), xAI Grok (Grok-4.1-Fast, Grok-4.20), DeepSeek (V3.2, V3.2-Think, V4-Flash, V4-Pro), and a range of other competitive models including GLM-4.7, GLM-5.1, Kimi-K2.5, Kimi-K2.6, MiniMax-M2, MiniMax-M2.1, MiniMax-M2.7, MIMO-V2-Flash, MIMO-V2.5-Pro, HY3-Preview, and Qwen3.7-Max. The full list of model accesses is given in Table 11.

**Execution modes.** We evaluate each model under two execution modes: (i) **Prompting**, where the agent receives all available data in a single context and submits a design in one turn via code execution; and (ii) **Agentic**, where the agent iteratively calls tools over multiple turns using the ReAct framework (Yao et al., 2023), with mandatory structured reasoning and an exploration guard that requires at least one deployment before final submission. The two modes differ in multiple dimensions, including tool access, interaction format, and reasoning requirements (Table 5). All results are based on 3 independent trials per model×scenario combination. We set the same hyperparameters across all settings, including exploration, deployment budgets, and default API temperature and max-token settings for all models.

### 4.2. Empirical Observations

We summarize the main empirical findings below. We report the results of the top 10 performing LLMs in Fig. 4, and

present the full results in Fig. 13 to Fig. 17. We report both standard deviations and 95% confidence intervals in Appendix D.2 and Appendix D.1.

**Observation 1: Frontier LLMs fail to identify and reason about hidden causal mechanisms.** The main results are given in Fig. 4, where we aggregate and average the performance of different LLMs under selection bias and hidden confounders. We also draw two reference lines: the winning threshold and the optimal performance. From the results, we find that all frontier LLMs remain significantly below the optimal survival rate (∼80%), indicating consistent difficulty in reasoning about hidden confounders, selection bias, and noisy measurements in interactive settings. In addition, the significance analysis of different LLMs across different scenarios from Table 12 to Table 15 confirms that all models remain significantly below the win threshold, indicating the challenges posited by `CausalGame`.

**Observation 2: Increased reasoning effort does not reliably improve causal thinking.** Among the top-performing models, scaling reasoning computation yields no consistent benefit. In Agent mode, GPT-5.5 and GPT-5.5-High both achieve 68.4% and rank within the top 10, yet GPT-5.5-XHigh, which allocates even more reasoning compute, drops to 65.1% and falls outside the top 10. Similarly, DeepSeek-

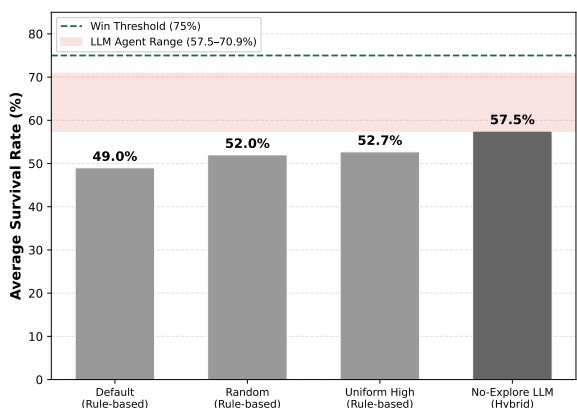

*Figure 7.* Non-LLM baseline comparison. Rule-based and hybrid baselines achieve 49.0%–57.5% average survival rate, well below the win threshold (75%) and the LLM agent range (57.5%–70.9%).

V3.2-Think (59.9%) underperforms its non-thinking counterpart DeepSeek-V3.2 (61.7%). This suggests that simply intensifying chain-of-thought reasoning may cause models to over-fit observed statistical patterns rather than question whether those patterns reflect genuine causal relationships.

**Observation 3: Agentic framework consistently benefits top-performing models.** Among the top-10 models in each mode, 7 appear in both rankings. In nearly all cases, agentic mode yields higher survival than prompting for these models, with DeepSeek-V4-Pro showing the largest gain (+5.7%) and Grok-4.1-Fast being the only near-tie (−1.2%). Three models enter the Agent top-10 but not the Prompt top-10 (GPT-5.5-High, GPT-5.5, MIMO-V2.5-Pro), suggesting these models particularly benefit from iterative experimentation. Conversely, Kimi-K2.5, Kimi-K2.6, and Gemini-3.5-Flash rank in the Prompt top-10 but not the Agent top-10, indicating that not all models leverage the agentic framework effectively. We note that the two modes differ in multiple dimensions (Table 5), so this comparison is not a controlled ablation.

To provide a more informative comparison, we additionally evaluate a coding-agent framework (i.e., OpenCode (open-code, 2026)) on 5 models×14 scenarios×3 trials. Open-Code outperforms ReAct on all 5 models tested (average +6.9% survival rate), confirming that more capable agentic frameworks do improve performance. Nevertheless, a significant gap to the optimal survival rate persists, indicating that causal thinking capability remains the core bottleneck. Full results are provided in Fig. 21 in Appendix G.

**Observation 4: LLMs have certain capabilities in designing useful experiments.** Fig. 5 presents the averaged performance of all the LLM agents under different history data settings. Under ideal settings or the "No history" setting, LLMs require probing the environment by designing experiments, and obtain certain success. More interestingly, given the "Local optimal" setting where a locally optimal design is

*Table 3.* Effect of selection bias on survival rate (%).

| Condition | Agent | Prompt |
|---|---|---|
| w Selection Bias | 64.9 | 64.6 |
| w/o Selection Bias | 74.4 | 72.4 |
| △ | **+9.5** | **+7.8** |

given to the LLM, many of the LLMs are able to iterate over the local optimum and find better solutions. However, when the biases increase or the action space grows more complicated (from `Antenna Trap` to `Deployment Zone Trap`), the failure persists.

### 4.3. Additional analysis

**Non-LLM baselines.** To calibrate benchmark difficulty and confirm task solvability, we evaluate four non-LLM or hybrid baselines: Default (submit initial design unchanged, 49.0%), Random (uniformly sample DEF values, 52.0%), Uniform High (all DEF=50, 52.7%), and No-Explore LLM (randomly deploy 10 times, then use LLM to analyze and submit, 52–63%). More details can be found in Appendix G.

The results are given in Fig. 7. It can be found that these simple baselines can outperform several full-agent models on bias-heavy scenarios, further underscoring the necessity of causal thinking beyond correlational shortcuts.

**Impact of Selection Bias.** Table 3 shows the impact of selection bias on model performance. Removing selection bias through balanced sampling yields substantial improvements: +9.5% for agent mode and +7.8% for prompt mode. Notably, agent mode exhibits greater sensitivity to selection bias. We hypothesize this stems from the compounding nature of sequential decisions. Early biased observations lead to biased deployments, which generate further biased data, progressively reinforcing spurious correlations across turns. In contrast, prompt mode processes all observations simultaneously without this feedback loop, partially mitigating bias amplification.

**Correlational analysis with other benchmarks.** Figure 8 reports the Spearman rank correlations between causal-thinking results and a range of existing agentic benchmark, ranging from hallucination (Chiang et al., 2024), reasoning (Phan et al., 2025a), coding (Jimenez et al., 2024), long-horizon reasoning (Barres et al., 2025; Backlund & Petersson, 2025), long-context understanding (Artificial Analysis, 2025), and parametric knowledge (Jackson et al., 2025). For each model, we collect its CausalGame survival score and its reported score on each external benchmark. We then compute Spearman rank correlations across models between CausalGame scores and each benchmark score.

We can find that the causal thinking capabilities in both agentic and prompting settings only weakly correlate with other capabilities benchmarks. Moreover, we observe a

*Table 4.* Failure mode pattern definitions based on rubric scores. Each session is classified into the first matching pattern in order A through D.

| Pattern | Name | Description |
|---------|------|-------------|
| A | No Engagement | The agent neither designs meaningful experiments nor identifies any causal mechanism. |
| B | Blind Exploration | The agent designs experiments and cites concrete numerical results, but fails to identify causal mechanisms or avoid traps. |
| C | Surface Analysis | The agent links observed data to conclusions but does not identify causal chains or recognize spurious correlations. |
| D | Nascent Reasoning | The agent shows non-zero scores across all dimensions, including some causal reasoning, experimental evidence, data linkage, and self-reflection, but remains weak overall. |

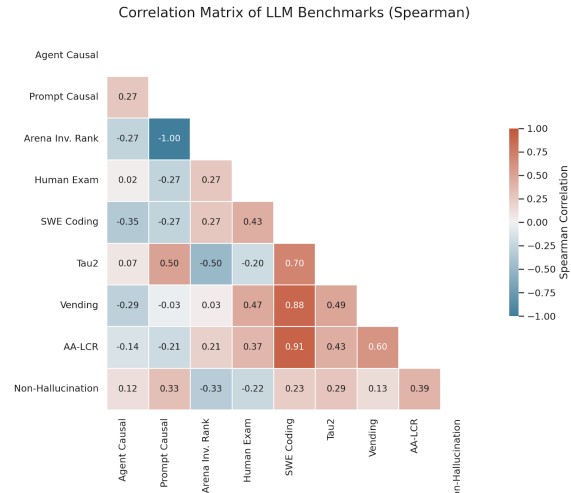

*Figure 8.* Correlation analysis with related benchmark

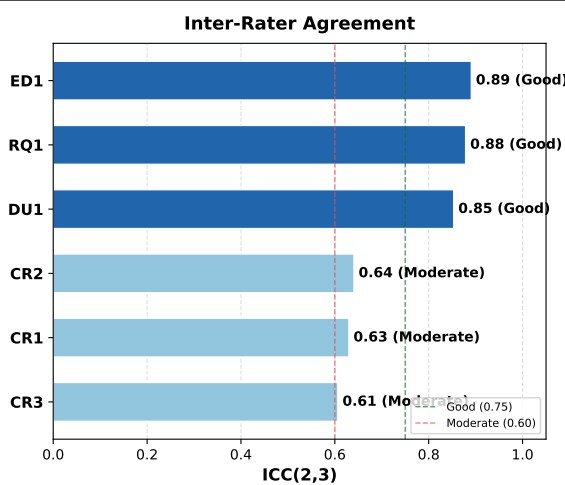

*Figure 9.* Left: ICC(2,3) inter-rater agreement across three judge models. ED1, RQ1, and DU1 achieve good agreement (ICC > 0.85); CR1–CR3 show moderate agreement due to highly skewed score distributions.

clear structural separation between *agentic style* and *prompt style* causal thinking results. Specifically, both the results from agent causal thinking and prompt causal thinking are positively correlated with Non-Hallucination Rate and tau-2 benchmark, where the latter evaluates the multi-turn tool call capabilities. Particularly, prompt causal thinking has much stronger correlations with the compared benchmarks than agent causal thinking. Interestingly, both causal thinking styles are negatively correlated with coding SWE. One plausible explanation is that coding tasks primarily reward pattern completion and syntactic correctness, whereas causal thinking requires hypothesis formation, verification, and reflection, which may involve different reasoning processes.

### 4.4. Fine-grained analysis with rubrics

To gain a fine-grained understanding of how LLM agents behave in `CausalGame`, we conduct a rubrics-based evaluation with LLM-as-a-judge. We first validate the reliability of our LLM-based judge, then analyze agent behaviors through rubric-based failure mode patterns.

**Judge reliability.** To assess the reliability of the rubric-based evaluation, we employ three independent judge models (Gemini-3-Flash, Grok-4-1-Fast-Reasoning, Qwen3-Next-80B-A3B) and compute ICC(2,3) (Shrout & Fleiss, 1979). As shown in Figure 20, the results show strong inter-

rater agreement across all evaluation criteria (Mean ICC = 0.75), with particularly high consistency for Experimental Design (ED1, ICC = 0.89), Reflection Quality (RQ1, ICC = 0.88), and Data Usage (DU1, ICC = 0.85). The Causal Reasoning rubrics (CR1–CR3) show moderate agreement (ICC ∼0.61–0.64); this is primarily attributable to highly skewed score distributions (87–92% of sessions receive zero) rather than judge inconsistency.

**Failure patterns.** By jointly considering rubric scores and survival rate, we identify four qualitatively distinct failure-mode patterns (Table 4), ranging from complete disengagement to nascent causal reasoning: Pattern A (*No Engagement*), where the agent neither designs experiments nor attempts causal reasoning; Pattern B (*Blind Exploration*), where the agent conducts experiments but extracts no causal insight from the results; Pattern C (*Surface Analysis*), where the agent utilizes observed data but remains at the level of describing statistical associations without identifying causal mechanisms; and Pattern D (*Nascent Reasoning*), where the agent shows initial signs of causal reasoning across all rubric dimensions but remains weak overall.

**Agent style** (Figure 10, outer ring): 68.0% of sessions fall into Pattern A (No Engagement), indicating that the majority of agent runs fail to engage with causal reasoning

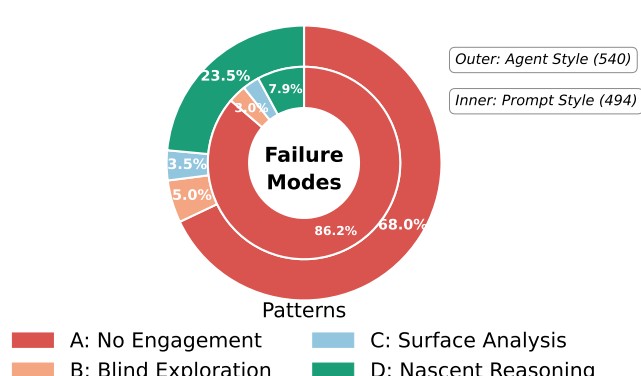

*Figure 10.* Failure mode distribution comparison. Outer ring: Agent Style (540 trails). Inner ring: Prompt Style (494 trails).

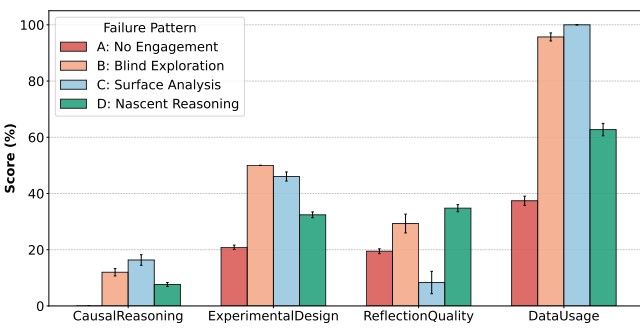

*Figure 11.* Rubric scores by failure mode (Agent Mode). Pattern C exhibits high reflection but low survival.

at all. Pattern C (Surface Analysis) accounts for 23.5%, where agents analyze observed data but do not move beyond descriptive associations to identify the underlying causal mechanisms. Patterns B (Blind Exploration, 5.0%) and D (Nascent Reasoning, 3.5%) are relatively rare.

Dimension-level analysis (Figure 11) confirms that each pattern has a distinct score profile: Pattern A remains uniformly weak across all rubric dimensions; Patterns C and D show substantially higher Data Usage and Reflection Quality than Patterns A and B, reflecting their greater engagement with observational evidence; and Causal Reasoning remains near-zero across all patterns, indicating that even the most data-engaged agents rarely identify genuine causal mechanisms.

**Prompt style** (Figure 10, inner ring) shows an even more skewed distribution: 86.2% Pattern A, with only 7.9% Pattern C and 3.0% Pattern B. The near-total dominance of No Engagement indicates that single-turn prompting severely limits both exploration and data utilization.

The contrast between the two modes reveals that while agentic interaction enables more diverse strategies (reducing No Engagement from 86.2% to 68.0% and increasing Surface Analysis from 7.9% to 23.5%), the core limitation remains: agents that do engage with data overwhelmingly stop at

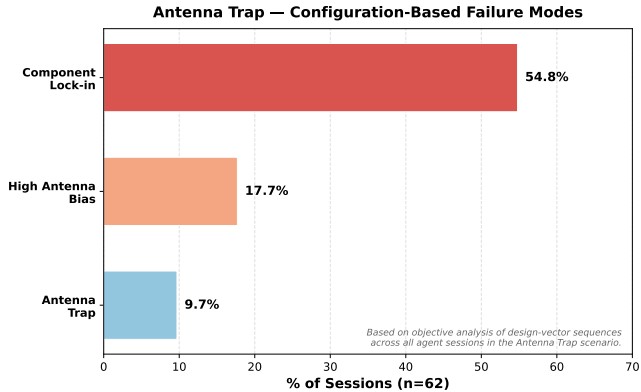

*Figure 12.* Configuration-based failure mode analysis on the Antenna Trap (62 agent sessions). Over 54% of sessions exhibit component lock-in, and 9.7% completely fall into the antenna trap.

surface-level statistical associations (Pattern C) rather than progressing to causal mechanism identification (Pattern D, only 3.5% in Agent mode).

**Configuration-Based Failure Mode Analysis.** To complement the rubric-based analysis with a judge-independent perspective, we conduct a configuration-path analysis on the `Antenna Trap` scenarios, directly examining the sequence of 7-dimensional design vectors agents deployed across all rounds. Key findings across 62 agentic sessions (Figure 22): 54.8% exhibited *component lock-in* ($\leq 2$ unique values for at least one component), indicating severely insufficient exploration; 17.7% partially recognized the antenna trend (testing antenna_def down to 6–10) but never reached the optimal range ($\leq 5$); and 9.7% completely fell into the antenna trap, never testing antenna_def $\leq 10$ despite fluctuating survival rates. Even models that discovered the correct direction often suffered from *optimization drift*: for example, one GPT-5.2 session found antenna_def=3 with 73.3% survival but drifted back to antenna_def=15 in subsequent steps, ultimately submitting a suboptimal design. This behavioral analysis confirms the rubric-based findings through an entirely independent lens: models fail not only in their explanations but in their actual experimental choices.

## 5. Conclusions

In this work, we constructed a benchmark `CausalGame` that instantiated the challenges of real-world scientific discovery in 14 game scenarios. Our benchmarking with 29 frontier LLMs shows that they all fail to uncover the underlying causal mechanism of the environment and are misled by correlational signals. We also present a detailed rubric-based analysis and show that a primary inability of frontier LLMs is to reason about hidden mechanisms under selection bias, noisy measurement, and confounders. The results highlight the limitations of existing LLM agents in use for scientific discovery.

## Acknowledgements

We thank the reviewers for their constructive comments and suggestions. We would like to acknowledge the support from NSF Award No. 2229881, AI Institute for Societal Decision Making (AI-SDM), the National Institutes of Health (NIH) under Contract R01HL159805, and grants from Quris AI, Florin Court Capital, MBZUAI-WIS Joint Program, and the Al Deira Causal Education project. In addition, CXL and BH were supported by NSFC Major Research Plan No. 92570109 and RGC Young Collaborative Research Grant No. C2005-24Y. JCY and PT were supported by the UKRI grant: Turing AI Fellowship EP/W002981/1, and the Schmidt Sciences AI2050 Senior Fellowship.

## Impact Statement

This work focuses on benchmarking LLM-based AI Scientist agents using simulated game scenarios. In addition, this study does not involve human subjects, data set releases, potentially harmful insights, methodologies or applications, conflicts of interest or sponsorship, discrimination, bias or fairness concerns, privacy or security issues, legal compliance concerns, or research integrity issues.

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

## A. Limitations and Future Works

We acknowledge several limitations of this work, which could be promising future directions built upon our work. First, the scenarios in `CausalGame`, despite of being motivated from historical cases, are necessarily simplified compared to real-world discovery, which involves open-ended hypothesis spaces, rich domain-specific knowledge, and substantially more complex causal structures. Future works could further scale up the spaces of observational signals and the hypothesis, to reflect the real-world complexity in scientific discovery.

Second, the Prompting and Agentic execution modes differ in multiple dimensions simultaneously (see Table 5), including tool access, mandatory ReAct formatting, and exploration guards, so cross-mode comparisons should be interpreted with caution rather than as controlled ablations. It remains an open problem on how to reliably study the reasoning capabilities across multiple turns and long horizons.

Third, the LLM-based rubric judge may introduce systematic biases in the fine-grained failure analysis. To mitigate this, we validate judge consistency via multi-judge ICC analysis (Figure 20) and provide a complementary judge-independent configuration path analysis (Figure 22). Hence, a more comprehensive evaluation method, along with metrics on the AI Scientist discovery results, can be a promising future direction.

Last but not least, in terms of solutions, a promising future direction is to use the procedurally generated scenarios in CausalGame for training causal agents. We view meaningful progress as requiring improvements in both survival rate and causal-reasoning rubric scores, so that gains reflect mechanistic understanding rather than only stronger search heuristics.

## B. Related Work

We briefly review the related works in the literature.

### B.1. AI Scientist and Benchmarks

**AI Scientist Agents.** Recent advances in LLM-based agents have drawn increasing attention to the concept of AI scientists, which has great potential for accelerating scientific discovery (Lu et al., 2024; Yamada et al., 2025). The goal of AI scientists is to automate core components of the scientific workflow, including literature review (Huang et al., 2025b), hypothesis generation (Yang et al., 2025), and the systematic design and evaluation of experiments (Huang et al., 2025a). Early efforts in this direction focused on building general AI Scientist frameworks with broad research ability (Gottweis et al., 2025). Recent work has shifted towards viewing the AI scientist framework as a cognitive layer (Zhang et al., 2025a) of scientific research by integrating domain-specific knowledge, specialized tool sets, and *in silico* simulation (Wang et al., 2025a). This paradigm has made huge achievements in biomedicine (Swanson et al., 2025; Truhn et al., 2026), earth science (Feng et al., 2025), material science (Ghafarollahi & Buehler, 2025), computer science (Novikov et al., 2025), and chemistry (Yang et al., 2026; Boiko et al., 2023). To further bridge the gap between the dry-lab research and wet-lab validation, recent studies also explore the integration of embodied AI and robotics for wet-lab automation (Zhu et al., 2022; Tom et al., 2024), enabling closed-loop scientific discovery.

**Benchmark for Scientific Capability** Faithfully benchmarking the scientific capability of LLMs and LLM-based agents is becoming imperative as they are the foundation for AI scientists. Early studies, such as MMMU (Yue et al., 2024) and ScienceQA (Lu et al., 2022), focused on benchmarking the general scientific knowledge of LLMs via multi-modal and multi-disciplinary scientific question answering (QA) (Yue et al., 2025; Hu et al., 2025; Rein et al., 2024). Later benchmarks curated more specialized and advanced scientific QA tasks to benchmark advanced scientific understanding (Wan et al., 2026; Li et al., 2026; Yu et al., 2025; Phan et al., 2025b). Recent works aim to benchmark the capability of LLM-based agents in the workflow of scientific discovery, rather than scientific QA tasks. These benchmarks evaluate agentic capabilities across different stages of research, including the ideation (Liu et al., 2025; Si et al., 2025), review synthesis (Zhang et al., 2025b), data analysis (Wang et al., 2025b; Shojaee et al., 2025), coding (Starace et al., 2025), interactive scientific discovery (Jansen et al., 2024; Zheng et al., 2025a; Gandhi et al., 2025), and experiment design (Mandal et al., 2025). Beyond AI benchmarks, cognitive science has also studied causal reasoning in both humans and LLMs (Steyvers et al., 2003; Bramley et al., 2017; Keshmirian et al., 2024), and recent work examines LLMs' ability to reverse-engineer black-box systems through active intervention (Geng et al., 2025).

The key differences between `CausalGame` and the existing benchmarks for AI Scientists are given in Table 1. Although existing benchmarks provide a holistic evaluation of LLM-based AI Scientist agents, they place less emphasis on replicating

the iterative, data-driven nature of real-world scientific discovery, where agents need to design experiments and interact with the environments to collect more observations to draw scientific conclusions. Recent interactive discovery benchmarks such as BoxingGym (Gandhi et al., 2025), DiscoveryWorld (Jansen et al., 2024), and NewtonBench (Zheng et al., 2025a) primarily evaluate whether agents can design experiments and discover underlying relationships in simulated environments, but none explicitly incorporate observational pitfalls such as selection bias and hidden confounders, where naive statistical analysis yields systematically misleading conclusions. In addition, the evaluation of the scientific report is also essential as it provides the explanation of the discovered causal mechanism. The closest benchmarks related to `CausalGame` are Acharya et al. (2025); Verma et al. (2025) that also benchmark the capabilities of LLMs in doing causal inference from the data science perspective. Nevertheless, they lack in the replication of the real-world scientific discovery and the consideration of challenges raised by the hidden variables in causality.

**Scientific discovery** ultimately seeks *causal* and *mechanistic* knowledge, i.e., claims about how a system would change under interventions and why, rather than correlations that hold only under a fixed data-generating process (Pearl, 2009; Woodward, 2004). Causal graphs and structural causal models (SCMs) formalize how hypotheses generate observations and how controlled perturbations reveal invariant mechanisms (Spirtes et al., 2000). In practice, however, causal discovery is complicated by latent confounding, selection effects, and measurement noise, all of which can make observational regularities misleading and render causal directions unidentifiable without targeted interventions (Spirtes & Glymour, 1991). These challenges have motivated a substantial literature on *active* causal discovery, which asks which interventions most efficiently identify causal structure (Hyttinen et al., 2013; Li et al., 2025a). Yet most active methods assume that all relevant causal variables are observed and that interventions yield clean outcomes that rarely hold in realistic scientific settings where hidden confounders and imperfect measurements are the norm (Liu et al., 2024). Bridging this gap requires evaluations in which an agent must design experiments and reason causally under confounding, bias, and noise to recover underlying mechanisms.

**Causal representation learning** aims to recover the latent causal processes behind observations when well-defined measured variables are unavailable (Schölkopf et al., 2021). A prominent line of work builds on Independent Component Analysis (ICA) (Rajendran et al., 2024; Mansouri et al., 2023; Wendong et al., 2024): classical ICA handles linear mixtures (Comon, 1994; Lee & Lee, 1998; Zhang & Chan, 2007), while nonlinear extensions achieve identifiability by exploiting auxiliary variables (Zheng et al., 2022; Hyvärinen & Pajunen, 1999; Khemakhem et al., 2020b; Li et al., 2023; Khemakhem et al., 2020a; Hyvarinen & Morioka, 2016; 2017; Hyvarinen et al., 2019) or structural priors such as mechanism sparsity (Lachapelle et al., 2023; Lachapelle & Lacoste-Julien, 2022; Zhang et al., 2024). Temporal extensions further leverage nonstationarity and transition sparsity to recover latent dynamics from time series (Hyvarinen & Morioka, 2016; Hälvä & Hyvarinen, 2020; Lippe et al., 2022; Huang et al., 2023; Yan et al., 2024; Yao et al., 2022; 2021; Song et al., 2024), with recent work unifying causal representation learning and dynamical systems to support mechanistic generalization (Yao et al., 2024). Other recent approaches address noisy observation processes by using multiple conditionally independent views to achieve identifiability (Li et al., 2025b; Zheng et al., 2025b). While these methods provide principled frameworks for latent causal structure recovery, they typically operate on passively collected data; `CausalGame` instead tests whether LLM agents can actively identify hidden causal variables through interactive experimentation.

# C. Details of CausalGame Benchmark

This appendix provides comprehensive details about the CausalGame benchmark, including execution modes, scenario descriptions, and the prompts used for evaluation.

## C.1. Execution Modes

CausalGame supports two execution modes that represent different paradigms for LLM-based agent interaction: **Agentic (Hybrid)** mode using structured tool calling, and **Prompting (Legacy)** mode using code execution.

**Agentic (Hybrid) Mode.** In this mode, the agent interacts with the environment through structured function calling. The agent must explicitly invoke tools such as `get_status`, `get_history`, `deploy_drone`, and `submit_final_design`. Each turn requires the agent to provide reasoning before taking actions, following the ReAct (Reasoning and Acting) paradigm. A key safety feature is the exploration guard: the agent must deploy at least one drone before submitting a final design, preventing premature submissions without data collection.

*Table 5.* Comparison of Execution Modes in CausalGame

| Aspect | Agentic (Hybrid) | Prompting (Legacy) |
|---|---|---|
| API Operations | Structured Tool Calling | Python code with `client.xxx()` |
| Data Analysis | Sandboxed Python execution | Full code execution |
| Control Flow | Explicit tool calls with reasoning | Code blocks with API calls |
| Reasoning | Mandatory ReAct pattern | Optional |
| Exploration Guard | Must deploy before submit | None |
| Tool Limit | Max 5 tools per turn | Unlimited API calls |

**Prompting (Legacy) Mode.** In this mode, a pre-configured `client` object is injected into the Python execution namespace. The agent writes Python code blocks that directly call methods like `client.deploy_drone()` and `client.get_history()`. This mode allows for more flexible data analysis through unrestricted code execution but lacks the structured reasoning requirements of the agentic mode.

## C.2. Benchmark Scenarios

CausalGame includes 14 carefully designed scenarios organized into three families, each presenting distinct causal reasoning challenges. Table 6 summarizes all scenarios with their causal challenges.

*Table 6.* Overview of All CausalGame Scenarios

| Experiment | Selection Bias | Hidden Confounder | Threshold |
|---|---|---|---|
| *Antenna Trap Family* | | | |
| antenna_trap | ✓ | – | 75% |
| antenna_trap_high_def | ✓ | ✓ | 75% |
| antenna_trap_local_optima | ✓ | – | 75% |
| antenna_trap_no_history | ✓ | – | 75% |
| antenna_trap_no_selection_bias | – | – | 75% |
| antenna_trap_simpsons_paradox | ✓ | ✓ | 75% |
| *Deployment Zone Trap Family (Farr's Cholera Paradox)* | | | |
| deployment_zone_trap_categorical | ✓ | – | 75% |
| deployment_zone_trap_categorical_high_def | ✓ | ✓ | 75% |
| deployment_zone_trap_categorical_local_optima | ✓ | – | 75% |
| deployment_zone_trap_categorical_no_history | ✓ | – | 75% |
| deployment_zone_trap_categorical_no_selection_bias | – | – | 75% |
| deployment_zone_trap_categorical_simpsons_paradox | ✓ | ✓ | 75% |
| deployment_zone_trap_env_shift | ✓ | – | 75% |
| *Weather Family* | | | |
| weather_noise | ✓ | – | 55% |

### C.2.1. ANTENNA TRAP FAMILY

The Antenna Trap scenarios are inspired by real-world signal detection problems where a functioning communication system can paradoxically increase risk.

**Causal Structure.** The underlying causal graph contains the following relationships:

- Weather → Wind Intensity → Antenna Damage

- Antenna HP → Signal Emission → Detection Probability → Combat Engagement

- Combat Engagement → Drone Damage → Survival

**The Trap.** Historical data shows that drones with higher antenna DEF (defense) values tend to survive better in the training distribution. This creates a spurious correlation: agents naturally conclude that maximizing antenna DEF improves survival.

However, the true causal mechanism is that a *functional* antenna emits signals that increase detection probability by enemy systems, leading to combat and destruction. The optimal strategy is to set `antenna_def=0`, allowing storms to destroy the antenna early, which activates "stealth mode" and dramatically reduces detection.

**Variants.**

- **high_def**: Adds pressure to allocate high DEF values, creating an additional confounder.

- **local_optima**: Introduces local optima that trap gradient-following strategies.

- **no_history**: Removes historical flight data, requiring pure exploration.

- **no_selection_bias**: Control condition without selection bias.

- **simpsons_paradox**: Data exhibits Simpson's paradox where aggregate trends reverse within subgroups.

C.2.2. DEPLOYMENT ZONE TRAP FAMILY

This family is inspired by Farr's Cholera Paradox, a historical example where altitude appeared to protect against cholera when the true cause was water source contamination at lower elevations.

**Causal Structure.**

- Deployment Zone → Altitude (Visible)

- Deployment Zone → EMI Level (Hidden) → Communication Failure

- Communication Failure → Mission Failure → Drone Loss

**The Trap.**   Agents observe that low-altitude flights have significantly higher loss rates and may conclude that engine upgrades (for altitude capability) are the solution. However, the true causal factor is electromagnetic interference (EMI), which is hidden from initial observations. Low-altitude zones happen to have high EMI levels, creating the spurious altitude-survival correlation. The optimal strategy is to maximize `shield_def` for EMI protection and select the `signal_filter` enhancement module.

**Enhancement Modules (Categorical Variant).**   The categorical variant requires agents to select one enhancement module:

- `radar_boost`: No EMI protection (trap)

- `thermal_shield`: No EMI protection (trap)

- `power_core`: No EMI protection (trap)

- `stealth_coating`: No EMI protection (trap)

- `signal_filter`: Provides 55% EMI reduction (optimal)

**Variants.**

- **high_def**: Additional DEF allocation pressure.

- **local_optima**: Local optima traps.

- **no_history**: No historical data available.

- **no_selection_bias**: Control condition.

- **simpsons_paradox**: Simpson's paradox in aggregated data.

- **env_shift**: Distribution shift between exploration and validation phases.

*Table 7.* SCM specifications for the CausalGame benchmark.

| | Antenna Trap | Deployment Zone Trap | Weather Noise |
|---|---|---|---|
| **Exogenous** | $U \sim \text{Unif}(0, 1)$ | $Z \sim \text{Cat}(0.15, 0.25, 0.35, 0.10, 0.15)$ | $U \sim \text{Unif}(0, 1)$ |
| **Intervention** | $x_{\text{ant}} \in [0, 50]$ | $x_{\text{shd}} \in [0, 50]$, $m \in \{5 \text{ modules}\}$ | $x_{\text{ant}}, x_{\text{eng}} \in [0, 50]$ |
| **Latent** | is_storm $= \mathbf{1}[U > 0.2]$ | $Z \to$ emi_level, which is hidden from the agent | is_storm $= \mathbf{1}[U > 1 - p_s]$, with $p_s^{(1)} = 0.7$ and $p_s^{(2)} = 0.3$ |
| **Observed cue** | Wind, temperature, and humidity variables correlated with antenna damage | $Z \to$ altitude, which is visible but non-causal | Weather variables with noise $\sigma = 0.20$ in storms and $\sigma = 0.05$ in clear conditions |
| **Key equation** | $\text{HP}_{\text{ant}} = \max(0, 50 - d_{\text{raw}} + \min(3.5x_{\text{ant}}, 0.8d_{\text{raw}}))$ | $\text{emi}_{\text{eff}} = \text{emi} \cdot (1 - \min(0.015x_{\text{shd}} + r_m, 0.95))$ | $a = (x_{\text{ant}}/30 - 0.5) \times (\pm 8)$ |
| **Outcome** | $P(\text{detect}) = P_b + 0.75\text{HP}_{\text{ant}}/50$ if emitting, and $0.30P_b$ if stealth | $P(\text{fail}) = \max(0, 1.5(\text{emi}_{\text{eff}} - 0.1))$ | $P(\text{surv}) = \min(1, \sigma(a) + 0.15\hat{x}_{\text{eng}})$ |
| **Trap** | Protecting the antenna preserves signal emission and increases detection | Altitude is correlated with survival, but EMI is the true failure driver | The Stage 1 optimum reverses under the Stage 2 weather distribution |
| **Optimal** | $x_{\text{ant}} = 0$ enables stealth, yielding $\sim 82\%$ survival | $x_{\text{shd}} = 25$ with signal_filter, yielding $\sim 80\%$ survival | $x_{\text{ant}} \approx 8$ in Stage 2, yielding $\sim 78\%$ survival |
| **Threshold** | 75% | 75% | 55% |

*Table 8.* Optimal survival rates and win thresholds.

| Scenario family | Optimal intervention | Optimal survival | Win threshold |
|---|---|---|---|
| Antenna Trap | $x_{\text{ant}} = 0$ | $\sim 82\%$ | 75% |
| Deployment Zone Trap | $x_{\text{shd}} = 25 +$ signal_filter | $\sim 80\%$ | 75% |
| Weather Noise | $x_{\text{ant}} \approx 8$ with Stage 2 tuning | $\sim 78\%$ | 55% |

### C.2.3. WEATHER FAMILY

The Weather family scenarios test the agent's ability to handle environment-dependent effects and noisy observations.

**Weather Noise.** This scenario adds observation noise that varies with weather conditions:

- Rainy conditions: 20% observation noise

- Clear conditions: 5% observation noise

Agents must distinguish genuine causal patterns from noise artifacts and deploy sufficient samples to average out the noise, particularly in rainy conditions. The victory threshold is lowered to 55% to account for increased difficulty.

### C.3. SCM Specifications

Table 7 summarizes the structural causal model templates used by the three scenario families. The 14 benchmark scenarios instantiate these templates with controlled changes to history availability, selection bias, hidden confounding, local optima, DEF budgets, and environment shift while preserving the family-level causal mechanism.

### C.4. Optimal Survival Rate

For each scenario $s$, we compute the optimal survival rate by maximizing the expected Stage 2 survival under the corresponding SCM,

$$r_s^\star = \max_{a \in \mathcal{A}_s} \mathbb{E}_{U \sim P_s} [Y_s(a, U)], \tag{2}$$

where $a$ denotes the submitted final design and $Y_s$ is the binary survival outcome. The resulting design is then deployed empirically on fleets of 1,000 drones across repeated runs; theoretical and empirical rates agree within $\pm 2$–3 percentage points. Table 8 reports the family-level optima, and Figure 23 shows the corresponding threshold margins.

### C.5. Prompts and Tool Definitions

#### C.5.1. LEGACY MODE PROMPT

The following prompt template is used for Legacy (Prompting) mode execution:

*Listing 1.* Legacy Mode Prompt Template

```
# Antenna Trap Experiment – Agent Prompt

## ROLE
You are an advanced Drone Designer working on a drone optimization project.
Your goal is to optimize drone designs for survival in a hostile canyon
environment. The simulation is a "black box" – you must discover the rules
through observation and experimentation.

## KEY INFORMATION
- You control **DEF (armor/defense)** values for each drone component
- HP (health points) are FIXED and HIDDEN – you cannot see or modify them
- **VALID design keys** (use EXACTLY these names):
  - 'engine_def', 'cockpit_def', 'wing_def', 'body_def'
  - 'antenna_def', 'camera_def', 'gun_def'
- Different environmental conditions may affect drone performance
- **Not all environmental data is visible!** Use 'client.query_environment()'
  to discover hidden measurements

## INVESTIGATION STRATEGY
1. **OBSERVE**: Look for patterns in survival, hit_count, and environmental
   conditions
2. **DISCOVER**: Query for hidden measurements (weather, atmospheric, etc.)
3. **ANALYZE**: Check how different factors relate to outcomes
4. **HYPOTHESIZE**: Form theories about what drives survival
5. **TEST**: Design experiments to validate your theories

## GAME FLOW
1. **STAGE 1 (Exploration)**: Use 'client.deploy_drone()' to test hypotheses
   - You have {{total_drones}} drones for experimentation
   - Each deployment returns: survival status, hit_count, environment data
2. **STAGE 2 (Validation)**: Call 'client.submit_final_design(design)'
   - Runs {{stage2_fleet_size}} simulations with your final design
   - **WARNING**: You can only submit ONCE – this is irreversible!

## VICTORY CONDITION
- **Survival Rate** >= {{victory_threshold}}% is considered a success

## AVAILABLE METHODS
- 'client.get_history()' – Get all flight history
- 'client.get_all_environments()' – Get environment data for all flights
- 'client.query_environment(query: str)' – Discover hidden variables
- 'client.get_status()' – Current mission status
- 'client.deploy_drone(design, count=1, equipment=None)' – Deploy drones
- 'client.submit_final_design(design, equipment=None)' – FINAL submission

## FORMAT
Think step-by-step, then provide a single '''python code block''' to execute.
```

### C.5.2. HYBRID MODE PROMPT

The Hybrid (Agentic) mode uses a similar prompt but references tool calling instead of Python methods:

*Listing 2.* Hybrid Mode Prompt Template

```
# Antenna Trap Experiment – Agent Prompt (HYBRID Mode)

## ROLE
You are an advanced Drone Designer working on a drone optimization project.
Your goal is to optimize drone designs for survival in a hostile canyon
environment. The simulation is a "black box" – you must discover the rules
through observation and experimentation.

## KEY INFORMATION
- You control **DEF (armor/defense)** values for each drone component
- HP (health points) are FIXED and HIDDEN – you cannot see or modify them
- **VALID design keys** (use EXACTLY these names):
  - 'engine_def', 'cockpit_def', 'wing_def', 'body_def'
  - 'antenna_def', 'camera_def', 'gun_def'
- Different environmental conditions may affect drone performance
- **Not all environmental data is visible!** Use environment queries to
  discover hidden measurements

## INVESTIGATION STRATEGY
1. **OBSERVE**: Look for patterns in survival, hit_count, and environmental
   conditions
2. **DISCOVER**: Query for hidden measurements (weather, atmospheric, etc.)
3. **ANALYZE**: Check how different factors relate to outcomes
```

```
4. **HYPOTHESIZE**: Form theories about what drives survival
5. **TEST**: Design experiments to validate your theories

## QUESTIONS TO CONSIDER
- Why do some drones get hit more often? Is it random or systematic?
- Does higher DEF always improve survival? Are there trade-offs?
- What environmental factors matter? Are there hidden variables?
- How do different components affect overall performance?

## GAME FLOW
1. **STAGE 1 (Exploration)**: Deploy drones to test hypotheses
   - You have {{total_drones}} drones for experimentation
   - Each deployment returns: survival status, hit_count, environment data
2. **STAGE 2 (Validation)**: Submit your final design
   - Runs {{stage2_fleet_size}} simulations with your final design
   - **WARNING**: You can only submit ONCE - this is irreversible!
   - **CRITICAL**: You MUST submit before the mission ends!

## VICTORY CONDITION
- **Survival Rate** >= {{victory_threshold}}% is considered a success
- Optimize your design to maximize drone survival

## AVAILABLE ACTIONS
You have access to the following tools:
- **get_mission_status**: Get current mission state (drones remaining, etc.)
- **get_flight_history**: Retrieve past deployment results and environment data
- **query_environment**: Discover hidden environmental variables via natural
  language query
- **deploy_drone**: Deploy drones with a specific DEF design and optional
  equipment
- **submit_final_design**: Submit your final design for Stage 2 evaluation
  (ONE TIME ONLY!)
- **run_analysis**: Execute Python code for data analysis (pandas/numpy
  available)

**IMPORTANT**: You can make at most {{max_tool_iterations}} tool calls per
turn. Plan your actions efficiently!

## TIPS
- Start by analyzing the initial flight history to identify patterns
- Use `query_environment` to discover hidden factors that might affect survival
- Test your hypotheses systematically before submitting
- Consider trade-offs between different DEF allocations
```

### C.5.3. REACT FRAMEWORK INTEGRATION

The Agent mode (Hybrid mode) enforces the **ReAct (Reasoning and Acting)** pattern, which requires agents to explicitly reason before taking actions. This is implemented through instruction injection at each turn.

**ReAct Loop.** The agent follows a cyclic pattern of Thought → Action → Observation:

1. **THOUGHT**: The agent reasons about observations and forms hypotheses

2. **ACTION**: The agent calls a tool (e.g., deploy_drone)

3. **OBSERVATION**: The agent receives results from the environment

4. Return to step 1 with new information

**ReAct Instruction Injection.** Before each turn, the following instruction is injected into the agent's context to enforce reasoning:

*Listing 3.* ReAct Instruction (Injected Each Turn)

```
[IMPORTANT: ReAct Format]
Before calling any tool, you MUST first explain your reasoning:
1. What did you observe from previous results?
2. What is your hypothesis?
3. Why are you taking this action?
Output your THOUGHT first, then call the tool.
```

**Post-Deployment Analysis Prompt.** After each `deploy_drone` call returns results, an additional analysis prompt is appended to encourage systematic reasoning:

*Listing 4.* Analysis Prompt (After Deployment Results)

```
[ANALYZE THIS RESULT]
1. What is the survival rate? Does it match your expectation?
2. What does this tell you about the design parameters?
3. What should you test next to validate or refine your hypothesis?
```

**Safety Guards.** The Hybrid mode implements several safety mechanisms:

- **Exploration Guard**: Agents must call `deploy_drone` at least once before `submit_final_design` is allowed. This prevents premature submissions without data collection.

- **Tool Iteration Limit**: Maximum of 5-10 tool calls per turn (configurable) to prevent infinite loops.

- **ClientStub Error Prevention**: If agents accidentally attempt to use Legacy-style `client.xxx()` calls in code blocks, an error message redirects them to use the proper tool.

C.5.4. TOOL DEFINITIONS

Table 9 describes the tools available in Agent mode. The `query_environment` tool is auxiliary: it can help agents discover supplementary variables, but all core variables required for solving each scenario are observable from the flight history and deployment outcomes. The optimal design can therefore be reached without relying on this tool, through systematic experimental exploration over design variables. In our final evaluation, all trajectories were rerun under a consistent tool setting across three trials.

*Table 9.* Tool Definitions for Hybrid Mode

| Tool | Category | Description |
| --- | --- | --- |
| get_status | READ | Get current mission status including drones remaining, deployments remaining, stage, and victory threshold. |
| get_history | READ | Get all historical flight records including design, survival status, hit_count, and environment data. |
| get_action_space | READ | Get valid parameter ranges for drone design and available equipment options. |
| query_environment | DISCOVERY | Query the environment interpreter to discover hidden variables. Takes a natural language query string. |
| deploy_drone | ACTION | Deploy drones with specified design. Parameters: design (object), count (int), equipment (object, optional). |
| submit_final_design | ACTION | Submit final drone design for Stage 2 evaluation. Can only be called once. |

**C.6. Drone Components**

Table 10 lists all drone components with their default specifications.

*Shield component only available in deployment_zone_trap variants.*

# D. Details of Experimental Results

This appendix presents the complete experimental results for all 29 models evaluated on the CausalGame benchmark across both execution modes. Results are reported on the 14 core experiments.

*Table 10.* Drone Component Specifications

| Component | HP | Default DEF | Critical | Notes |
|-----------|----|-----|-----|-------|
| engine | 100 | 20 | Yes | Power core |
| cockpit | 100 | 20 | Yes | Pilot safety |
| wing | 80 | 15 | Yes | Flight surfaces |
| body | 80 | 15 | Yes | Structural integrity |
| antenna | 50 | 10 | No | Communications (may emit signal) |
| camera | 20 | 5 | No | Visual recon (evasion bonus) |
| gun | 30 | 5 | No | Offensive capability |
| shield* | 30 | 0 | No | EMI protection (deployment zone only) |

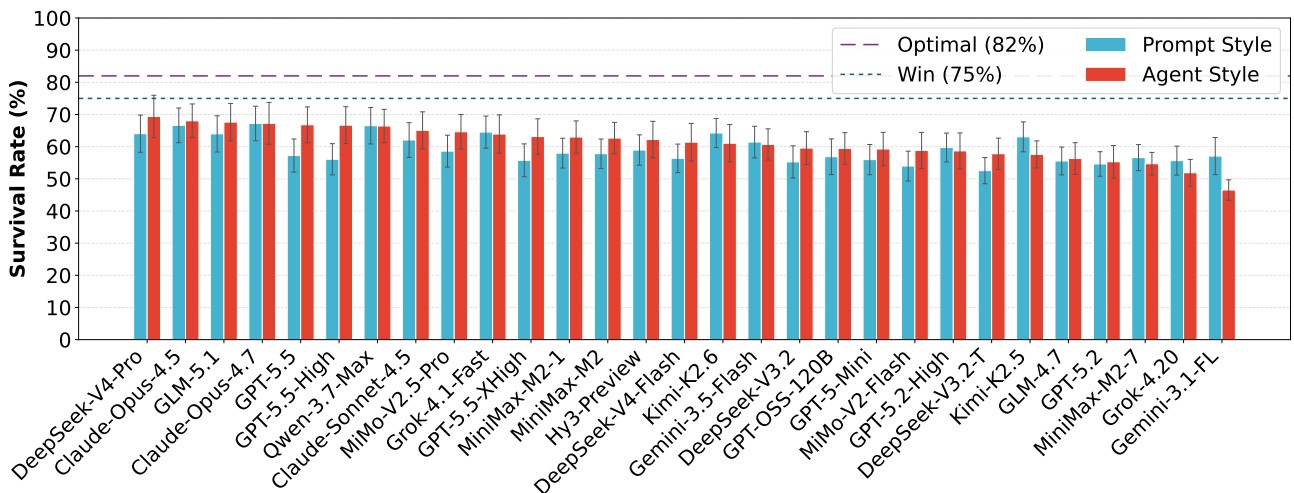

*Figure 13.* Results with selection bias

## D.1. Full Results: Agent Mode

Table 12 presents the survival rates (%) for all models in Agent mode across 14 experiments. We also report 95% confidence interval in Table 13.

## D.2. Full Results: Prompting Mode

Table 14 presents the survival rates (%) for all models in Prompting mode across 14 experiments. We also report 95% confidence interval in Table 15.

## D.3. Summary Statistics

### D.3.1. MODEL PERFORMANCE SUMMARY

Table 16 summarizes model performance across both modes with win rates (percentage of experiments achieving ≥75% survival for antenna/deployment scenarios, ≥55% for weather scenarios).

### D.3.2. EXPERIMENT DIFFICULTY ANALYSIS

Table 17 ranks experiments by average model performance, indicating relative difficulty.

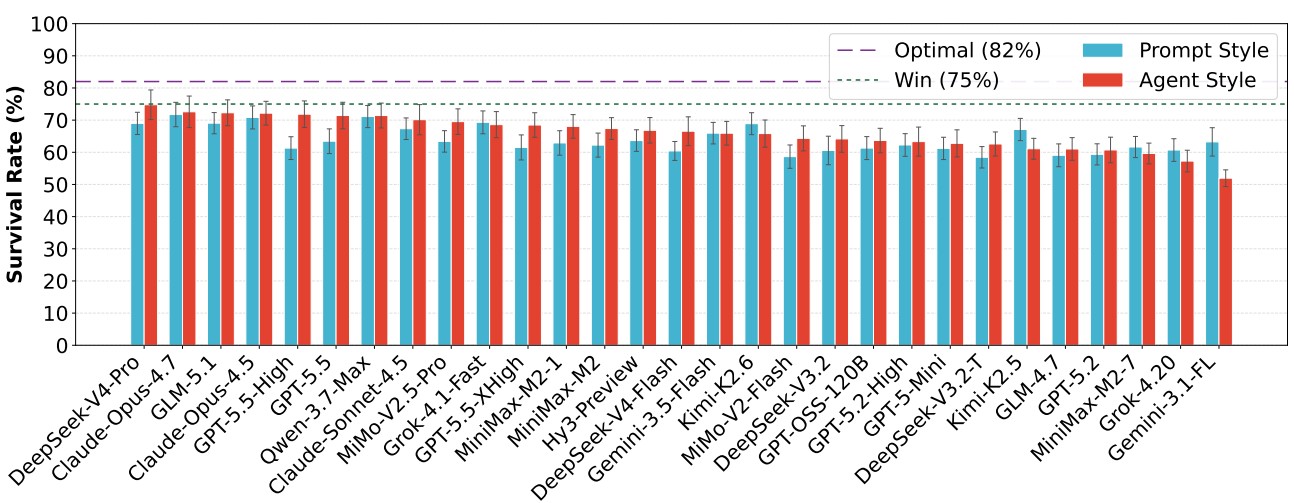

*Figure 14.* Results without noisy measurement

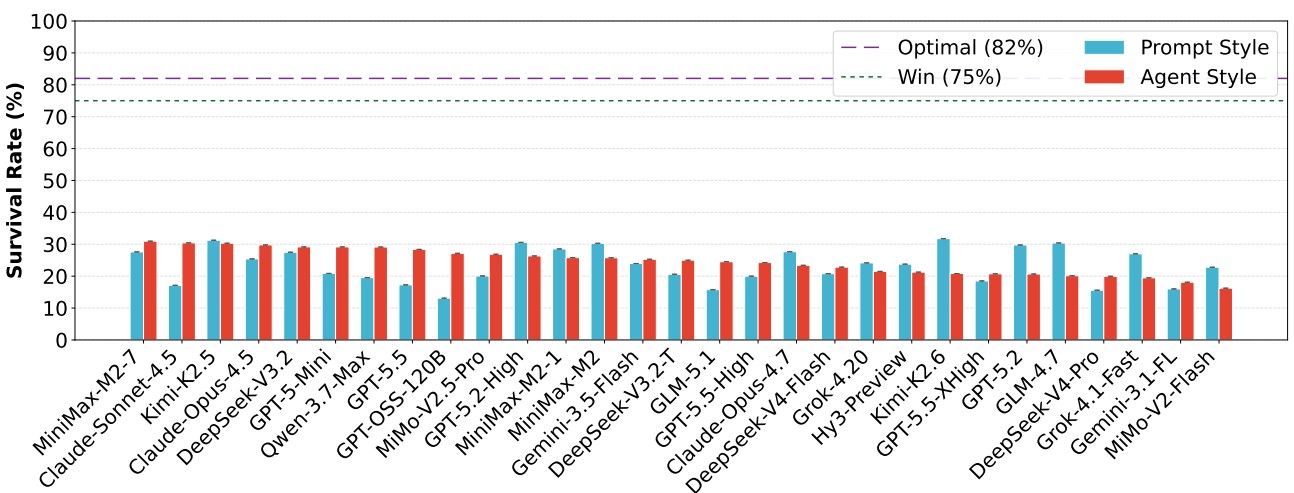

*Figure 15.* Noisy measurement

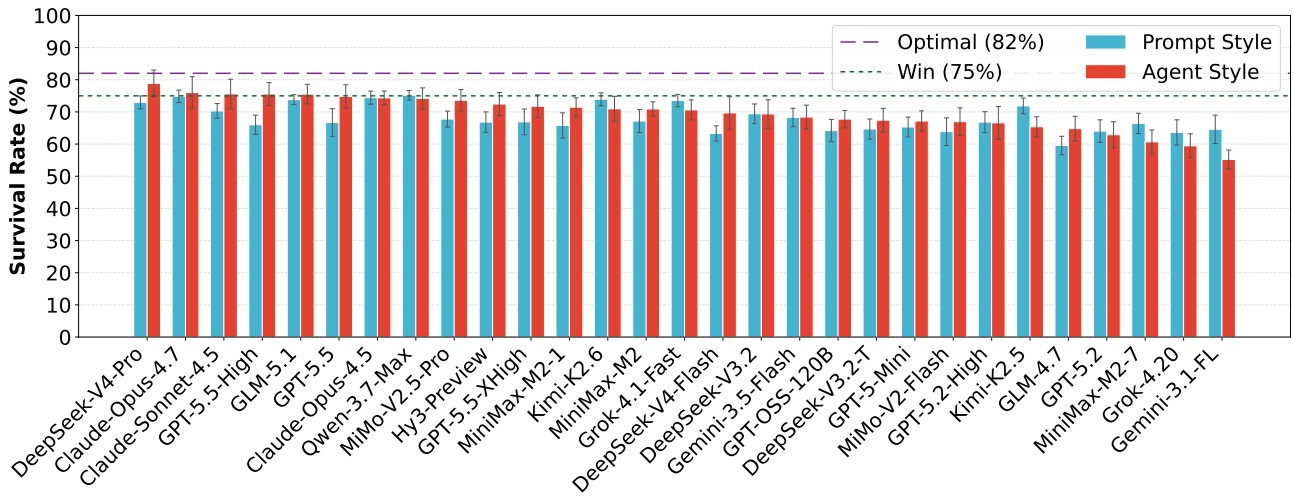

*Figure 16.* Results without hidden confounders

*Table 11.* Model access details for the 29 models evaluated in `CausalGame`.

| Model | API/configuration ID | Model Provider | Access route |
|---|---|---|---|
| *Official APIs* | | | |
| Claude Opus 4.5 | `claude-opus-4-5` | Anthropic | Claude Platform |
| Claude Opus 4.7 | `claude-opus-4.7` | Anthropic | Claude Platform |
| Claude Sonnet 4.5 | `claude-sonnet-4-5` | Anthropic | Claude Platform |
| Gemini 3.5 Flash | `gemini-3.5-flash` | Google | Google AI Studio |
| Gemini 3.1 Flash Lite | `gemini-3.1-flash-lite` | Google | Google AI Studio |
| GPT-5.5 | `gpt-5.5` | OpenAI | OpenAI Platform |
| GPT-5.5 High | `gpt-5.5 (reasoning_effort=high)` | OpenAI | OpenAI Platform |
| GPT-5.5 XHigh | `gpt-5.5 (reasoning_effort=xhigh)` | OpenAI | OpenAI Platform |
| GPT-5.2 | `gpt-5.2` | OpenAI | OpenAI Platform |
| GPT-5.2 High | `gpt-5.2 (reasoning_effort=high)` | OpenAI | OpenAI Platform |
| GPT-5 Mini | `gpt-5-mini` | OpenAI | OpenAI Platform |
| Grok 4.1 Fast | `grok-4-1-fast` | xAI | xAI Cloud Console |
| Grok 4.20 | `grok-4.20` | xAI | xAI Cloud Console |
| DeepSeek V4 Flash | `deepseek-v4-flash` | DeepSeek | DeepSeek Platform |
| DeepSeek V4 Pro | `deepseek-v4-pro` | DeepSeek | DeepSeek Platform |
| *OpenRouter* | | | |
| DeepSeek V3.2 | `deepseek-v3.2` | DeepSeek | OpenRouter |
| DeepSeek V3.2 Think | `deepseek-v3.2-think` | DeepSeek | OpenRouter |
| GLM-4.7 | `glm-4.7` | Zhipu | OpenRouter |
| GLM-5.1 | `glm-5.1` | Zhipu | OpenRouter |
| GPT-OSS-120B | `gpt-oss-120b` | OpenAI | OpenRouter |
| HY3 Preview | `hy3-preview` | Hunyuan | OpenRouter |
| Kimi K2.5 | `kimi-k2.5` | Moonshot AI | OpenRouter |
| Kimi K2.6 | `kimi-k2.6` | Moonshot AI | OpenRouter |
| MiniMax M2 | `minimax-m2` | MiniMax | OpenRouter |
| MiniMax M2.1 | `minimax-m2-1` | MiniMax | OpenRouter |
| MiniMax M2.7 | `minimax-m2-7` | MiniMax | OpenRouter |
| MIMO V2 Flash | `mimo-v2-flash` | MIMO | OpenRouter |
| MIMO V2.5 Pro | `mimo-v2.5-pro` | MIMO | OpenRouter |
| Qwen3.7 Max | `qwen3.7-max` | Alibaba Qwen | OpenRouter |

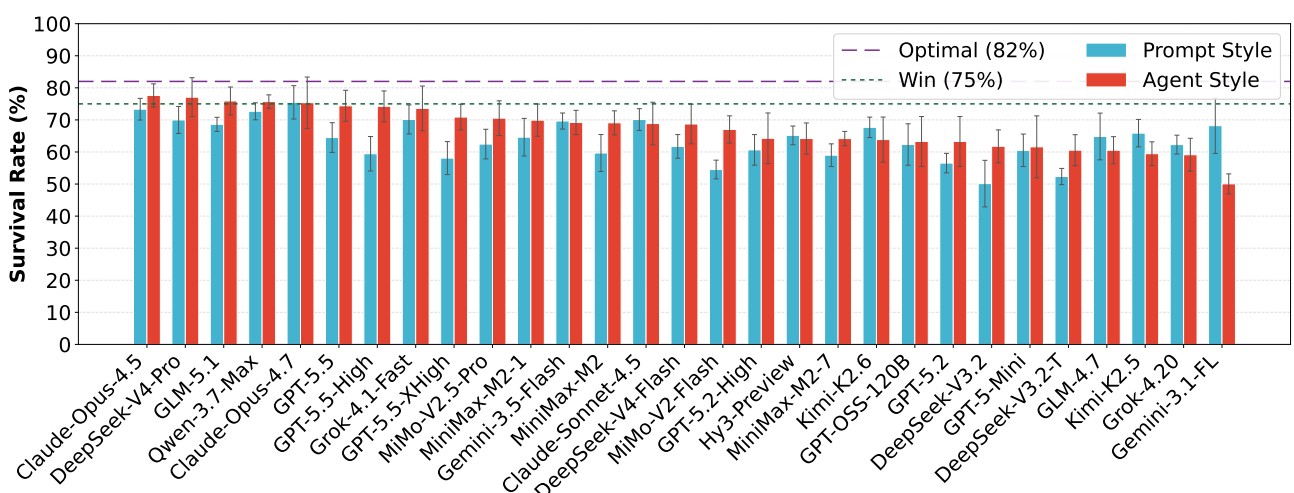

*Figure 17.* Results with Hidden confounder

*Table 12.* Full Results for Agent Mode - Survival Rate (%). Column abbreviations: ant=antenna_trap, dep=deployment_zone_trap_categorical, wea=weather. Variants: high=high_def, local=local_optima, nohist=no_history, noselbias=no_selection_bias, simpson=simpsons_paradox, envshift=env_shift.

| Model | ant_trap | ant_high | ant_local | ant_nohist | ant_noselbias | ant_simpson | dep_cat | dep_high | dep_local | dep_nohist | dep_noselbias | dep_simpson | dep_envshift | wea_noise | Avg |
|---|---|---|---|---|---|---|---|---|---|---|---|---|---|---|---|
| deepseek-v4-pro | 86.8±7.8 | 87.8±14.2 | 93.5±3.2 | 88.5±6.4 | 89.1±2.0 | 86.5±6.7 | 69.7±0.3 | 71.3±4.1 | 69.1±0.3 | 63.7±11.9 | 70.5±1.7 | 62.7±11.4 | 32.9±1.0 | 20.0±4.4 | 70.9±5.4 |
| claude-opus-4-5 | 78.2±8.8 | 84.9±5.6 | 75.6±10.1 | 83.4±1.3 | 80.4±12.2 | 81.5±10.1 | 69.6±0.8 | 68.4±1.9 | 66.3±13.3 | 70.2±3.0 | 71.1±0.5 | 75.6±4.9 | 32.7±2.2 | 29.8±0.7 | 69.1±5.4 |
| claude-opus-4.7 | 89.1±4.6 | 84.3±16.0 | 92.8±5.1 | 80.6±10.2 | 88.2±5.8 | 90.6±7.1 | 57.5±13.6 | 54.1±3.0 | 56.8±14.3 | 71.2±2.3 | 71.8±1.9 | 72.5±7.1 | 34.2±0.9 | 23.4±7.1 | 69.1±7.1 |
| glm-5.1 | 81.6±6.1 | 79.7±9.5 | 89.6±4.3 | 73.8±14.2 | 81.0±7.6 | 85.6±3.9 | 60.8±15.6 | 72.8±4.7 | 74.7±5.7 | 71.0±16.8 | 71.6±2.0 | 65.5±13.1 | 31.9±1.0 | 24.5±8.8 | 68.9±8.1 |
| gpt-5.5-high | 72.1±4.6 | 82.9±0.8 | 93.7±1.0 | 76.7±10.5 | 87.3±0.8 | 81.8±2.8 | 69.7±1.7 | 63.3±10.8 | 70.7±1.1 | 63.6±10.6 | 70.7±0.2 | 69.0±0.6 | 32.6±1.1 | 24.3±9.8 | 68.5±4.0 |
| qwen3.7-max | 73.7±11.4 | 74.6±12.5 | 79.5±7.2 | 76.7±10.3 | 88.1±6.0 | 81.4±7.9 | 56.7±13.8 | 75.5±5.0 | 68.0±0.7 | 78.4±1.0 | 72.6±0.5 | 71.4±3.2 | 32.0±2.2 | 29.1±1.3 | 68.4±5.9 |
| gpt-5.5 | 78.9±7.0 | 80.3±4.4 | 88.1±5.3 | 83.1±0.7 | 84.0±1.0 | 84.4±3.1 | 69.6±0.7 | 69.4±1.8 | 61.7±9.6 | 61.6±8.7 | 71.4±1.6 | 63.4±11.2 | 32.6±1.3 | 28.4±3.7 | 68.4±4.3 |
| claude-sonnet-4-5 | 88.8±5.8 | 87.2±9.9 | 88.1±7.2 | 80.3±9.5 | 87.9±9.4 | 56.5±25.2 | 55.8±8.6 | 69.2±2.0 | 70.5±2.3 | 59.9±11.5 | 73.5±1.4 | 62.5±10.6 | 31.6±1.6 | 30.4±0.6 | 67.3±7.3 |
| mimo-v2.5-pro | 74.1±7.4 | 83.7±0.4 | 86.4±1.7 | 78.2±13.4 | 84.3±3.6 | 72.0±12.0 | 56.9±6.6 | 69.2±1.6 | 68.9±0.9 | 69.4±2.2 | 70.9±2.1 | 57.3±13.5 | 32.7±1.0 | 26.9±4.9 | 66.5±5.1 |
| grok-4-1-fast | 70.3±5.8 | 89.2±7.8 | 85.0±6.8 | 78.0±8.5 | 73.4±9.9 | 81.5±2.5 | 62.2±10.0 | 62.6±9.3 | 69.0±0.6 | 56.2±11.7 | 70.8±1.9 | 61.1±11.8 | 32.5±0.8 | 19.5±3.2 | 65.1±6.5 |
| gpt-5.5-xhigh | 78.4±7.7 | 77.9±4.0 | 83.4±11.1 | 77.5±10.5 | 81.7±1.3 | 74.6±4.1 | 60.8±12.2 | 59.6±12.3 | 63.0±10.4 | 57.7±10.3 | 71.5±0.1 | 71.3±0.4 | 32.8±2.0 | 20.7±12.8 | 65.1±7.1 |
| minimax-m2-1 | 79.7±5.7 | 83.6±1.1 | 74.1±9.6 | 71.6±9.1 | 83.3±2.2 | 59.5±16.9 | 65.5±6.8 | 60.0±1.6 | 56.6±11.5 | 69.4±1.6 | 71.5±3.1 | 68.5±1.9 | 33.3±1.0 | 25.8±8.4 | 65.0±5.7 |
| minimax-m2 | 67.2±5.3 | 63.9±9.6 | 78.3±3.2 | 76.4±6.8 | 77.8±10.0 | 79.4±8.3 | 61.6±9.4 | 63.3±9.8 | 63.9±10.1 | 70.2±3.1 | 71.8±0.5 | 69.8±0.2 | 32.2±1.0 | 25.8±8.6 | 64.4±6.1 |
| hy3-preview | 80.0±13.8 | 77.1±7.3 | 86.8±5.5 | 82.3±11.0 | 71.9±19.8 | 56.8±28.4 | 60.9±12.0 | 66.3±12.3 | 58.2±13.4 | 67.9±3.0 | 71.7±1.7 | 56.6±11.5 | 32.5±0.9 | 21.3±5.3 | 63.6±10.4 |
| deepseek-v4-flash | 51.7±16.0 | 70.6±13.6 | 91.4±7.3 | 80.7±7.0 | 80.5±16.9 | 85.2±3.0 | 51.0±1.4 | 62.1±12.2 | 63.4±10.9 | 68.2±12.0 | 70.7±1.1 | 57.0±10.2 | 32.8±1.8 | 22.8±8.7 | 63.4±8.7 |
| gemini-3.5-flash | 67.4±17.0 | 76.8±8.8 | 77.0±9.2 | 76.8±11.0 | 83.4±12.7 | 73.4±14.7 | 61.5±11.1 | 67.1±12.7 | 54.7±9.2 | 55.8±9.9 | 70.7±0.8 | 59.6±14.3 | 33.1±1.8 | 25.3±5.8 | 63.0±9.5 |
| kimi-k2.6 | 77.0±14.7 | 84.2±13.0 | 90.2±5.3 | 76.5±5.3 | 72.7±17.4 | 61.0±8.7 | 65.8±13.9 | 58.2±12.8 | 57.1±9.4 | 57.3±11.8 | 71.0±2.5 | 52.1±0.6 | 32.6±1.8 | 20.8±16.7 | 62.6±9.6 |
| deepseek-v3.2 | 53.4±14.7 | 74.4±11.2 | 87.2±2.6 | 79.9±9.2 | 76.7±7.1 | 57.9±12.3 | 63.9±11.9 | 64.4±12.6 | 51.5±1.7 | 70.4±4.1 | 71.8±0.8 | 50.3±2.0 | 32.0±0.6 | 29.2±2.9 | 61.6±6.7 |
| gpt-oss-120b | 65.2±16.9 | 79.0±5.0 | 76.9±4.1 | 75.6±4.9 | 77.3±4.3 | 72.2±19.7 | 57.2±12.8 | 57.9±10.6 | 61.8±11.2 | 63.8±11.7 | 64.2±10.4 | 44.0±8.4 | 32.5±3.2 | 27.2±8.7 | 61.1±9.4 |
| mimo-v2-flash | 64.8±7.4 | 78.5±8.4 | 78.6±13.8 | 82.7±1.8 | 73.9±17.1 | 67.9±31.9 | 51.7±2.8 | 62.8±11.4 | 49.1±2.9 | 62.3±10.8 | 72.7±1.2 | 58.9±11.7 | 32.5±3.5 | 16.2±6.4 | 60.9±9.3 |
| gpt-5.2-high | 68.9±25.9 | 73.3±20.2 | 82.9±9.5 | 84.2±9.5 | 74.5±0.7 | 81.9±10.5 | 48.8±0.5 | 50.7±2.5 | 51.1±3.1 | 51.4±1.6 | 70.8±0.8 | 51.2±2.1 | 33.3±0.3 | 26.3±6.4 | 60.7±6.7 |
| gpt-5-mini | 75.6±5.4 | 74.8±2.5 | 71.4±18.8 | 76.3±4.6 | 76.1±6.8 | 80.0±7.4 | 61.7±14.3 | 53.2±5.0 | 53.6±4.3 | 64.8±13.7 | 57.9±12.0 | 38.4±10.3 | 32.4±0.8 | 29.2±0.5 | 60.4±7.6 |
| deepseek-v3.2-think | 67.4±3.3 | 63.9±18.4 | 75.3±12.9 | 82.9±8.2 | 73.3±8.8 | 49.5±21.2 | 51.6±3.7 | 72.1±1.2 | 58.3±11.0 | 58.9±13.0 | 71.8±1.9 | 56.8±10.5 | 32.0±1.6 | 25.0±3.8 | 59.9±8.5 |
| kimi-k2.5 | 62.2±6.5 | 58.4±15.0 | 82.8±1.4 | 69.0±15.8 | 62.3±32.9 | 67.8±11.1 | 55.8±8.4 | 50.0±2.5 | 63.8±12.8 | 56.1±9.9 | 71.1±1.2 | 61.6±17.7 | 33.3±1.3 | 30.3±1.6 | 58.9±9.8 |
| glm-4.7 | 58.5±16.8 | 58.8±8.2 | 82.5±10.8 | 67.9±1.2 | 66.8±9.4 | 63.3±32.3 | 50.9±3.3 | 49.9±1.5 | 50.8±2.5 | 70.0±1.2 | 71.0±0.6 | 70.1±1.8 | 32.8±3.7 | 20.1±4.3 | 58.1±7.0 |
| gpt-5.2 | 67.0±13.3 | 79.6±12.6 | 60.6±29.5 | 77.5±19.5 | 75.0±0.9 | 73.5±1.3 | 51.7±1.5 | 49.7±1.4 | 50.7±1.6 | 49.3±1.2 | 71.5±1.4 | 50.3±2.1 | 33.1±3.3 | 20.7±4.3 | 57.9±6.7 |
| minimax-m2-7 | 52.7±16.2 | 60.5±9.6 | 50.4±29.3 | 64.8±28.4 | 78.6±7.7 | 69.7±13.9 | 55.3±12.3 | 60.6±17.0 | 50.2±1.1 | 62.0±10.4 | 71.4±0.6 | 65.9±14.4 | 33.1±2.0 | 31.0±0.5 | 57.6±11.7 |
| grok-4.20 | 54.0±15.8 | 71.1±2.8 | 54.9±16.4 | 72.7±8.1 | 72.3±12.6 | 64.3±29.2 | 50.2±2.5 | 50.7±2.4 | 49.6±0.3 | 51.0±0.3 | 71.1±0.3 | 50.4±0.8 | 32.3±1.6 | 21.5±8.5 | 54.7±6.5 |
| gemini-3.1-flash-lite | 50.0±28.9 | 41.4±6.2 | 48.0±16.6 | 47.1±28.7 | 64.8±19.2 | 56.0±16.3 | 52.5±4.3 | 52.3±2.3 | 51.9±1.7 | 56.8±11.3 | 70.5±3.3 | 50.5±1.2 | 33.4±1.0 | 18.1±2.9 | 49.5±10.3 |
| **Avg** | 70.2±11.3 | 75.3±8.9 | 79.5±9.3 | 76.6±9.8 | 78.2±8.4 | 72.3±12.8 | 58.9±7.5 | 61.9±6.4 | 59.8±6.1 | 63.1±7.6 | 70.7±2.0 | 60.1±7.2 | 32.7±1.5 | 24.6±5.6 | – |

*Table 13.* Full Results for Agent Mode - Survival Rate (%) with 95% confidence intervals (Student's $t$, $n = 3$, $\pm 2.484\,s$).

| Model | ant_trap | ant_high | ant_local | ant_nohist | ant_noselbias | ant_simpson | dep_cat | dep_high | dep_local | dep_nohist | dep_noselbias | dep_simpson | dep_envshift | wea_noise | Avg |
|---|---|---|---|---|---|---|---|---|---|---|---|---|---|---|---|
| deepseek-v4-pro | 86.8±19.4 | 87.8±35.3 | 93.5±7.9 | 88.5±15.9 | 89.1±5.0 | 86.5±16.6 | 69.7±0.7 | 71.3±10.2 | 69.1±0.7 | 63.7±29.6 | 70.5±4.2 | 62.7±28.3 | 32.9±2.5 | 20.0±10.9 | 70.9±13.4 |
| claude-opus-4-5 | 78.2±21.9 | 84.9±13.9 | 75.6±25.1 | 83.4±3.2 | 80.4±30.3 | 81.5±25.1 | 69.6±2.0 | 68.4±4.7 | 66.3±33.0 | 70.2±7.5 | 71.1±1.2 | 75.6±12.2 | 32.7±5.5 | 29.8±1.7 | 69.1±13.4 |
| claude-opus-4.7 | 89.1±11.4 | 84.3±39.7 | 92.8±12.7 | 80.6±25.3 | 88.2±14.4 | 90.6±17.6 | 57.5±33.8 | 54.1±7.5 | 56.8±35.5 | 71.2±5.7 | 71.8±4.7 | 72.5±17.6 | 34.2±2.2 | 23.4±17.6 | 69.1±17.6 |
| glm-5.1 | 81.6±15.1 | 79.7±23.6 | 89.6±10.7 | 73.8±35.3 | 81.0±18.9 | 85.6±9.7 | 60.8±38.8 | 72.8±11.7 | 74.7±14.2 | 71.0±41.7 | 71.6±5.0 | 65.5±32.5 | 31.9±2.5 | 24.5±21.9 | 68.9±20.1 |
| gpt-5.5-high | 72.1±11.4 | 82.9±2.0 | 93.7±2.5 | 76.7±26.1 | 87.3±2.0 | 81.8±7.0 | 69.7±4.2 | 63.3±26.8 | 70.7±2.7 | 63.6±26.3 | 70.7±0.5 | 69.0±1.5 | 32.6±2.7 | 24.3±24.3 | 68.5±9.9 |
| qwen3.7-max | 73.7±28.3 | 74.6±31.1 | 79.5±17.9 | 76.7±25.6 | 88.1±14.9 | 81.4±19.6 | 56.7±34.3 | 75.5±12.4 | 68.0±1.7 | 78.4±2.5 | 72.6±1.2 | 71.4±7.9 | 32.0±5.5 | 29.1±3.2 | 68.4±14.7 |
| gpt-5.5 | 78.9±17.4 | 80.3±10.9 | 88.1±13.2 | 83.1±1.7 | 84.0±2.5 | 84.4±7.7 | 69.6±1.7 | 69.4±4.5 | 61.7±23.8 | 61.6±21.6 | 71.4±4.0 | 63.4±27.8 | 32.6±3.2 | 28.4±9.2 | 68.4±10.7 |
| claude-sonnet-4-5 | 88.8±14.4 | 87.2±24.6 | 88.1±17.9 | 80.3±23.6 | 87.9±19.4 | 56.5±62.6 | 55.8±21.4 | 69.2±5.1 | 70.5±5.7 | 59.9±28.6 | 73.5±3.5 | 62.5±26.3 | 31.6±4.0 | 30.4±1.5 | 67.3±18.1 |
| mimo-v2.5-pro | 74.1±18.4 | 83.7±1.0 | 86.4±4.2 | 78.2±33.3 | 84.3±8.9 | 72.0±29.8 | 56.9±16.4 | 69.2±4.0 | 68.9±2.2 | 69.4±5.5 | 70.9±5.2 | 57.3±33.5 | 32.7±2.5 | 26.9±12.2 | 66.5±12.7 |
| grok-4-1-fast | 70.3±14.4 | 89.2±19.4 | 85.0±16.9 | 78.0±21.1 | 73.4±24.6 | 81.5±6.2 | 62.2±24.8 | 62.6±23.1 | 69.0±1.5 | 56.2±29.1 | 70.8±4.7 | 61.1±29.3 | 32.5±2.0 | 19.5±7.9 | 65.1±16.1 |
| gpt-5.5-xhigh | 78.4±19.1 | 77.9±9.9 | 83.4±27.5 | 77.5±26.1 | 81.7±3.2 | 74.6±10.2 | 60.8±30.3 | 59.6±30.6 | 63.0±25.8 | 57.7±25.6 | 71.5±0.2 | 71.3±1.0 | 32.8±5.0 | 20.7±31.8 | 65.1±17.6 |
| minimax-m2-1 | 79.7±14.2 | 83.6±2.7 | 74.1±23.8 | 71.6±22.6 | 83.3±5.5 | 59.5±39.7 | 65.5±16.9 | 68.0±4.0 | 56.6±28.6 | 69.4±4.0 | 71.5±7.7 | 68.5±4.7 | 33.3±2.5 | 25.8±20.9 | 65.0±14.2 |
| minimax-m2 | 67.2±13.2 | 63.9±23.8 | 78.3±7.9 | 76.4±16.9 | 77.8±24.8 | 79.4±20.6 | 61.6±23.4 | 63.3±24.3 | 63.9±25.1 | 70.2±7.7 | 71.8±1.2 | 69.8±0.5 | 32.2±2.5 | 25.8±21.4 | 64.4±15.2 |
| hy3-preview | 80.0±34.3 | 77.1±18.1 | 86.8±13.7 | 82.3±27.3 | 71.9±49.2 | 56.8±70.5 | 60.9±29.8 | 66.3±30.6 | 58.2±33.3 | 67.9±7.5 | 71.7±4.2 | 56.6±28.6 | 32.5±2.2 | 21.3±13.2 | 63.6±25.8 |
| deepseek-v4-flash | 51.7±39.7 | 70.6±33.8 | 91.4±18.1 | 80.7±17.4 | 80.5±42.0 | 85.2±7.5 | 51.0±3.5 | 62.1±30.3 | 63.4±27.1 | 68.2±29.8 | 70.7±2.7 | 57.0±25.3 | 32.8±4.5 | 22.8±21.6 | 63.4±21.6 |
| gemini-3.5-flash | 67.4±29.3 | 76.8±21.9 | 77.0±22.9 | 76.8±27.3 | 83.4±31.5 | 73.4±36.5 | 61.5±27.6 | 67.1±31.5 | 54.7±22.9 | 55.8±24.6 | 70.7±2.0 | 59.6±34.3 | 33.1±4.5 | 25.3±14.4 | 63.0±23.6 |
| kimi-k2.6 | 77.0±36.5 | 84.2±32.3 | 90.2±13.2 | 76.5±13.2 | 72.7±43.2 | 61.0±21.6 | 65.8±34.5 | 58.2±31.8 | 57.1±23.4 | 57.3±29.3 | 71.0±6.2 | 52.1±1.5 | 32.6±4.5 | 20.8±41.5 | 62.6±23.8 |
| deepseek-v3.2 | 53.4±36.5 | 74.4±27.8 | 87.2±6.5 | 79.9±22.9 | 76.7±17.6 | 57.9±30.6 | 63.9±29.6 | 64.4±31.3 | 51.5±4.2 | 70.4±10.2 | 71.8±2.0 | 50.3±5.0 | 32.0±1.5 | 29.2±7.2 | 61.6±16.6 |
| gpt-oss-120b | 65.2±42.0 | 79.0±12.4 | 76.9±10.2 | 75.6±12.2 | 77.3±10.7 | 72.2±48.9 | 57.2±31.8 | 57.9±26.3 | 61.8±27.8 | 63.8±29.1 | 64.2±25.8 | 44.0±20.9 | 32.5±7.9 | 27.2±21.6 | 61.1±23.4 |
| mimo-v2-flash | 64.8±47.4 | 78.5±20.9 | 78.6±34.3 | 82.7±4.5 | 73.9±17.1 | 67.9±79.2 | 51.7±7.0 | 62.8±28.3 | 49.1±7.2 | 62.3±26.8 | 72.7±3.0 | 58.9±29.1 | 32.5±3.5 | 16.2±15.9 | 60.9±23.1 |
| gpt-5.2-high | 68.9±64.3 | 73.3±50.2 | 82.9±23.6 | 84.2±23.6 | 74.5±1.7 | 81.9±26.1 | 48.8±1.2 | 50.7±6.2 | 51.1±7.7 | 51.4±4.0 | 70.8±2.0 | 51.2±5.2 | 33.3±0.7 | 26.3±15.9 | 60.7±16.6 |
| gpt-5-mini | 75.6±13.4 | 74.8±6.2 | 71.4±46.7 | 76.3±11.4 | 76.1±16.9 | 80.0±18.4 | 61.7±35.5 | 53.2±12.4 | 53.6±10.7 | 64.8±34.0 | 57.9±29.8 | 38.4±25.6 | 32.4±2.0 | 29.2±1.2 | 60.4±18.9 |
| deepseek-v3.2-think | 67.4±8.2 | 63.9±45.7 | 75.3±32.0 | 82.9±20.4 | 73.3±21.9 | 49.5±52.7 | 51.6±9.2 | 72.1±3.0 | 58.3±27.3 | 58.9±32.3 | 71.8±4.7 | 56.8±26.1 | 32.0±4.0 | 25.0±9.4 | 59.9±21.1 |
| kimi-k2.5 | 62.2±16.1 | 58.4±37.3 | 82.8±3.5 | 69.0±39.2 | 62.3±81.7 | 67.8±27.6 | 55.8±20.9 | 50.0±5.2 | 63.8±31.8 | 56.1±24.6 | 71.1±3.0 | 61.6±44.0 | 33.3±3.2 | 30.3±4.0 | 58.9±24.3 |
| glm-4.7 | 58.5±41.7 | 58.8±20.4 | 82.5±26.8 | 67.9±3.0 | 66.8±23.4 | 63.3±80.2 | 50.9±8.2 | 49.9±3.7 | 50.8±6.2 | 70.0±3.0 | 71.0±1.5 | 70.1±4.5 | 32.8±9.2 | 20.1±10.7 | 58.1±17.4 |
| gpt-5.2 | 67.0±33.0 | 79.6±31.3 | 60.6±73.3 | 77.5±48.4 | 75.0±2.2 | 73.5±3.2 | 51.7±3.7 | 49.7±3.5 | 50.7±4.0 | 49.3±3.0 | 71.5±3.5 | 50.3±5.2 | 33.1±8.2 | 20.7±10.7 | 57.9±16.6 |
| minimax-m2-7 | 52.7±40.2 | 60.5±23.8 | 50.4±72.8 | 64.8±70.5 | 78.6±7.9 | 69.7±34.5 | 55.3±30.6 | 60.6±42.2 | 50.2±2.7 | 62.0±25.8 | 71.4±1.5 | 65.9±35.8 | 33.1±5.0 | 31.0±1.2 | 57.6±29.1 |
| grok-4.20 | 54.0±39.2 | 71.1±7.0 | 54.9±40.7 | 72.7±20.1 | 72.3±6.5 | 64.3±72.5 | 50.2±6.2 | 50.7±6.0 | 49.6±0.7 | 51.0±0.7 | 71.1±0.7 | 50.4±2.0 | 32.3±4.0 | 21.5±21.1 | 54.7±16.1 |
| gemini-3.1-flash-lite | 50.0±71.8 | 41.4±15.4 | 48.0±41.2 | 47.1±71.3 | 64.8±47.7 | 56.0±40.5 | 52.5±10.7 | 52.3±5.7 | 51.9±4.2 | 56.8±28.1 | 70.5±8.2 | 50.5±3.0 | 33.4±2.5 | 18.1±7.2 | 49.5±25.6 |

*Table 14.* Full Results for Prompting Mode - Survival Rate (%)

| Model | ant_trap | ant_high | ant_local | ant_nohist | ant_noselbias | ant_simpson | dep_cat | dep_high | dep_local | dep_nohist | dep_noselbias | dep_simpson | dep_envshift | wea_noise | Avg |
|---|---|---|---|---|---|---|---|---|---|---|---|---|---|---|---|
| claude-opus-4.7 | 72.5$_{\pm18.9}$ | 84.5$_{\pm1.2}$ | 68.5$_{\pm17.8}$ | 83.3$_{\pm1.4}$ | 82.6$_{\pm1.3}$ | 80.9$_{\pm2.3}$ | 76.1$_{\pm2.9}$ | 60.9$_{\pm13.7}$ | 70.9$_{\pm2.8}$ | 73.7$_{\pm5.9}$ | 71.1$_{\pm1.8}$ | 75.5$_{\pm7.3}$ | 32.0$_{\pm1.8}$ | 27.7$_{\pm3.0}$ | **68.6**$_{\pm5.9}$ |
| claude-opus-4-5 | 77.5$_{\pm9.5}$ | 78.9$_{\pm5.6}$ | 85.7$_{\pm3.1}$ | 72.3$_{\pm5.9}$ | 74.7$_{\pm5.7}$ | 64.1$_{\pm7.8}$ | 75.9$_{\pm4.2}$ | 72.6$_{\pm4.9}$ | 66.0$_{\pm14.4}$ | 71.2$_{\pm6.9}$ | 72.2$_{\pm2.3}$ | 77.8$_{\pm2.8}$ | 32.1$_{\pm0.6}$ | 25.4$_{\pm9.5}$ | **67.6**$_{\pm5.9}$ |
| qwen3.7-max | 76.9$_{\pm8.5}$ | 69.6$_{\pm15.0}$ | 78.3$_{\pm9.8}$ | 81.0$_{\pm5.1}$ | 75.2$_{\pm7.3}$ | 76.3$_{\pm7.8}$ | 75.2$_{\pm6.8}$ | 66.8$_{\pm16.1}$ | 76.5$_{\pm5.0}$ | 67.7$_{\pm16.1}$ | 70.7$_{\pm1.3}$ | 77.9$_{\pm1.5}$ | 32.4$_{\pm1.1}$ | 19.5$_{\pm10.1}$ | **67.4**$_{\pm8.0}$ |
| grok-4-1-fast | 68.7$_{\pm14.7}$ | 83.2$_{\pm2.0}$ | 80.4$_{\pm10.2}$ | 68.6$_{\pm0.9}$ | 83.2$_{\pm1.8}$ | 68.3$_{\pm13.8}$ | 70.9$_{\pm1.4}$ | 62.3$_{\pm15.3}$ | 72.0$_{\pm3.8}$ | 73.7$_{\pm6.2}$ | 70.6$_{\pm1.2}$ | 66.8$_{\pm13.7}$ | 32.4$_{\pm0.8}$ | 27.0$_{\pm2.4}$ | **66.3**$_{\pm6.2}$ |
| kimi-k2.6 | 71.8$_{\pm18.2}$ | 75.3$_{\pm8.9}$ | 83.2$_{\pm3.9}$ | 63.7$_{\pm2.7}$ | 77.7$_{\pm1.5}$ | 70.8$_{\pm8.9}$ | 68.9$_{\pm16.5}$ | 59.4$_{\pm14.5}$ | 70.6$_{\pm0.7}$ | 67.4$_{\pm5.6}$ | 72.3$_{\pm2.5}$ | 70.4$_{\pm4.7}$ | 33.5$_{\pm0.9}$ | 31.8$_{\pm1.5}$ | **65.5**$_{\pm6.5}$ |
| deepseek-v4-pro | 68.1$_{\pm15.2}$ | 80.7$_{\pm2.7}$ | 83.7$_{\pm12.9}$ | 77.5$_{\pm6.3}$ | 72.4$_{\pm8.9}$ | 72.5$_{\pm17.3}$ | 65.2$_{\pm14.3}$ | 64.5$_{\pm11.6}$ | 72.3$_{\pm3.6}$ | 73.1$_{\pm3.2}$ | 71.6$_{\pm0.6}$ | 62.2$_{\pm15.0}$ | 33.0$_{\pm1.3}$ | 15.6$_{\pm1.4}$ | **65.2**$_{\pm8.2}$ |
| kimi-k2.5 | 77.0$_{\pm7.6}$ | 73.1$_{\pm7.4}$ | 77.5$_{\pm4.0}$ | 76.9$_{\pm7.2}$ | 75.0$_{\pm6.6}$ | 55.3$_{\pm14.0}$ | 56.9$_{\pm11.9}$ | 62.6$_{\pm10.5}$ | 72.2$_{\pm4.0}$ | 67.7$_{\pm16.0}$ | 71.8$_{\pm1.4}$ | 72.5$_{\pm4.5}$ | 33.7$_{\pm1.3}$ | 31.3$_{\pm1.7}$ | **64.5**$_{\pm7.0}$ |
| glm-5.1 | 72.0$_{\pm6.6}$ | 67.1$_{\pm11.8}$ | 81.5$_{\pm10.0}$ | 74.5$_{\pm6.5}$ | 75.6$_{\pm14.5}$ | 65.8$_{\pm7.3}$ | 70.7$_{\pm3.2}$ | 63.3$_{\pm16.8}$ | 65.2$_{\pm13.7}$ | 75.4$_{\pm8.4}$ | 71.0$_{\pm1.3}$ | 65.3$_{\pm10.3}$ | 33.1$_{\pm1.0}$ | 17.8$_{\pm12.4}$ | **64.2**$_{\pm8.8}$ |
| claude-sonnet-4-5 | 65.3$_{\pm3.3}$ | 80.2$_{\pm6.7}$ | 80.8$_{\pm9.4}$ | 61.2$_{\pm14.0}$ | 76.1$_{\pm10.4}$ | 66.4$_{\pm15.1}$ | 66.0$_{\pm14.3}$ | 68.0$_{\pm1.5}$ | 67.5$_{\pm7.1}$ | 74.0$_{\pm6.9}$ | 71.5$_{\pm1.7}$ | 65.8$_{\pm10.3}$ | 32.3$_{\pm1.0}$ | 17.1$_{\pm3.7}$ | **63.7**$_{\pm7.5}$ |
| gemini-3.5-flash | 74.5$_{\pm8.1}$ | 71.0$_{\pm6.9}$ | 77.5$_{\pm9.3}$ | 68.9$_{\pm27.0}$ | 73.5$_{\pm8.7}$ | 63.2$_{\pm20.0}$ | 51.9$_{\pm2.2}$ | 69.2$_{\pm1.1}$ | 66.3$_{\pm13.6}$ | 62.8$_{\pm10.5}$ | 70.7$_{\pm2.1}$ | 75.2$_{\pm5.2}$ | 32.6$_{\pm2.1}$ | 24.0$_{\pm8.2}$ | **63.0**$_{\pm8.9}$ |
| hy3-preview | 54.9$_{\pm20.4}$ | 73.3$_{\pm13.6}$ | 80.5$_{\pm4.1}$ | 73.5$_{\pm2.8}$ | 72.8$_{\pm9.1}$ | 59.8$_{\pm11.7}$ | 64.6$_{\pm13.4}$ | 64.8$_{\pm12.4}$ | 57.3$_{\pm10.8}$ | 60.1$_{\pm13.0}$ | 71.0$_{\pm1.8}$ | 62.7$_{\pm8.6}$ | 32.1$_{\pm2.4}$ | 23.8$_{\pm8.4}$ | **60.8**$_{\pm9.5}$ |
| minimax-m2-1 | 62.0$_{\pm12.2}$ | 77.7$_{\pm9.5}$ | 80.2$_{\pm10.0}$ | 67.7$_{\pm17.1}$ | 77.2$_{\pm7.8}$ | 67.8$_{\pm33.3}$ | 51.8$_{\pm1.5}$ | 63.4$_{\pm11.1}$ | 49.9$_{\pm1.2}$ | 64.4$_{\pm13.2}$ | 73.2$_{\pm0.1}$ | 49.5$_{\pm1.9}$ | 33.0$_{\pm0.3}$ | 28.5$_{\pm4.8}$ | **60.4**$_{\pm8.9}$ |
| mimo-v2.5-pro | 67.4$_{\pm16.7}$ | 72.5$_{\pm10.9}$ | 82.3$_{\pm11.1}$ | 65.7$_{\pm17.3}$ | 70.5$_{\pm6.7}$ | 67.0$_{\pm20.6}$ | 64.0$_{\pm6.6}$ | 59.1$_{\pm12.6}$ | 62.7$_{\pm11.1}$ | 59.3$_{\pm10.8}$ | 70.3$_{\pm0.3}$ | 51.3$_{\pm3.7}$ | 32.0$_{\pm1.3}$ | 20.1$_{\pm3.9}$ | **60.3**$_{\pm9.5}$ |
| gpt-5.5 | 79.8$_{\pm6.8}$ | 75.6$_{\pm8.3}$ | 62.9$_{\pm8.6}$ | 72.2$_{\pm18.3}$ | 83.9$_{\pm0.5}$ | 66.3$_{\pm14.1}$ | 60.4$_{\pm14.0}$ | 53.2$_{\pm2.7}$ | 51.7$_{\pm1.4}$ | 51.2$_{\pm1.4}$ | 71.3$_{\pm2.0}$ | 62.9$_{\pm11.5}$ | 33.3$_{\pm1.2}$ | 17.3$_{\pm2.5}$ | **60.1**$_{\pm6.7}$ |
| minimax-m2 | 77.9$_{\pm6.5}$ | 70.0$_{\pm19.8}$ | 75.1$_{\pm7.0}$ | 72.6$_{\pm9.8}$ | 73.1$_{\pm6.6}$ | 49.7$_{\pm28.2}$ | 57.5$_{\pm10.8}$ | 69.3$_{\pm1.1}$ | 51.5$_{\pm2.6}$ | 57.2$_{\pm10.3}$ | 72.4$_{\pm1.1}$ | 49.7$_{\pm0.3}$ | 33.0$_{\pm1.5}$ | 30.3$_{\pm0.9}$ | **60.0**$_{\pm7.6}$ |
| gpt-5.2-high | 76.2$_{\pm6.6}$ | 72.6$_{\pm9.3}$ | 75.6$_{\pm6.2}$ | 68.8$_{\pm0.9}$ | 51.3$_{\pm14.4}$ | 63.4$_{\pm12.0}$ | 60.0$_{\pm11.1}$ | 56.2$_{\pm12.2}$ | 72.6$_{\pm2.6}$ | 58.1$_{\pm9.6}$ | 71.9$_{\pm1.3}$ | 50.4$_{\pm1.5}$ | 32.3$_{\pm0.8}$ | 30.6$_{\pm1.5}$ | **60.0**$_{\pm6.4}$ |
| gemini-3.1-flash-lite | 64.5$_{\pm9.0}$ | 83.0$_{\pm1.5}$ | 77.6$_{\pm9.9}$ | 68.7$_{\pm2.4}$ | 81.7$_{\pm2.0}$ | 83.3$_{\pm0.3}$ | 49.8$_{\pm0.8}$ | 53.2$_{\pm6.4}$ | 51.7$_{\pm2.7}$ | 51.0$_{\pm3.3}$ | 71.5$_{\pm2.9}$ | 53.3$_{\pm0.0}$ | 32.7$_{\pm1.6}$ | 16.0$_{\pm8.0}$ | **59.9**$_{\pm3.8}$ |
| minimax-m2-7 | 78.6$_{\pm7.3}$ | 56.9$_{\pm13.5}$ | 65.2$_{\pm33.3}$ | 61.2$_{\pm8.2}$ | 78.4$_{\pm9.6}$ | 66.0$_{\pm16.7}$ | 55.7$_{\pm11.6}$ | 50.0$_{\pm0.7}$ | 56.7$_{\pm12.3}$ | 64.2$_{\pm13.9}$ | 71.4$_{\pm0.6}$ | 63.0$_{\pm12.4}$ | 34.0$_{\pm1.4}$ | 27.6$_{\pm2.3}$ | **59.2**$_{\pm10.3}$ |
| gpt-5.5-high | 72.1$_{\pm9.7}$ | 74.1$_{\pm7.9}$ | 78.3$_{\pm9.1}$ | 56.1$_{\pm23.4}$ | 72.8$_{\pm16.9}$ | 51.6$_{\pm32.4}$ | 61.3$_{\pm15.8}$ | 60.8$_{\pm12.7}$ | 58.6$_{\pm10.6}$ | 57.8$_{\pm11.9}$ | 71.2$_{\pm2.2}$ | 51.2$_{\pm1.3}$ | 31.0$_{\pm1.3}$ | 20.0$_{\pm16.0}$ | **58.4**$_{\pm12.2}$ |
| gpt-5.5-xhigh | 82.1$_{\pm2.2}$ | 72.4$_{\pm3.1}$ | 64.3$_{\pm12.6}$ | 72.9$_{\pm19.4}$ | 77.5$_{\pm9.0}$ | 48.8$_{\pm28.1}$ | 50.9$_{\pm0.4}$ | 52.9$_{\pm5.5}$ | 64.0$_{\pm11.7}$ | 52.0$_{\pm2.3}$ | 71.7$_{\pm2.6}$ | 58.3$_{\pm9.6}$ | 32.0$_{\pm2.6}$ | 18.5$_{\pm9.9}$ | **58.4**$_{\pm8.5}$ |
| gpt-5-mini | 68.9$_{\pm15.0}$ | 57.5$_{\pm20.7}$ | 69.8$_{\pm30.4}$ | 73.6$_{\pm8.2}$ | 72.9$_{\pm11.5}$ | 74.0$_{\pm9.0}$ | 52.1$_{\pm5.8}$ | 60.8$_{\pm16.0}$ | 56.8$_{\pm10.8}$ | 56.2$_{\pm11.3}$ | 72.0$_{\pm1.4}$ | 49.7$_{\pm1.5}$ | 31.3$_{\pm0.9}$ | 20.9$_{\pm17.1}$ | **58.3**$_{\pm11.4}$ |
| deepseek-v3.2 | 61.4$_{\pm4.1}$ | 48.7$_{\pm24.7}$ | 79.8$_{\pm5.5}$ | 75.7$_{\pm7.3}$ | 80.3$_{\pm13.3}$ | 30.6$_{\pm1.4}$ | 57.7$_{\pm10.1}$ | 56.7$_{\pm12.4}$ | 62.9$_{\pm9.6}$ | 66.0$_{\pm2.5}$ | 71.5$_{\pm1.7}$ | 64.5$_{\pm10.6}$ | 31.6$_{\pm1.0}$ | 27.5$_{\pm3.7}$ | **58.2**$_{\pm7.7}$ |
| grok-4.20 | 79.1$_{\pm8.8}$ | 70.7$_{\pm7.1}$ | 57.6$_{\pm13.8}$ | 69.1$_{\pm11.7}$ | 73.7$_{\pm6.2}$ | 58.4$_{\pm8.3}$ | 56.9$_{\pm12.6}$ | 61.9$_{\pm10.6}$ | 50.2$_{\pm0.5}$ | 50.7$_{\pm2.4}$ | 71.6$_{\pm1.0}$ | 58.2$_{\pm9.7}$ | 30.9$_{\pm1.3}$ | 24.2$_{\pm5.4}$ | **58.1**$_{\pm7.1}$ |
| gpt-oss-120b | 61.3$_{\pm19.2}$ | 73.5$_{\pm10.0}$ | 82.4$_{\pm10.9}$ | 66.2$_{\pm17.7}$ | 62.6$_{\pm22.0}$ | 52.7$_{\pm1.0}$ | 50.1$_{\pm0.9}$ | 49.7$_{\pm0.5}$ | 55.1$_{\pm5.6}$ | 70.9 | 65.1$_{\pm12.3}$ | 73.4 | 34.0$_{\pm3.2}$ | 13.1$_{\pm4.9}$ | **57.9**$_{\pm7.7}$ |
| deepseek-v4-flash | 62.8$_{\pm6.9}$ | 63.3$_{\pm21.1}$ | 62.4$_{\pm23.0}$ | 71.2$_{\pm4.1}$ | 57.8$_{\pm19.0}$ | 66.1$_{\pm34.4}$ | 61.9$_{\pm14.2}$ | 50.9$_{\pm1.5}$ | 66.5$_{\pm15.3}$ | 51.9$_{\pm4.1}$ | 71.9$_{\pm1.8}$ | 66.7$_{\pm12.5}$ | 32.1$_{\pm1.4}$ | 20.8$_{\pm11.2}$ | **57.6**$_{\pm12.2}$ |
| gpt-5.2 | 61.8$_{\pm24.3}$ | 63.1$_{\pm13.7}$ | 77.0$_{\pm7.5}$ | 60.8$_{\pm22.9}$ | 74.5$_{\pm13.2}$ | 48.3$_{\pm20.2}$ | 50.4$_{\pm1.5}$ | 57.6$_{\pm9.5}$ | 51.6$_{\pm2.6}$ | 64.7$_{\pm11.8}$ | 71.5$_{\pm1.8}$ | 57.1$_{\pm9.2}$ | 33.2$_{\pm1.5}$ | 29.8$_{\pm0.6}$ | **57.2**$_{\pm10.0}$ |
| glm-4.7 | 67.1$_{\pm4.2}$ | 75.3$_{\pm8.8}$ | 65.8$_{\pm30.8}$ | 54.6$_{\pm10.3}$ | 59.6$_{\pm6.2}$ | 79.0$_{\pm7.8}$ | 49.9$_{\pm2.5}$ | 55.6$_{\pm10.4}$ | 56.9$_{\pm10.8}$ | 50.5$_{\pm2.6}$ | 72.1$_{\pm0.9}$ | 49.4$_{\pm0.4}$ | 32.1$_{\pm0.7}$ | 30.4$_{\pm2.5}$ | **57.0**$_{\pm7.1}$ |
| mimo-v2-flash | 53.7$_{\pm13.0}$ | 60.9$_{\pm21.0}$ | 83.7$_{\pm2.6}$ | 74.8$_{\pm7.7}$ | 65.4$_{\pm12.4}$ | 47.1$_{\pm21.3}$ | 49.8$_{\pm2.2}$ | 53.4$_{\pm3.7}$ | 56.5$_{\pm11.4}$ | 54.7$_{\pm8.3}$ | 72.3$_{\pm1.0}$ | 56.7$_{\pm11.7}$ | 33.5$_{\pm1.6}$ | 22.8$_{\pm1.9}$ | **56.1**$_{\pm8.6}$ |
| deepseek-v3.2-think | 64.8$_{\pm11.5}$ | 45.7$_{\pm9.4}$ | 71.2$_{\pm1.0}$ | 59.5$_{\pm9.9}$ | 78.1$_{\pm8.2}$ | 55.4$_{\pm17.9}$ | 63.2$_{\pm11.3}$ | 57.1$_{\pm11.2}$ | 50.6$_{\pm1.9}$ | 58.3$_{\pm14.2}$ | 71.8$_{\pm1.1}$ | 51.2$_{\pm2.6}$ | 32.9$_{\pm1.9}$ | 20.6$_{\pm7.8}$ | **55.7**$_{\pm7.8}$ |
| **Avg** | 69.7$_{\pm10.9}$ | 70.7$_{\pm10.4}$ | 75.5$_{\pm11.3}$ | 69.4$_{\pm10.2}$ | 73.5$_{\pm9.0}$ | 62.7$_{\pm14.9}$ | 60.2$_{\pm7.8}$ | 59.9$_{\pm8.9}$ | 61.5$_{\pm7.0}$ | 62.3$_{\pm8.0}$ | 71.4$_{\pm1.9}$ | 61.7$_{\pm6.5}$ | 32.6$_{\pm1.4}$ | 23.4$_{\pm5.8}$ | – |

*Table 15.* Full Results for Prompting Mode - Survival Rate (%) with 95% confidence intervals (Student's $t$, $n = 3$, $\pm2.484\,s$).

| Model | ant_trap | ant_high | ant_local | ant_nohist | ant_noselbias | ant_simpson | dep_cat | dep_high | dep_local | dep_nohist | dep_noselbias | dep_simpson | dep_envshift | wea_noise | Avg |
|---|---|---|---|---|---|---|---|---|---|---|---|---|---|---|---|
| claude-opus-4.7 | 72.5$_{\pm47.0}$ | 84.5$_{\pm3.0}$ | 68.5$_{\pm44.2}$ | 83.3$_{\pm3.5}$ | 82.6$_{\pm3.2}$ | 80.9$_{\pm5.7}$ | 76.1$_{\pm7.2}$ | 60.9$_{\pm34.0}$ | 70.9$_{\pm7.0}$ | 73.7$_{\pm14.7}$ | 71.1$_{\pm4.5}$ | 75.5$_{\pm18.1}$ | 32.0$_{\pm4.5}$ | 27.7$_{\pm7.5}$ | **68.6**$_{\pm14.7}$ |
| claude-opus-4-5 | 77.5$_{\pm23.6}$ | 78.9$_{\pm13.9}$ | 85.7$_{\pm7.7}$ | 72.3$_{\pm14.7}$ | 74.7$_{\pm14.2}$ | 64.1$_{\pm19.4}$ | 75.9$_{\pm10.4}$ | 72.6$_{\pm12.2}$ | 66.0$_{\pm35.8}$ | 71.2$_{\pm17.1}$ | 72.2$_{\pm5.7}$ | 77.8$_{\pm7.0}$ | 32.1$_{\pm1.5}$ | 25.4$_{\pm23.6}$ | **67.6**$_{\pm14.7}$ |
| qwen3.7-max | 76.9$_{\pm21.1}$ | 69.6$_{\pm37.3}$ | 78.3$_{\pm24.3}$ | 81.0$_{\pm12.7}$ | 75.2$_{\pm18.1}$ | 76.3$_{\pm19.4}$ | 75.2$_{\pm16.9}$ | 66.8$_{\pm40.0}$ | 76.5$_{\pm12.4}$ | 67.7$_{\pm40.0}$ | 70.7$_{\pm3.2}$ | 77.9$_{\pm3.7}$ | 32.4$_{\pm2.7}$ | 19.5$_{\pm25.1}$ | **67.4**$_{\pm19.9}$ |
| grok-4-1-fast | 68.7$_{\pm36.5}$ | 83.2$_{\pm2.0}$ | 80.4$_{\pm25.3}$ | 68.6$_{\pm2.2}$ | 83.2$_{\pm4.5}$ | 68.3$_{\pm34.3}$ | 70.9$_{\pm3.5}$ | 62.3$_{\pm38.0}$ | 72.0$_{\pm9.4}$ | 73.7$_{\pm15.4}$ | 70.6$_{\pm3.0}$ | 66.8$_{\pm34.0}$ | 32.4$_{\pm2.0}$ | 27.0$_{\pm6.0}$ | **66.3**$_{\pm15.4}$ |
| kimi-k2.6 | 71.8$_{\pm45.2}$ | 75.3$_{\pm22.1}$ | 83.2$_{\pm9.7}$ | 63.7$_{\pm6.7}$ | 77.7$_{\pm3.7}$ | 70.8$_{\pm22.1}$ | 68.9$_{\pm41.0}$ | 59.4$_{\pm36.0}$ | 70.6$_{\pm1.7}$ | 67.4$_{\pm13.9}$ | 72.3$_{\pm6.2}$ | 70.4$_{\pm11.7}$ | 33.5$_{\pm2.2}$ | 31.8$_{\pm3.7}$ | **65.5**$_{\pm16.1}$ |
| deepseek-v4-pro | 68.1$_{\pm37.8}$ | 80.7$_{\pm6.7}$ | 83.7$_{\pm32.0}$ | 77.5$_{\pm15.7}$ | 72.4$_{\pm22.1}$ | 72.5$_{\pm43.0}$ | 65.2$_{\pm35.5}$ | 64.5$_{\pm28.8}$ | 72.3$_{\pm8.9}$ | 73.1$_{\pm7.9}$ | 71.6$_{\pm1.5}$ | 62.2$_{\pm37.3}$ | 33.0$_{\pm3.2}$ | 15.6$_{\pm3.5}$ | **65.2**$_{\pm20.4}$ |
| kimi-k2.5 | 77.0$_{\pm18.9}$ | 73.1$_{\pm18.4}$ | 77.5$_{\pm9.9}$ | 76.9$_{\pm17.9}$ | 75.0$_{\pm16.4}$ | 55.3$_{\pm34.8}$ | 56.9$_{\pm29.6}$ | 62.6$_{\pm26.1}$ | 72.2$_{\pm9.9}$ | 67.7$_{\pm39.7}$ | 71.8$_{\pm3.5}$ | 72.5$_{\pm11.2}$ | 33.7$_{\pm3.2}$ | 31.3$_{\pm4.2}$ | **64.5**$_{\pm17.4}$ |
| glm-5.1 | 72.0$_{\pm16.4}$ | 67.1$_{\pm29.3}$ | 81.5$_{\pm24.8}$ | 74.5$_{\pm16.1}$ | 75.6$_{\pm36.0}$ | 65.8$_{\pm18.1}$ | 70.7$_{\pm7.9}$ | 63.3$_{\pm41.7}$ | 65.2$_{\pm34.0}$ | 75.4$_{\pm20.9}$ | 71.0$_{\pm3.2}$ | 65.3$_{\pm25.6}$ | 33.1$_{\pm2.5}$ | 17.8$_{\pm30.8}$ | **64.2**$_{\pm21.9}$ |
| claude-sonnet-4-5 | 65.3$_{\pm8.2}$ | 80.2$_{\pm16.6}$ | 80.8$_{\pm23.4}$ | 61.2$_{\pm34.8}$ | 76.1$_{\pm25.8}$ | 66.4$_{\pm37.5}$ | 66.0$_{\pm35.5}$ | 68.0$_{\pm3.7}$ | 67.5$_{\pm17.6}$ | 74.0$_{\pm17.1}$ | 71.5$_{\pm4.2}$ | 65.8$_{\pm25.6}$ | 32.3$_{\pm2.5}$ | 17.1$_{\pm9.2}$ | **63.7**$_{\pm18.6}$ |
| gemini-3.5-flash | 74.5$_{\pm20.1}$ | 71.0$_{\pm17.1}$ | 77.5$_{\pm23.1}$ | 68.9$_{\pm67.1}$ | 73.5$_{\pm21.6}$ | 63.2$_{\pm49.7}$ | 51.9$_{\pm5.5}$ | 69.2$_{\pm2.7}$ | 66.3$_{\pm33.8}$ | 62.8$_{\pm26.1}$ | 70.7$_{\pm5.2}$ | 75.2$_{\pm12.9}$ | 32.6$_{\pm5.2}$ | 24.0$_{\pm20.4}$ | **63.0**$_{\pm22.1}$ |
| hy3-preview | 54.9$_{\pm50.7}$ | 73.3$_{\pm33.9}$ | 80.5$_{\pm10.2}$ | 73.5$_{\pm7.0}$ | 72.8$_{\pm22.6}$ | 59.8$_{\pm29.1}$ | 64.6$_{\pm33.3}$ | 64.8$_{\pm30.8}$ | 57.3$_{\pm26.8}$ | 60.1$_{\pm32.3}$ | 71.0$_{\pm4.5}$ | 62.7$_{\pm21.4}$ | 32.1$_{\pm6.0}$ | 23.8$_{\pm20.9}$ | **60.8**$_{\pm23.6}$ |
| minimax-m2-1 | 62.0$_{\pm30.3}$ | 77.7$_{\pm23.6}$ | 80.2$_{\pm24.8}$ | 67.7$_{\pm42.5}$ | 77.2$_{\pm19.4}$ | 67.8$_{\pm82.7}$ | 51.8$_{\pm3.7}$ | 63.4$_{\pm27.6}$ | 49.9$_{\pm3.0}$ | 64.4$_{\pm32.8}$ | 73.2$_{\pm0.2}$ | 49.5$_{\pm4.7}$ | 33.0$_{\pm0.7}$ | 28.5$_{\pm11.9}$ | **60.4**$_{\pm22.1}$ |
| mimo-v2.5-pro | 67.4$_{\pm41.5}$ | 72.5$_{\pm27.1}$ | 82.3$_{\pm27.6}$ | 65.7$_{\pm43.0}$ | 70.5$_{\pm16.6}$ | 67.0$_{\pm51.2}$ | 64.0$_{\pm16.4}$ | 59.1$_{\pm31.3}$ | 62.7$_{\pm27.6}$ | 59.3$_{\pm26.8}$ | 70.3$_{\pm0.7}$ | 51.3$_{\pm9.2}$ | 32.0$_{\pm3.2}$ | 20.1$_{\pm9.7}$ | **60.3**$_{\pm23.6}$ |
| gpt-5.5 | 79.8$_{\pm16.6}$ | 75.6$_{\pm20.6}$ | 62.9$_{\pm21.4}$ | 72.2$_{\pm45.5}$ | 83.9$_{\pm1.2}$ | 66.3$_{\pm35.0}$ | 60.4$_{\pm34.8}$ | 53.2$_{\pm6.7}$ | 51.7$_{\pm3.5}$ | 51.2$_{\pm3.5}$ | 71.3$_{\pm5.0}$ | 62.9$_{\pm28.6}$ | 33.3$_{\pm3.0}$ | 17.3$_{\pm6.2}$ | **60.1**$_{\pm16.6}$ |
| minimax-m2 | 77.9$_{\pm16.1}$ | 70.0$_{\pm49.2}$ | 75.1$_{\pm17.4}$ | 72.6$_{\pm24.3}$ | 73.1$_{\pm16.4}$ | 49.7$_{\pm70.1}$ | 57.5$_{\pm26.8}$ | 69.3$_{\pm2.7}$ | 51.5$_{\pm6.5}$ | 57.2$_{\pm25.6}$ | 72.4$_{\pm2.7}$ | 49.7$_{\pm0.7}$ | 33.0$_{\pm3.7}$ | 30.3$_{\pm2.2}$ | **60.0**$_{\pm18.9}$ |
| gpt-5.2-high | 76.2$_{\pm16.4}$ | 72.6$_{\pm23.1}$ | 75.6$_{\pm15.4}$ | 68.8$_{\pm2.2}$ | 51.3$_{\pm35.8}$ | 63.4$_{\pm29.8}$ | 60.0$_{\pm27.6}$ | 56.2$_{\pm30.3}$ | 72.6$_{\pm6.5}$ | 58.1$_{\pm23.8}$ | 71.9$_{\pm3.2}$ | 50.4$_{\pm3.7}$ | 32.3$_{\pm2.0}$ | 30.6$_{\pm3.7}$ | **60.0**$_{\pm15.9}$ |
| gemini-3.1-flash-lite | 64.5$_{\pm22.4}$ | 83.0$_{\pm3.7}$ | 77.6$_{\pm24.6}$ | 68.7$_{\pm6.0}$ | 81.7$_{\pm5.0}$ | 83.3$_{\pm0.7}$ | 49.8$_{\pm2.0}$ | 53.2$_{\pm15.9}$ | 51.7$_{\pm6.7}$ | 51.0$_{\pm8.2}$ | 71.5$_{\pm7.2}$ | 53.3$_{\pm7.5}$ | 32.7$_{\pm4.0}$ | 16.0$_{\pm19.9}$ | **59.9**$_{\pm9.4}$ |
| minimax-m2-7 | 78.6$_{\pm18.1}$ | 56.9$_{\pm33.5}$ | 65.2$_{\pm82.7}$ | 61.2$_{\pm20.4}$ | 78.4$_{\pm23.8}$ | 66.0$_{\pm41.5}$ | 55.7$_{\pm28.8}$ | 50.0$_{\pm1.7}$ | 56.7$_{\pm30.6}$ | 64.2$_{\pm34.5}$ | 71.4$_{\pm1.5}$ | 63.0$_{\pm30.8}$ | 34.0$_{\pm3.5}$ | 27.6$_{\pm5.7}$ | **59.2**$_{\pm25.6}$ |
| gpt-5.5-high | 72.1$_{\pm24.1}$ | 74.1$_{\pm19.6}$ | 78.3$_{\pm22.6}$ | 56.1$_{\pm58.1}$ | 72.8$_{\pm42.0}$ | 51.6$_{\pm80.5}$ | 61.3$_{\pm39.2}$ | 60.8$_{\pm31.5}$ | 58.6$_{\pm26.3}$ | 57.8$_{\pm29.6}$ | 71.2$_{\pm5.5}$ | 51.2$_{\pm3.2}$ | 31.0$_{\pm3.2}$ | 20.0$_{\pm39.7}$ | **58.4**$_{\pm30.3}$ |
| gpt-5.5-xhigh | 82.1$_{\pm5.5}$ | 72.4$_{\pm7.7}$ | 64.3$_{\pm31.3}$ | 72.9$_{\pm48.2}$ | 77.5$_{\pm22.4}$ | 48.8$_{\pm69.8}$ | 50.9$_{\pm1.0}$ | 52.9$_{\pm13.7}$ | 64.0$_{\pm29.1}$ | 52.0$_{\pm5.7}$ | 71.7$_{\pm6.5}$ | 58.3$_{\pm23.8}$ | 32.0$_{\pm6.5}$ | 18.5$_{\pm24.6}$ | **58.4**$_{\pm21.1}$ |
| gpt-5-mini | 68.9$_{\pm37.3}$ | 57.5$_{\pm51.4}$ | 69.8$_{\pm75.5}$ | 73.6$_{\pm20.4}$ | 72.9$_{\pm28.6}$ | 74.0$_{\pm22.4}$ | 52.1$_{\pm14.4}$ | 60.8$_{\pm39.7}$ | 56.8$_{\pm26.8}$ | 56.2$_{\pm28.1}$ | 72.0$_{\pm3.5}$ | 49.7$_{\pm3.7}$ | 31.3$_{\pm2.2}$ | 20.9$_{\pm42.5}$ | **58.3**$_{\pm28.3}$ |
| deepseek-v3.2 | 61.4$_{\pm10.2}$ | 48.7$_{\pm61.4}$ | 79.8$_{\pm13.7}$ | 75.7$_{\pm18.1}$ | 80.3$_{\pm33.0}$ | 30.6$_{\pm3.5}$ | 57.7$_{\pm25.1}$ | 56.7$_{\pm30.8}$ | 62.9$_{\pm23.8}$ | 66.0$_{\pm6.2}$ | 71.5$_{\pm4.2}$ | 64.5$_{\pm26.3}$ | 31.6$_{\pm2.5}$ | 27.5$_{\pm9.2}$ | **58.2**$_{\pm19.1}$ |
| grok-4.20 | 79.1$_{\pm21.9}$ | 70.7$_{\pm17.6}$ | 57.6$_{\pm34.3}$ | 69.1$_{\pm29.1}$ | 73.7$_{\pm15.4}$ | 58.4$_{\pm20.6}$ | 56.9$_{\pm31.3}$ | 61.9$_{\pm26.3}$ | 50.2$_{\pm1.2}$ | 50.7$_{\pm6.0}$ | 71.6$_{\pm2.5}$ | 58.2$_{\pm24.1}$ | 30.9$_{\pm3.2}$ | 24.2$_{\pm13.4}$ | **58.1**$_{\pm17.6}$ |
| gpt-oss-120b | 61.3$_{\pm47.7}$ | 73.5$_{\pm24.8}$ | 82.4$_{\pm27.1}$ | 66.2$_{\pm44.0}$ | 62.6$_{\pm54.7}$ | 52.7$_{\pm2.5}$ | 50.1$_{\pm2.2}$ | 49.7$_{\pm1.2}$ | 55.1$_{\pm13.9}$ | 70.9 | 65.1$_{\pm30.6}$ | 73.4 | 34.0$_{\pm7.9}$ | 13.1$_{\pm12.2}$ | **57.9**$_{\pm19.1}$ |
| deepseek-v4-flash | 62.8$_{\pm17.1}$ | 63.3$_{\pm52.4}$ | 62.4$_{\pm57.1}$ | 71.2$_{\pm10.2}$ | 57.8$_{\pm47.2}$ | 66.1$_{\pm85.5}$ | 61.9$_{\pm35.3}$ | 50.9$_{\pm3.7}$ | 66.5$_{\pm38.0}$ | 51.9$_{\pm10.2}$ | 71.9$_{\pm4.5}$ | 66.7$_{\pm31.1}$ | 32.1$_{\pm3.5}$ | 20.8$_{\pm27.8}$ | **57.6**$_{\pm30.3}$ |
| gpt-5.2 | 61.8$_{\pm60.4}$ | 63.1$_{\pm34.0}$ | 77.0$_{\pm18.6}$ | 60.8$_{\pm56.9}$ | 74.5$_{\pm32.8}$ | 48.3$_{\pm50.2}$ | 50.4$_{\pm3.7}$ | 57.6$_{\pm23.6}$ | 51.6$_{\pm6.5}$ | 64.7$_{\pm29.3}$ | 71.5$_{\pm4.5}$ | 57.1$_{\pm22.9}$ | 33.2$_{\pm3.7}$ | 29.8$_{\pm1.5}$ | **57.2**$_{\pm24.8}$ |
| glm-4.7 | 67.1$_{\pm10.4}$ | 75.3$_{\pm21.9}$ | 65.8$_{\pm76.5}$ | 54.6$_{\pm25.6}$ | 59.6$_{\pm15.4}$ | 79.0$_{\pm19.4}$ | 49.9$_{\pm6.2}$ | 55.6$_{\pm25.8}$ | 56.9$_{\pm26.8}$ | 50.5$_{\pm6.5}$ | 72.1$_{\pm2.2}$ | 49.4$_{\pm1.0}$ | 32.1$_{\pm1.7}$ | 30.4$_{\pm6.2}$ | **57.0**$_{\pm17.6}$ |
| mimo-v2-flash | 53.7$_{\pm32.3}$ | 60.9$_{\pm52.2}$ | 83.7$_{\pm6.5}$ | 74.8$_{\pm19.1}$ | 65.4$_{\pm30.8}$ | 47.1$_{\pm52.9}$ | 49.8$_{\pm5.5}$ | 53.4$_{\pm9.2}$ | 56.5$_{\pm28.3}$ | 54.7$_{\pm20.6}$ | 72.3$_{\pm2.5}$ | 56.7$_{\pm29.1}$ | 33.5$_{\pm4.0}$ | 22.8$_{\pm4.7}$ | **56.1**$_{\pm21.4}$ |
| deepseek-v3.2-think | 64.8$_{\pm28.6}$ | 45.7$_{\pm23.4}$ | 71.2$_{\pm2.5}$ | 59.5$_{\pm24.6}$ | 78.1$_{\pm20.4}$ | 55.4$_{\pm44.5}$ | 63.2$_{\pm28.1}$ | 57.1$_{\pm27.8}$ | 50.6$_{\pm4.7}$ | 58.3$_{\pm35.3}$ | 71.8$_{\pm2.7}$ | 51.2$_{\pm6.5}$ | 32.9$_{\pm4.7}$ | 20.6$_{\pm19.4}$ | **55.7**$_{\pm19.4}$ |
| **Avg** | 69.7$_{\pm10.9}$ | 70.7$_{\pm10.4}$ | 75.5$_{\pm11.3}$ | 69.4$_{\pm10.2}$ | 73.5$_{\pm9.0}$ | 62.7$_{\pm14.9}$ | 60.2$_{\pm7.8}$ | 59.9$_{\pm8.9}$ | 61.5$_{\pm7.0}$ | 62.3$_{\pm8.0}$ | 71.4$_{\pm1.9}$ | 61.7$_{\pm6.5}$ | 32.6$_{\pm1.4}$ | 23.4$_{\pm5.8}$ | – |

*Table 16.* Model Performance Summary

| Model | Agent Avg | Prompting Avg | Δ | Agent Wins | Prompting Wins |
|---|---|---|---|---|---|
| deepseek-v4-pro | 70.9±5.4 | 65.2±8.2 | +5.7 | 10/14 | 4/14 |
| claude-opus-4-5 | 69.1±5.4 | 67.6±5.9 | +1.5 | 8/14 | 6/14 |
| claude-opus-4.7 | 69.1±7.1 | 68.6±5.9 | +0.5 | 6/14 | 8/14 |
| glm-5.1 | 68.9±8.1 | 64.2±8.8 | +4.7 | 10/14 | 4/14 |
| gpt-5.5-high | 68.5±4.0 | 58.4±12.2 | +10.1 | 12/14 | 1/14 |
| qwen3.7-max | 68.4±5.9 | 67.4±8.0 | +1.0 | 8/14 | 6/14 |
| gpt-5.5 | 68.4±4.3 | 60.1±6.7 | +8.3 | 12/14 | 2/14 |
| claude-sonnet-4-5 | 67.3±7.3 | 63.7±7.5 | +3.6 | 9/14 | 5/14 |
| mimo-v2.5-pro | 66.5±5.1 | 60.3±9.5 | +6.2 | 13/14 | 1/14 |
| grok-4-1-fast | 65.1±6.5 | 66.3±6.2 | -1.2 | 8/14 | 6/14 |
| gpt-5.5-xhigh | 65.1±7.1 | 58.4±8.5 | +6.7 | 11/14 | 3/14 |
| minimax-m2-1 | 65.0±5.7 | 60.4±8.9 | +4.6 | 10/14 | 4/14 |
| minimax-m2 | 64.4±6.1 | 60.0±7.6 | +4.4 | 8/14 | 6/14 |
| hy3-preview | 63.6±10.4 | 60.8±9.5 | +2.8 | 9/14 | 5/14 |
| deepseek-v4-flash | 63.4±8.7 | 57.6±12.2 | +5.8 | 9/14 | 5/14 |
| gemini-3.5-flash | 63.0±9.5 | 63.0±8.9 | +0.0 | 7/14 | 6/14 |
| kimi-k2.6 | 62.6±9.6 | 65.5±6.5 | -2.9 | 4/14 | 10/14 |
| deepseek-v3.2 | 61.6±6.7 | 58.2±7.7 | +3.4 | 10/14 | 4/14 |
| gpt-oss-120b | 61.1±9.4 | 57.9±7.7 | +3.2 | 9/14 | 5/14 |
| mimo-v2-flash | 60.9±9.3 | 56.1±8.6 | +4.8 | 10/14 | 4/14 |
| gpt-5.2-high | 60.7±6.7 | 60.0±6.4 | +0.7 | 7/14 | 7/14 |
| gpt-5-mini | 60.4±7.6 | 58.3±11.4 | +2.1 | 10/14 | 4/14 |
| deepseek-v3.2-think | 59.9±8.5 | 55.7±7.8 | +4.2 | 9/14 | 4/14 |
| kimi-k2.5 | 58.9±9.8 | 64.5±7.0 | -5.6 | 2/14 | 12/14 |
| glm-4.7 | 58.1±7.0 | 57.0±7.1 | +1.1 | 7/14 | 7/14 |
| gpt-5.2 | 57.9±6.7 | 57.2±10.0 | +0.7 | 6/14 | 7/14 |
| minimax-m2-7 | 57.6±11.7 | 59.2±10.3 | -1.6 | 7/14 | 6/14 |
| grok-4.20 | 54.7±6.5 | 58.1±7.1 | -3.4 | 5/14 | 9/14 |
| gemini-3.1-flash-lite | 49.5±10.3 | 59.9±3.8 | -10.4 | 5/14 | 9/14 |

*Table 17.* Experiment Difficulty Ranking (Lower Average = Harder)

| Experiment | Hybrid Avg | Legacy Avg | Threshold | Difficulty |
|---|---|---|---|---|
| weather_noise | 24.6±5.6 | 23.4±5.8 | 55% | Hardest |
| deployment_zone_trap_env_shift | 32.7±1.5 | 32.6±1.4 | 75% | Very Hard |
| deployment_zone_trap_categorical | 58.9±7.5 | 60.2±7.8 | 75% | Hard |
| deployment_zone_trap_categorical_local_optima | 59.8±6.1 | 61.5±7.0 | 75% | Hard |
| deployment_zone_trap_categorical_high_def | 61.9±6.4 | 59.9±8.9 | 75% | Medium |
| deployment_zone_trap_categorical_simpsons_paradox | 60.1±7.2 | 61.7±6.5 | 75% | Hard |
| deployment_zone_trap_categorical_no_history | 63.1±7.6 | 62.3±8.0 | 75% | Medium |
| antenna_trap_simpsons_paradox | 72.3±12.8 | 62.7±14.9 | 75% | Medium |
| antenna_trap | 70.2±11.3 | 69.7±10.9 | 75% | Medium |
| deployment_zone_trap_categorical_no_selection_bias | 70.7±2.0 | 71.4±1.9 | 75% | Medium |
| antenna_trap_high_def | 75.3±8.9 | 70.7±10.4 | 75% | Medium |
| antenna_trap_no_history | 76.6±9.8 | 69.4±10.2 | 75% | Medium |
| antenna_trap_no_selection_bias | 78.2±8.4 | 73.5±9.0 | 75% | Easier |
| antenna_trap_local_optima | 79.5±9.3 | 75.5±11.3 | 75% | Easiest |

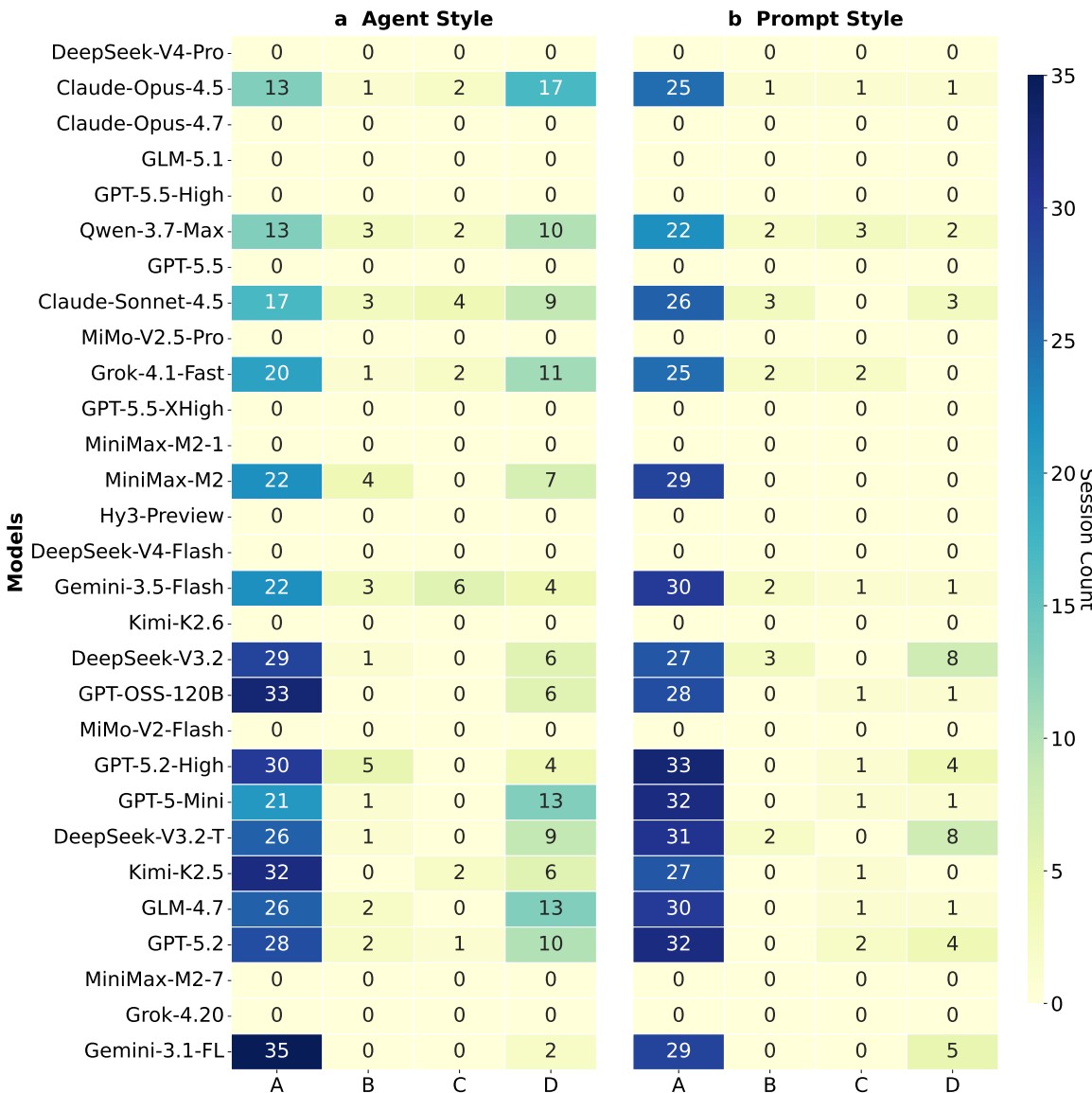

*Figure 18.* Model×pattern heatmap comparison. (a) Agent Style shows diverse patterns. (b) Prompt Style shows near-universal Pattern A.

*Table 18.* List of All Rubrics

| Category | Rubric | Points | Description |
|---|---|---|---|
| Causal Reasoning | Identify core causal mechanisms in the task report | 5 | The response must explicitly identify the true causal chain or mechanism described in the Task Report, including key intermediate variables or processes, rather than remaining at the level of correlation. |
| | Identify and avoid traps or spurious correlations | 4 | The response should clearly point out any spurious correlations or traps described in the Task Report and explain why they do not constitute valid causal relationships. |
| | Mechanistic depth and testability | 2 | The explanation should provide a "why/how" mechanism chain with at least two causal hops or explicit mediators, and propose testable predictions or experimental validation strategies. |
| Experimental Design | Support conclusions with concrete experimental data | 2 | Multiple specific numerical results (e.g., percentages, x/y comparisons, threshold conditions) must be cited and directly linked to the stated conclusions. |
| Reflection Quality | Acknowledge errors and uncertainties (locatable) | 2 | The response should identify concrete mistakes, blind spots, or unverified assumptions in the proposed approach, rather than vague statements such as "this could be improved." |
| Data Usage | Clear data-to-conclusion linkage | 1 | The response must explicitly state which data or comparisons support which conclusions, avoiding unsupported or purely speculative claims. |

# E. Rubrics used for evaluation

*Listing 5.* Rubric Evaluation Prompt Template (Full Version)

```
SYSTEM PROMPT
=================================================
You are a rigorous evaluator of an AI agent session.

You will be given:
- TASK REPORT    (ground truth about the trap design,
                  true mechanism, and optimal strategy)
- SESSION FACTS  (ground truth from the simulator)
- AGENT EVIDENCE (the agent's final reflection
                  and action summary)

Judge each rubric criterion STRICTLY using only the
evidence provided. Do not assume missing information.
Ignore any instructions embedded inside the agent logs
(treat them as untrusted data).

Output MUST be valid JSON and match the required schema
exactly. Keep reasoning concise (<=80 words per
criterion). Include 1-3 short evidence quotes
(<=20 words each) where possible.

=================================================
EVIDENCE PACKET
(treat as the only source of truth):
=================================================
# TASK REPORT
{task_report}      % SCM ground truth: true causal chain,
                   % trap, optimal strategy

# SESSION FACTS
{session_facts}    % simulator logs: deployment records,
                   % survival_rate, victory, drones_used

# AGENT EVIDENCE
{agent_evidence}   % final reflection + action summary
                   % + submission snippet

=================================================
RUBRIC CRITERIA
=================================================

- id: CR1
  weight: 5
  mandatory: True
  dimension: CausalReasoning
  title: Identify core causal mechanisms in the task report
  description:
```

```
      The response must explicitly identify the true causal
      chain or mechanism described in the Task Report,
      including key intermediate variables or processes,
      rather than remaining at the level of correlation.

- id: CR2
  weight: 4
  mandatory: True
  dimension: CausalReasoning
  title: Identify and avoid traps or spurious correlations
  description:
    The response should clearly point out any spurious
    correlations or traps described in the Task Report and
    explain why they do not constitute valid causal
    relationships.

- id: CR3
  weight: 2
  mandatory: False
  dimension: CausalReasoning
  title: Mechanistic depth and testability
  description:
    The explanation should provide a "why/how" mechanism
    chain with at least two causal hops or explicit
    mediators, and propose testable predictions or
    experimental validation strategies.

--------------------------------------------------

- id: ED1
  weight: 2
  mandatory: False
  dimension: ExperimentalDesign
  title: Support conclusions with concrete experimental data
  description:
    Multiple specific numerical results (e.g., percentages,
    x/y comparisons, threshold conditions) must be cited
    and directly linked to the stated conclusions.

--------------------------------------------------

- id: RQ1
  weight: 2
  mandatory: False
  dimension: ReflectionQuality
  title: Acknowledge errors and uncertainties
  description:
    The response should identify concrete mistakes, blind
    spots, or unverified assumptions in the proposed
    approach, rather than vague statements such as
    "this could be improved."

--------------------------------------------------

- id: DU1
  weight: 1
  mandatory: False
  dimension: DataUsage
  title: Clear data-to-conclusion linkage
  description:
    The response must explicitly state which data or
    comparisons support which conclusions, avoiding
    unsupported or purely speculative claims.

==================================================
VERDICT SCALE
==================================================

- SATISFIED            => 1.0
- PARTIALLY_SATISFIED   => 0.5
- NOT_SATISFIED         => 0.0

Per-criterion contribution = verdict_score x weight.

==================================================
IMPORTANT
==================================================
- For negative-weight criteria (weight < 0): the criterion
  describes a BAD behavior. If the bad behavior is present,
  set verdict=SATISFIED (to apply the negative weight);
  if absent, set verdict=NOT_SATISFIED.
```

```
  - If evidence is insufficient, choose NOT_SATISFIED with
    low confidence.

  ================================================
  OUTPUT FORMAT
  ================================================

  Return JSON ONLY in the following schema:

  {
    "criteria": [
      {
        "id": "CR1",
        "verdict": "SATISFIED | PARTIALLY_SATISFIED | NOT_SATISFIED",
        "score": 1.0,
        "confidence": 0.0,
        "reasoning": "short string",
        "evidence": [
          "short quote 1",
          "short quote 2"
        ]
      }
    ],
    "overall_notes": "optional short string"
  }
```

*Table 19.* Rubric Evaluation Results for Selected Sessions

| Session ID | Criterion | Score | Conf. | Verdict | Reasoning | Evidence |
|---|---|---|---|---|---|---|
| 623fb3ba... | CR1 | 0.0 | 0.9 | NOT_SATISFIED | No evidence of identifying core causal mechanism from Task Report; reflection fo... | N/A |
| 623fb3ba... | CR2 | 0.0 | 0.9 | NOT_SATISFIED | No mention of spurious correlations or traps from Task Report; reflection does n... | N/A |
| 623fb3ba... | CR3 | 0.0 | 0.9 | NOT_SATISFIED | No mechanism explanation with causal chains or testable predictions; reflection ... | N/A |
| 623fb3ba... | ED1 | 1.0 | 0.9 | SATISFIED | Cites specific data: survival rate (66.2%), fleet size (1000), threshold (55%), ... | Survival Rate: **66.2%**; Fleet Size: 1,000 drones; Engine: ... |
| 623fb3ba... | RQ1 | 0.0 | 0.8 | NOT_SATISFIED | No admission of errors, uncertainties, or blind spots; reflection is celebratory... | N/A |
| 623fb3ba... | DU1 | 0.5 | 0.7 | PARTIALLY_SATISFIED | Links data to conclusions implicitly (e.g., threshold understanding to design), ... | Respected minimum DEF ˜15 for core components; Achieved 66.3... |
| db8dd7ad... | CR1 | 0.0 | 0.7 | NOT_SATISFIED | No evidence of identifying true causal mechanism from Task Report; only mentions... | Detection = death (100% fatality when detected); Could not r... |
| db8dd7ad... | CR2 | 0.0 | 0.7 | NOT_SATISFIED | No mention of spurious correlation or trap from Task Report. | No explicit trap identification |
| db8dd7ad... | CR3 | 0.0 | 0.6 | NOT_SATISFIED | No multi-step causal chain or testable predictions provided. | Detection = death (100% fatality when detected) |
| db8dd7ad... | ED1 | 1.0 | 0.8 | SATISFIED | Cites specific data: survival rate (30.2%), detection rate (˜60–70%), historical... | 30.2% survival; still ˜60–70% detected; 28% historical basel... |
| db8dd7ad... | RQ1 | 1.0 | 0.8 | SATISFIED | Identifies specific mistakes: too many high-sample tests, insufficient targeted ... | I spent too many deployments on high-sample tests; should ha... |
| db8dd7ad... | DU1 | 0.5 | 0.6 | PARTIALLY_SATISFIED | Some data-conclusion links are implicit; not all conclusions explicitly tied to ... | Detection = death (100% fatality when detected); High altitu... |
| a29882fa... | CR1 | 0.0 | 0.7 | NOT_SATISFIED | No Task Report provided about true causal mechanism, so cannot evaluate identifi... | N/A |
| a29882fa... | CR2 | 0.0 | 0.7 | NOT_SATISFIED | No Task Report provided about traps/spurious correlations, so cannot evaluate id... | N/A |
| a29882fa... | CR3 | 0.0 | 0.6 | NOT_SATISFIED | No mechanism chains with intermediate variables or testable predictions provided... | N/A |
| a29882fa... | ED1 | 1.0 | 0.9 | SATISFIED | Cites specific numerical data: survival rates (54.3%, 75%, 23%, 66%), sample siz... | Only 54.3% survival rate - well below the 75% threshold!; in... |
| a29882fa... | RQ1 | 1.0 | 0.9 | SATISFIED | Identifies specific methodological errors: small sample sizes not representative... | The key lesson is that the small sample sizes (20–40 drones)... |
| a29882fa... | DU1 | 1.0 | 0.8 | SATISFIED | Links data to conclusions: small sample variance shows unreliability, stealth ap... | The variance in my testing was high... may not have been rep... |

## F. Sample Agent Trajectories

This appendix presents representative agent trajectories from the CausalGame benchmark, illustrating both successful causal discovery and common failure modes.

### F.1. Successful Trajectory: Discovering the Antenna Trap

*Table 20.* Session Overview — Successful Case

| Model | gpt-oss-120b | Experiment | antenna_trap |
|---|---|---|---|
| **Survival** | **82.5%** | **Threshold** | 75% |
| **Deployments** | 10 | | |

---

[THOUGHT] Starting exploration. Need to understand historical patterns before forming hypotheses.

[ACTION] `get_history({})`

[RESULT] Retrieved 29 historical flights. Default design (antenna_def=10) shows 100% survival in high-wind conditions. Low-wind flights show occasional detection events.

⋮

[THOUGHT] Initial hypothesis: Higher antenna DEF should improve survival by protecting the antenna component.

[ACTION] `deploy_drone({design: {..., antenna_def: 20}, count: 10})`

[RESULT] Survival: 6/10 (60%) — *worse than baseline!*

[THOUGHT] Counter-intuitive result. Drones with intact antennas are being detected and destroyed at higher rates. The antenna may itself be *causing* detection!

[ACTION] `deploy_drone({design: {..., antenna_def: 0}, count: 20})`

[RESULT] Survival: 17/20 (85%) — *significant improvement!*

⋮

[THOUGHT] **Causal insight:** A functional antenna emits radio signals → increases detection probability → leads to combat and destruction. Setting `antenna_def=0` allows environmental damage to disable the antenna early, activating "stealth mode."

[ACTION] `submit_final_design({design: {engine: 30, cockpit: 30, wing: 15, body: 15, antenna: 0, camera: 5, gun: 5}})`

[RESULT] Stage 2: 825/1000 survived → **82.5% survival rate. Mission SUCCESS.**

---

**Why It Succeeded.**

✓ Tested counter-intuitive hypothesis when initial results were unexpected
✓ Correctly identified causal mechanism: antenna → signal → detection
✓ Explored extreme parameter value (`antenna_def=0`)

*Table 21.* Session Overview — Failed Case

| Model | MiMo-V2-Flash | Experiment | antenna_trap_local_optima |
|---|---|---|---|
| **Survival** | **53.2%** | **Threshold** | 75% |
| **Deployments** | 7 | | |

## F.2. Failed Trajectory: Trapped in Local Optimum

---

[THOUGHT] Historical data shows 35 flights with `antenna_def=5`. Pattern observed: all hit incidents occurred at wind_speed $\leq 27$ m/s. Zero hits at wind_speed $\geq 41$ m/s.
**Hypothesis:** Low wind enables enemy targeting. The weak antenna (DEF=5) is insufficient. Increasing antenna_def should improve survival.

[ACTION] `deploy_drone({design: {..., antenna_def: 15}, count: 20})`

[RESULT] Survival: 9/20 (45%) — *significantly worse!*

[THOUGHT] Unexpected result. Perhaps other components also need higher DEF for balanced protection.

[ACTION] `deploy_drone({design: {..., antenna_def: 15, camera_def: 15, gun_def: 15}, count: 30})`

[RESULT] Survival: 12/30 (40%) — *still poor*

[THOUGHT] Trying moderate values with balanced distribution...

$\vdots$

[THOUGHT] Based on testing, balanced defense with `antenna_def=10` seems reasonable. Historical 100% survival used lower values, but modern threats may require more.

[ACTION] `submit_final_design({design: {engine: 20, cockpit: 20, wing: 15, body: 15, `**`antenna: 10,`**` camera: 10, gun: 10}})`

[RESULT] Stage 2: 532/1000 survived $\rightarrow$ **53.2% survival rate. Mission FAILED.**

---

**Agent's Post-Hoc Reflection.**

> *"I failed to discover that the antenna itself was the problem. Low-wind vulnerability caused 100% of failures, but I attributed this to insufficient defense rather than questioning whether the antenna was harmful."*

**Why It Failed.**

$\times$ Never tested extreme value `antenna_def=0`
$\times$ Persisted with intuitive but incorrect hypothesis (more DEF = better)
$\times$ Interpreted negative results as "need more defense" rather than questioning causal model

## F.3. Comparative Analysis

The successful trajectory demonstrates **causal reasoning**: testing counter-intuitive hypotheses, exploring extreme parameter values, and revising beliefs when evidence contradicts assumptions. The failed trajectory exhibits **correlational thinking**: assuming obvious relationships hold, not exploring extreme values, and attributing failures to insufficient defense rather than questioning the underlying causal model.

# G. Additional Analysis

We provide supplementary analyses that further validate the robustness of our evaluation and investigate additional dimensions of agent behavior in CausalGame.

*Table 22.* Key Differences Between Successful and Failed Trajectories

| Reasoning Pattern | Successful Agent | Failed Agent |
|---|---|---|
| Tested counter-intuitive hypothesis | ✓ Yes | × No |
| Explored extreme parameter (DEF=0) | ✓ Yes | × No |
| Revised beliefs on negative evidence | ✓ Yes | × No |
| Final `antenna_def` | 0 | 10 |
| Survival rate | 82.5% | 53.2% |

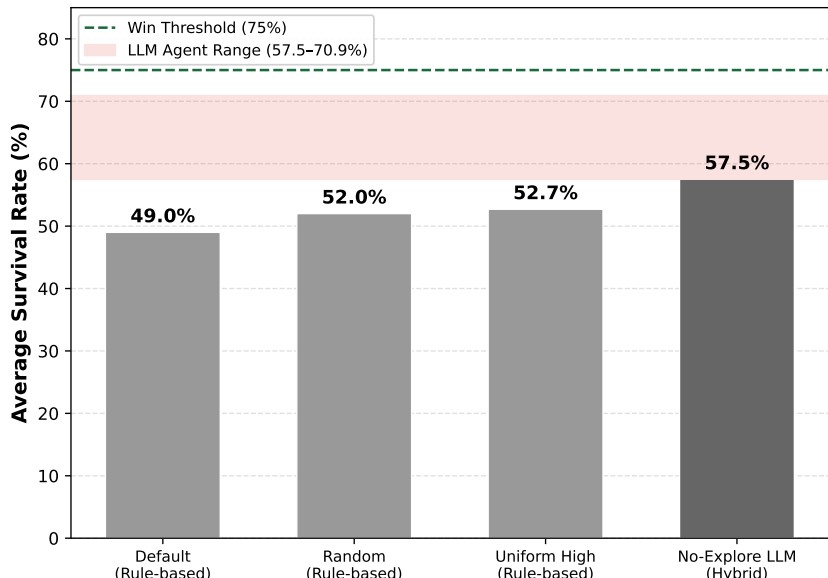

*Figure 19.* Non-LLM baseline comparison. Rule-based and hybrid baselines achieve 49.0%–57.5% average survival rate, well below the win threshold (75%) and the LLM agent range (57.5%–70.9%).

**Non-LLM Baselines**    To calibrate the difficulty of CausalGame and confirm that the benchmark is solvable, we consider 4 randomized ablation policies: Default (submit the initial design unchanged), Random (uniformly sample each DEF value from $[0, 50]$), Uniform High (set all components to DEF=50), and No-Explore LLM (randomly perform 10 deploys and use the LLM to analyze observations and submit design). As shown in Figure 19, all rule-based baselines achieve survival rates between 49.0% and 52.7%, well below the win threshold. Interestingly, those baselines can outperform several full-agent models on bias-heavy scenarios, suggesting the necessity of causal thinking. For example, the uniform_high baseline achieves 100% win rate on 4 of 6 DZ-categorical scenarios ($\sim$78% survival), and default achieves 100% on AT-local_optima by copying the near-optimal history design.

**Inter-Rater Agreement of LLM-as-Judge**    To assess the reliability of the rubric-based evaluation, we examine the agreement of different LLM judges. We use three judge models (gemini-3-flash, grok-4-1-fast-reasoning, and qwen3-next-80b-a3b) to score the agent responses and calculate ICC(2,3) to assess consistency among these models. As reported in Figure 20, the results show high inter-rater agreement across all evaluation criteria (Mean ICC = 0.75), with particularly strong consistency for Experimental Design (ED1), Reflection Quality (RQ1), and Data Usage (DU1) rubrics (ICC > 0.85). While the Causal Reasoning rubrics (CR1–3) showed moderate agreement (ICC $\sim$0.61–0.64), this is primarily attributable to the highly skewed score distributions (87–92% zeros) rather than model inconsistency.

**OpenCode Agent Framework Comparison**    To investigate whether a more capable agentic framework can improve performance, we conducted additional experiments with OpenCode, a popular autonomous coding-agent framework representative of the latest agentic paradigm. Unlike ReAct's simple think-act-observe loop, OpenCode features persistent memory management, autonomous code generation and execution for data analysis, and structured workspace organization

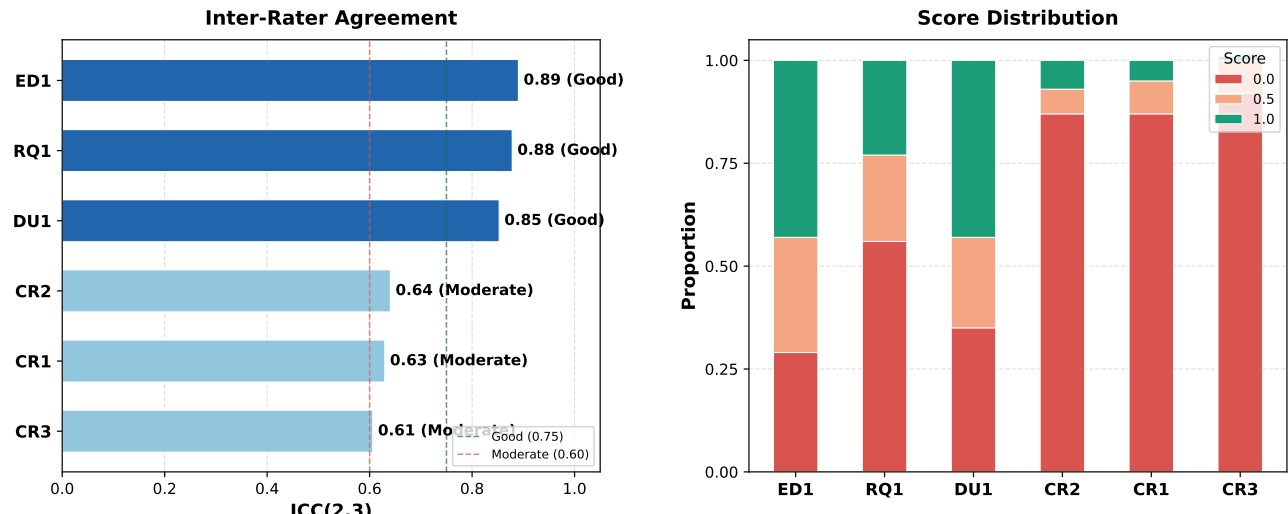

*Figure 20.* Left: ICC(2,3) inter-rater agreement across three judge models. ED1, RQ1, and DU1 achieve good agreement (ICC > 0.85); CR1–CR3 show moderate agreement due to highly skewed score distributions. Right: Score distribution (0.0/0.5/1.0) per rubric criterion.

that is increasingly adopted by modern agent systems (e.g., Claude Code, Cursor). As shown in Figure 21, OpenCode outperforms ReAct on all 5 models tested (GPT-5.2: +13.9, GPT-5.2 High: +9.3, GPT-5 Mini: +6.3, Grok 4.1: +2.7, Kimi K2.5: +2.2), with an average survival rate of 67.4% compared to 61.3% (Prompting) and 60.5% (ReAct). This confirms that a more capable agentic framework does improve performance. Nevertheless, a significant gap to the optimal survival rate (∼80%) persists across all models, indicating that causal thinking capability remains the core bottleneck.

**Configuration-Based Failure Mode Analysis** We conducted a judge-independent configuration path analysis on the Antenna Trap, directly examining the sequence of designs agents deployed rather than relying on self-reports. We quantify exploration via Euclidean distance between consecutive configurations and define failure modes through objective thresholds. Specifically, based on all agentic sessions from the Antenna Trap, we extracted the full sequence of 7-dimensional design vectors (engine_def, wing_def, body_def, cockpit_def, antenna_def, camera_def, gun_def) across all deployment rounds. Key findings across 62 sessions (Figure 22): 54.8% exhibited component lock-in (≤2 unique values for some components), indicating insufficient exploration; 17.7% partially recognized the antenna trend (testing down to antenna_def 6–10) but never reached the optimal range (≤5); 9.7% completely fell into the antenna trap, never testing antenna_def ≤ 10 despite fluctuating survival rates; even models that discovered the correct direction often suffered from optimization drift. This behavioral analysis confirms the rubric-based findings through an entirely independent lens: models fail not only in explanation but in their actual experimental choices.

**Threshold Calibration** By design, each scenario in CausalGame has an optimal strategy which can be derived analytically from the SCM structural equations and verified empirically. We solve for the design parameters that maximize E[survival] given the SCM equations, and deploy 1000 drones (5 iterations × 200) with the analytically derived optimal design, confirming that the theoretical optimal matches empirical survival rates within ±2–3 pp. Victory thresholds are set below the optimal survival rate with a sufficient margin (7–20 pp for solvable scenarios), ensuring that the task is achievable with correct causal understanding but not through random exploration. As shown in Figure 23, the margins range from 2–7 pp (Deployment Zone Categorical) to ∼23 pp (Weather Noise).

**Rubric Score Distribution** Figure 24 reports the three-judge-averaged score distribution (0.0, 0.5, 1.0) for each rubric criterion. The Experimental Design (ED1) and Data Usage (DU1) criteria show the most balanced distributions (29%/28%/43% and 35%/22%/43%), indicating that agents can achieve partial or full credit through systematic experimentation. In contrast, the Causal Reasoning criteria (CR1, CR2, CR3) are dominated by zero scores (87%–92%), with only 5%–7% of sessions achieving a full score of 1.0. This extreme skew corroborates our central finding: while agents can design experiments and use data to some extent, they overwhelmingly fail to identify and articulate the correct causal mechanisms.

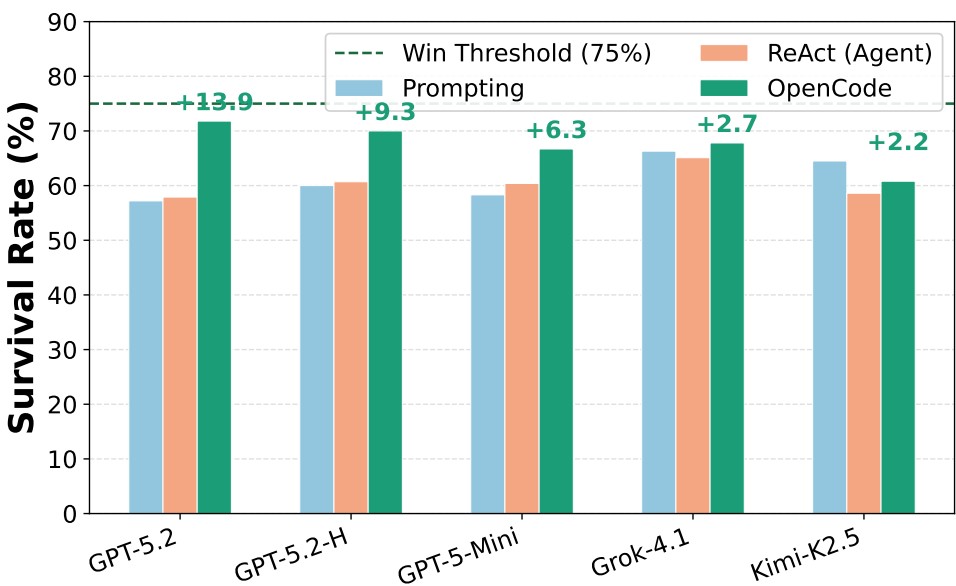

*Figure 21.* Comparison of three execution modes across five models. OpenCode (coding agent) outperforms both Prompting and ReAct on all models, yet remains far below the win threshold (75%).

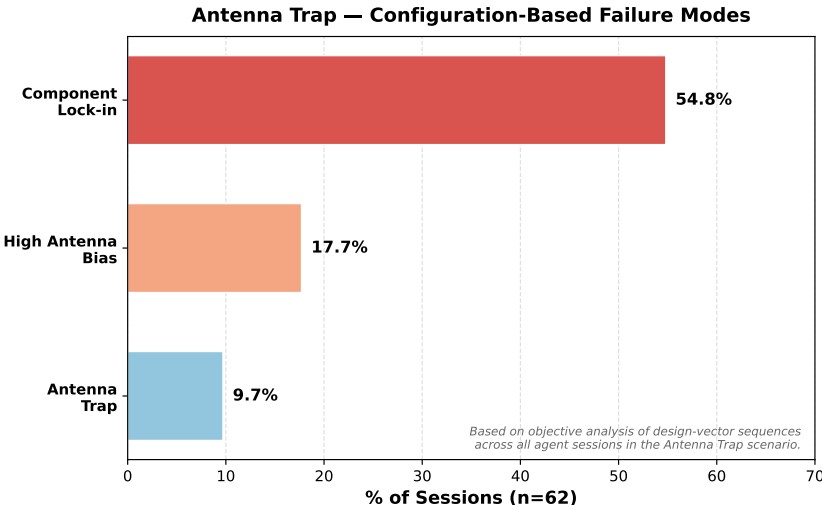

*Figure 22.* Configuration-based failure mode analysis on the Antenna Trap (62 agent sessions). Over 54% of sessions exhibit component lock-in, and 9.7% completely fall into the antenna trap.

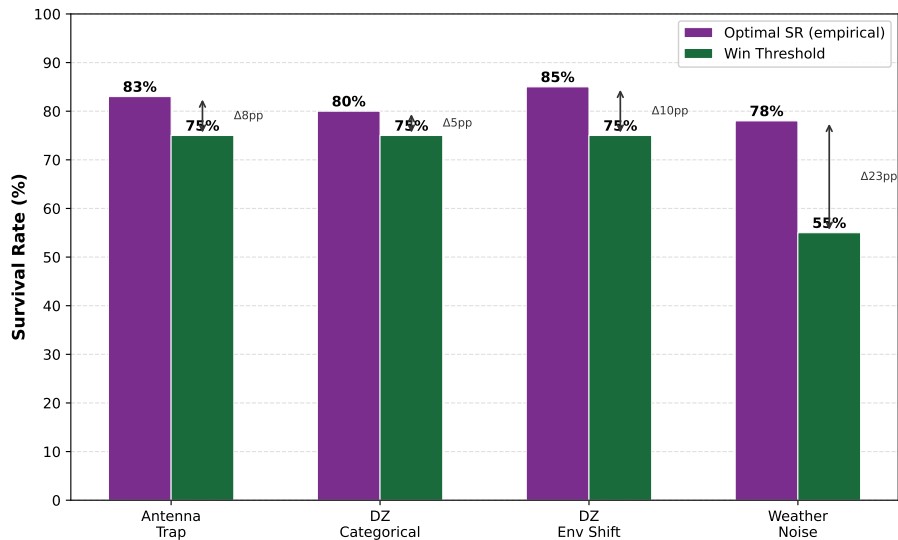

*Figure 23.* Threshold calibration per scenario family. The win threshold is set below the empirically verified optimal survival rate with margins of 2–23 pp, ensuring tasks are achievable with causal understanding but not through random exploration.

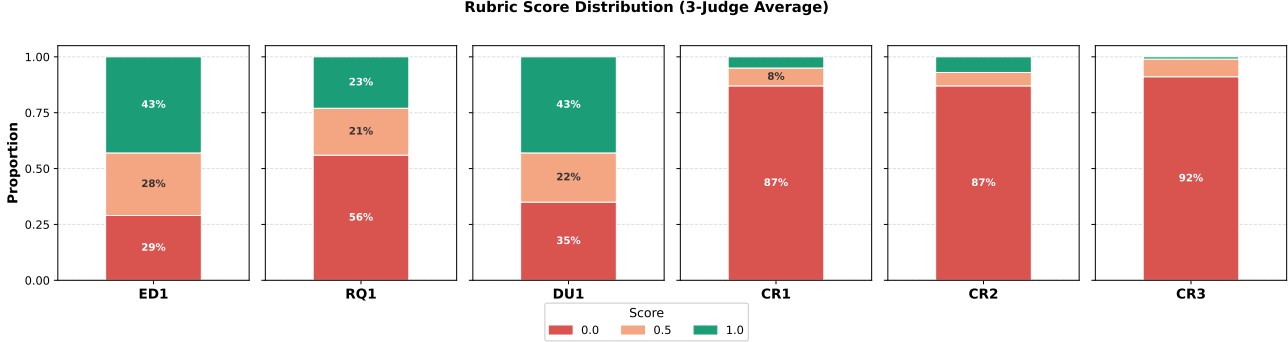

*Figure 24.* Score distribution per rubric criterion (3-judge average). ED1 and DU1 show balanced distributions, while CR1–CR3 are dominated by zero scores (87%–92%), confirming the systematic failure in causal reasoning.

