# OpenReview forum: "CausalGame: Benchmarking Causal Thinking of LLM Agents in Games"
_ICML.cc/2026/Conference — ICML 2026 spotlight_

### Official Review · Reviewer_fihW · 2026-03-08

**Soundness:** 3
**Presentation:** 3
**Significance:** 3
**Originality:** 3
**Overall Recommendation:** 5
**Confidence:** 4

**Summary:**

This paper provides an analysis of causal reasoning in large language models using a set of games created by the authors. Models generally perform poorly in these games, falling significantly short of optimal performance. Analysis of failure models identifies a set of patterns the models demonstrate across these games.

**Compliance With Llm Reviewing Policy:**

Affirmed.

**Final Justification:**

The rebuttal addressed my main concerns and I moved from "weak accept" to "accept".

**Key Questions For Authors:**

1. I didn't understand how the pictures in Figure 1 relate to the text -- the outcome in the green panel doesn't look great to me either (not what I want my doctor to do necessarily!).

2. How was the optimal answer identified for scoring the models? More information on what the correct solutions are and how they are computed would be helpful for readers to understand the paper.

3. What do the error bars show in Figures 4, 5, and 8?

**Limitations:**

There is no discussion of limitations in the paper. Societal impacts are considered in the impact statement.

**Strengths And Weaknesses:**

The main strengths of this paper are that it provides a more rigorous analysis of causal reasoning in these models than previous work, grounding its approach in causal Bayes nets and translating this into simple game environments, and that it provides an analysis across a wide range of models as well as some error analysis.

Weaknesses:

1. The choice of games seems relatively arbitrary. There is already an extensive literature in cognitive science evaluating human causal reasoning that seems like it could have been drawn upon for a less arbitrary set of choices. See for example:

https://web.mit.edu/cocosci/archive/Papers/steyvers-etal-2003.pdf (classic paper)
https://www.bramleylab.ppls.ed.ac.uk/pdfs/bramley2017neurath.pdf (more recent work)

There's also recent work using approaches inspired by cognitive science to look at scientific reasoning in LLMs, e.g.:

https://openreview.net/forum?id=544P6YidFk
https://arxiv.org/abs/2505.17968

2. The empirical observations in section 4.2 should be supported by statistical significance tests.

3. It's not clear exactly how the failure modes were identified. It seems like this was by the authors looking for patterns. If so it should be explained how these patterns were found, and how the results were coded (ideally this should be done by multiple coders blind to the models etc.), with agreement reported.

---

> ### Author Rebuttal · Authors · 2026-03-31
>
> Thank you for your time and insightful comments in reviewing our work! Our explanations to your questions are as follows:
>
> > W1 The choice of games seems
>
> Thank you for providing the references, which we included in our revised manuscript. The design of CausalGame is mainly focused on **instantiating the causal challenges of historical scientific discovery scenarios in interactive agentic environments**, such that we could evaluate whether LLMs can tackle those challenges as AI Scientist agents. We believe it’d bring more insights and extend the scope of CausalGame by evaluating the cognitive biases of LLMs in causal reasoning, which we consider a promising future direction!
>
> > W2 statistical significance tests
>
> Following your suggestion, we rerun the evaluation with 3 trials and report the variance. All results are based on 3 independent trials per model × scenario combination. We report mean survival rates with standard deviations and 95% confidence intervals.
>
>
> **Aggregate by model (14 scenarios × 3 trials per mode):**
>
> | Model | Prompting | Std | 95% CI | Agentic | Std | 95% CI |
> |-------|:---------:|:---:|:------:|:-------:|:---:|:------:|
> | Claude Opus 4.5 | 67.6% | 5.9 | ±6.7 | 69.1% | 5.4 | ±6.1 |
> | Claude Sonnet 4.5 | 63.7% | 7.5 | ±8.5 | 67.3% | 7.3 | ±8.3 |
> | Grok 4.1 Fast | 66.3% | 6.2 | ±7.0 | 65.1% | 6.5 | ±7.4 |
> | MiniMax M2-1 | 60.4% | 8.9 | ±10.1 | 65.0% | 5.7 | ±6.5 |
> | MiniMax M2 | 60.0% | 7.6 | ±8.6 | 64.4% | 6.1 | ±6.9 |
> | DeepSeek V3.2 | 57.5% | 8.5 | ±9.6 | 61.7% | 6.7 | ±7.6 |
> | GPT-OSS-120B | 56.6% | 10.3 | ±11.7 | 61.1% | 9.4 | ±10.6 |
> | GPT-5.2 High | 60.0% | 6.4 | ±7.2 | 60.7% | 6.7 | ±7.6 |
> | GPT-5 Mini | 58.3% | 11.4 | ±12.9 | 60.4% | 7.6 | ±8.6 |
> | DeepSeek V3.2 Thinking | 55.7% | 7.9 | ±8.9 | 59.9% | 8.5 | ±9.6 |
> | Kimi K2.5 | 64.5% | 7.0 | ±7.9 | 58.6% | 9.8 | ±11.1 |
> | GLM-4.7 | 57.7% | 8.1 | ±9.2 | 58.1% | 7.0 | ±7.9 |
> | GPT-5.2 | 57.2% | 10.0 | ±11.3 | 57.9% | 6.7 | ±7.6 |
>
> (Std = mean of per-scenario standard deviations across 3 trials; 95% CI = 1.96 × Std / √3; sorted by Agentic SR.  Full per-trial data: [Link](https://anonymous.4open.science/r/CausalGame-Rebuttal-7CB5/))
>
> The multi-trial results are consistent with our original findings: **all models remain significantly below the win threshold, and the ranking among models is largely preserved**. Qwen3-235B was excluded because it frequently hallucinated successful API calls (fabricating tool responses that never occurred), causing most sessions to timeout and preventing complete trials. Gemini-3-Flash/Pro were excluded as their endpoints are expired now.
>
>
> > W3 More analysis of failure modes.
>
> Please refer to our response to Reviewer ZoF6.
>
> > Q1 Interpretation of Figure 1.
>
> We have revised the outcome from quantitative to qualitative in Figure 1 to make the story more explicit. The intent of Figure 1 is to illustrate that a naive agent following observed correlations (which are distorted by a hidden confounder) arrives at a suboptimal treatment, while a causal agent actively experiments to identify the true mechanism and achieves a substantially better outcome. The specific numbers (35% vs 78%) are illustrative rather than meant to represent realistic clinical outcomes, as updated in this [link](https://anonymous.4open.science/r/CausalGame-Rebuttal-7CB5/updated_Fig1.pdf).
>
> > Q2 Optimal answer
>
> By design, each scenario in CausalGame has an optimal strategy which can be **derived analytically from the SCM structural equations** and **verified empirically**:
> 1. **Analytical derivation**: We solve for the design parameters that maximize E[survival] given the SCM equations. For example, in the Antenna Trap, setting antenna_def=0 ensures antenna destruction → stealth mode, which analytically minimizes detection probability.
> 2. **Empirical verification**: We deploy 1000 drones (5 iterations × 200) with the analytically derived optimal design using the admin test-deploy endpoint, confirming that the theoretical optimal matches empirical survival rates within ±2-3 pp.
> 3. **Threshold calibration**: Victory thresholds are set below the optimal survival rate with a sufficient margin (7-20 pp for solvable scenarios), ensuring that the task is achievable with correct causal understanding but not through random exploration.
>
> | Family | Optimal SR (empirical) | Threshold | Margin |
> |--------|:---------------------:|:---------:|:------:|
> | Antenna Trap | 82–84% | 75% | 7–9 pp |
> | Deployment Zone  Categorical | 77–82% | 75% | 2–7 pp |
> | Deployment Zone Env Shift | ~85% | 75% | ~10 pp |
> | Weather Noise | ~78% | 55% | ~23 pp |
>
>
> > Q3 Error bars in Figures 4, 5, and 8.
>
> The error bars in Figure 4-6 are calculated according to the LLMs’ performances across different settings in CausalGame under each category.
>
> > Limitations
>
> We have included more limitation discussions regarding the setting and evaluation following your and Reviewer R6d8’s suggestions.

---

> > ### Author Rebuttal · Reviewer_fihW · 2026-04-01
> >
> > Thank you these responses largely address my concerns. I will adjust my score accordingly.

---

> > > ### Author Response · Authors · 2026-04-03
> > >
> > > Dear Reviewer fihW,
> > >
> > > Thank you so much for your prompt engagement and for agreeing to increase the score. We are sincerely grateful for your insightful comments that helped strengthen our work!
> > >
> > > Best regards,
> > > Authors

---

### Official Review · Reviewer_R6d8 · 2026-03-12

**Soundness:** 3
**Presentation:** 3
**Significance:** 3
**Originality:** 3
**Overall Recommendation:** 4
**Confidence:** 3

**Summary:**

This paper introduces CausalGame, an interactive causal thinking benchmark for LLM-based AI Scientist agents. Rather than using static causal QA, the paper formulates the task as a multi-turn game environment driven by SCM, where models must actively design experiments, collect observations, analyze results, and ultimately submit a design proposal together with a brief explanation under a constrained budget. The benchmark contains 14 game scenarios and explicitly incorporates selection bias, noisy measurements, and hidden confounders. Performance is evaluated using both survival rate and rubric-based explanation evaluation. The authors further compare a range of frontier LLMs under both prompting and agentic execution modes, and use the results to discuss the current limitations of these models in recovering the underlying causal mechanism.

**Compliance With Llm Reviewing Policy:**

Affirmed.

**Final Justification:**

The rebuttal has basically addressed my main concerns, and I am therefore changing my recommendation from weak reject to weak accept.

**Key Questions For Authors:**

- Could you provide more detail on the rubric judge setup and its reliability validation, including the judge model, decoding setting, whether a single judge is used, and whether you conducted second-judge or human agreement analysis? This would directly affect my assessment of the paper’s soundness.
- Could you include a more tightly controlled Prompting versus Agentic comparison, so that the two modes are as comparable as possible in terms of tool access, action budget, whether ReAct is mandatory, and whether an exploration guard is used? If the conclusion still holds under a more controlled comparison, I would find Observation 3 more convincing.
- For Figure 4, Figure 5, Table 4, and several key observations in the main text, could you provide repeated runs, confidence intervals, or significance analysis? This would affect my confidence in the statistical robustness of the current empirical conclusions.
- The main text refers to four rubric dimensions, but the appendix additionally includes items such as D5_ActionabilityAndLearning and TASK.M1, and the failure analysis uses an overall rubric score out of 22. Please clarify more explicitly which items are included in the final score, how they are aggregated, and what role the more fine-grained appendix rubric plays in the main results.
- In the appendix, query_environment is described as a discovery tool, but the failure trajectories also show that this tool is unavailable in some variants. Please clarify which scenario variants provide this tool, which do not, and whether the main results mix these two settings when comparing across scenarios.

**Limitations:**

No.

The paper currently includes only a brief Impact Statement, and I did not see a sufficiently developed discussion of limitations. At a minimum, I would recommend discussing the gap between the benchmark as a synthetic simulator and real scientific discovery, the protocol asymmetry between Prompting and Agentic, and the potential bias introduced by the LLM-based judge.

**Strengths And Weaknesses:**

### **Soundness**

**Strengths**

The task setup is well-motivated and substantially closer than static causal QA to the process of scientific discovery. The benchmark uses SCMs as the environment engine and explicitly builds selection bias, measurement error, and hidden confounders into the scenario design, making the overall framework coherent.

**Weaknesses**

- My main concern is that many of the later analyses rely heavily on the rubric judge, but the main text only states that an LLM-based judge is used, without providing further details about the judge model, judge setting, stability, or agreement with human evaluation. As a result, the failure mode taxonomy and the interpretation that the key attribution of failure is a lack of causal thinking capability are not yet sufficiently well supported.
- The appendix already makes clear that Prompting and Agentic are not symmetric in terms of tool permissions, mandatory ReAct, exploration guard, and tool-call limits. As a result, the paper’s conclusion that the agentic framework does not necessarily bring improvements is still confounded by protocol design.
- The main results are primarily presented through aggregate survival rate and mean-based comparisons. I did not see repeated runs, confidence intervals, or significance analysis, so the statistical robustness of several empirical observations remains under-supported.

### **Presentation**

**Strengths**

The paper is generally well organized and easy to follow. It first motivates why AI Scientist agents require causal thinking, then distinguishes CausalGame from existing benchmarks in Table 1, and subsequently introduces the benchmark design, the main experimental findings, and the failure analysis. The appendix also provides useful supplementary material, including execution modes, scenario families, prompts, and trajectory examples.

**Weaknesses**

- One issue that affects readability is that the asymmetry between Prompting and Agentic is not stated early enough or clearly enough in the main text, even though this directly bears on several of the core observations. The relevant details are mainly deferred to the appendix.

- Some of the paper’s claims are stated somewhat too strongly, including provides a rigorous measurement, none of the existing benchmarks examine the causal thinking capability, and the attribution of failure to a lack of causal thinking capability. These formulations go beyond what the current evidence can fully support.

- The final rubric scoring scheme also remains somewhat unclear. The main text describes four rubric dimensions, but the later failure analysis uses an overall rubric score with a maximum of 22 points, while the appendix additionally includes items such as D5_ActionabilityAndLearning and TASK.M1.

### **Significance**

**Strengths**

I think the paper addresses an important problem. Many existing AI Scientist benchmarks emphasize workflow execution, coding, analysis, or scientific QA, whereas this benchmark isolates active mechanism identification under selection bias, hidden confounding, and noisy measurements. That focus is valuable in its own right.

**Weaknesses**

In its current form, the paper more convincingly supports the claim that this is a challenging interactive causal benchmark than the stronger claim that it already provides a rigorous, stable, and independently attributable measurement of causal thinking.

### **Originality**

**Strengths**

The main originality of the work lies in the benchmark formulation. Combining SCM-driven hidden-mechanism discovery, multi-turn experiment design, selection bias, hidden confounders, and survival outcome together with explanation rubric evaluation is, in my opinion, a genuinely novel direction among existing AI Scientist benchmarks.

**Weaknesses**

I do think the paper introduces an interesting and valuable evaluation framework, but its distinctiveness and persuasiveness as a causal-thinking measurement tool still require a more rigorous validity study.

---

> ### Author Rebuttal · Authors · 2026-03-31
>
> Thank you for your time and constructive comments on our work! Please find our detailed responses to each of your concerns.
>
> > W1.1 & Q1 More details of the LLM-as-Judge.
>
> We have revised our manuscript to include the full details of the LLM-as-Judge setting. Furthermore, we also examined the sensitivity and fine-grained analysis of failure attribution. Please refer to our response to Reviewer ZoF6.
>
> > W1.2 & Q2 Agentic protocol design.
>
> Following your suggestion, we conducted additional experiments with **OpenCode**, a popular autonomous coding-agent framework representative of the latest agentic paradigm. Unlike ReAct's simple think-act-observe loop, OpenCode features persistent memory management, autonomous code generation and execution for data analysis, and structured workspace organization that is increasingly adopted by modern agent systems (e.g., Claude Code, Cursor). We evaluated 5 models × 14 scenarios × 3 trials:
>
> | Mode |Avg Survival Rate |
> |------|:------------------:|
> | Prompting | 61.3% |
> | ReAct (our original agentic) | 60.5% |
> | **OpenCode (coding agent)** | **67.4%** |
>
> OpenCode outperforms ReAct on all 5 models tested (GPT-5.2: +13.9, GPT-5.2 High: +9.3, GPT-5 Mini: +6.3, Grok 4.1: +2.7, Kimi K2.5: +2.2). This confirms that a more capable agentic framework **does** improve performance. Nevertheless, a significant gap to the optimal survival rate (~80%) persists across all models, indicating that **causal thinking capability remains the core bottleneck**. Please refer to [this link](https://anonymous.4open.science/r/CausalGame-Rebuttal-7CB5/opencode_results.md) for the full results.
>
>
> > W1.3 & Q3 Significance analysis.
>
> Following your suggestion, we reran all evaluations with 3 independent trials. The median per-scenario standard deviation is 5.8% (Agentic) and 7.4% (Prompting), indicating reasonable stability across trials. Please refer to our response to Reviewer fihw (W2) and the [anonymous repository](https://anonymous.4open.science/r/CausalGame-Rebuttal-7CB5/).
>
> > W2.1 Presentation of Prompting and Agentic settings.
>
> Following your suggestion, in the revised version, we have moved the introduction of prompting and agentic modes (Appendix A.1) to the experiment section.
>
> > W2.2 & W3.1 & W4.1 Some claims may not be supported.
>
> Following your suggestion, we have revised our wording to be more precise. For example, we changed "provides a rigorous measurement" to "provides a controlled testbed for evaluating causal thinking"; changed "none of the existing benchmarks examine the causal thinking capability" to "existing benchmarks do not explicitly incorporate challenges from hidden confounders, selection bias, and noisy measurements that are central to causal reasoning in scientific discovery".
>
> > W2.3 & Q4 Final rubric scheme.
>
> Thank you for your careful reading. The additional items in the appendix are typos of the previous version of the rubric evaluation. We double-check and provide a clean version of the rubrics in the revision, which can be found in the response to Reviewer ZoF6
>
> > Q5 Clarification on query_environment.
>
> Thanks for your careful observation. query_environment is an auxiliary tool for discovering supplementary variables (e.g., radio_noise, moon_phase) not in the initial data, backed by a Gemini 2.5 Flash interpreter that blocks jailbreak queries.
> In our current setting, all core variables needed to solve each scenario are directly observable in flight history, and the tool aids mechanistic understanding but the optimal design can also be reached through systematic experimental exploration (e.g., testing DEF values).
> The issue in the trajectory is caused by the Gemini interpreter's instability. Hence, we double-check all the trajectories and re-run the evaluation with three trials that are free of the issue.
>
>
>
>
> > Limitation.
>
> We have revised our manuscript to include the discussion. Here is a brief summary: CausalGame's synthetic scenarios, while grounded in historical scientific cases (Table 2), are necessarily simplified compared to real-world discovery involving open-ended hypothesis spaces and domain-specific knowledge; The Prompting and Agentic modes differ in multiple dimensions simultaneously (Table 7), so cross-mode comparisons should be interpreted cautiously; The LLM-based rubric judge may introduce systematic biases, though the objective survival rate metric provides a complementary judge-independent evaluation axis.

---

> > ### Author Rebuttal · Reviewer_R6d8 · 2026-04-03
> >
> > Thank you for the detailed rebuttal. I will raise my score.

---

> > > ### Author Response · Authors · 2026-04-03
> > >
> > > Dear Reviewer R6d8,
> > >
> > > Thank you so much for recognizing our efforts and for raising your score. We sincerely appreciate your constructive comments, which helped strengthen our work!
> > >
> > > Best regards,
> > > Authors

---

### Official Review · Reviewer_ZoF6 · 2026-03-13

**Soundness:** 3
**Presentation:** 3
**Significance:** 4
**Originality:** 3
**Overall Recommendation:** 5
**Confidence:** 4

**Summary:**

This work introduces a novel benchmark, CausalGame, that seeks to evaluate the causal reasoning capabilities of frontier models. The evaluation setting is based around a straightforward multi-turn game in which a drone is deployed with the objective of reaching some target area while avoiding adversarial detection under defined environmental settings. This takes the shape of an interactive environment which the evaluated model is able to test out various configurations of the drone to discover the optimal settings for the given environment. The evaluations include several causal mechanisms: selection bias, measurement error, and latent confounders. Selection bias allows models to observe only successful trials, which allows the authors to induce statistical patterns that obfuscate the optimal settings. For instance, the authors provide an example of a damaged antenna leading to a lower detection probability, which will potentially push the model down the incorrect solution path of increasing the defense of the drone's antenna defense. The other two settings induce additional difficulties via noisy reported data (measurement error) and unobserved variables impacting outcomes (latent confounders). The authors evaluate the results of the models by conducting simulations using the models' final reports as well as evaluation the solution paths via LLM-as-a-judge to distinguish why models failed under certain settings. The authors find that the frontier models fail to both find the optimal settings under all settings and also fail to achieve a 75% survival rate (designated the 'win' threshold by the authors).

**Compliance With Llm Reviewing Policy:**

Affirmed.

**Key Questions For Authors:**

- How do you know the benchmark measures causal reasoning rather than general exploration competence or robustness to misleading feedback?
- How sensitive are the conclusions to the judge model and rubric design?
- What kinds of interventions actually help, and what would count as progress on this benchmark?

**Limitations:**

Yes

**Strengths And Weaknesses:**

Strengths:

The benchmark setting is well-constructed and original (to the best of my knowledge) for evaluating the causal reasoning abilities of LLMs. The construction of a highly controlled framework in which the model can iteratively explore configurations, receive environmental responses, and adjust the configurations provides a valuable setting for evaluating the underlying reasoning patterns of the model.

The authors provide a thorough set of experiments under the different environmental settings that illustrate how specific environmental conditions impact model performance, in addition to the set of traps that construct difficult settings.

Weaknesses:
A key claim of the benchmark is measuring causal reasoning, but the benchmark still mixes together several abilities: experimental search, exploration under budget, robustness to misleading feedback, and strategy revision. The SCM-based design is strong, but it does not by itself establish that poor performance can be cleanly attributed to failures of causal reasoning rather than broader weaknesses in long-horizon agentic decision-making or adaptive experimentation. The authors claim that "state-of-the-art frontier LLMs fail to think with causality", which may be too general of a claim given the scope of the benchmark.

A weakness in the analysis of results is a heavy reliance on LLM-as-a-judge evaluation for analyzing model failure modes. I believe the inclusion of a model's actual response, the Judge prompt, and the Judge's response would provide more qualitative insight into the construction of this evaluation. Further, it would be useful to evaluate the configurations that the model generated to evaluate failure modes rather than relying solely on the responses. While LLM-as-a-judge is the best that can be done in some settings, it seems as though the models produce multiple explorations with different configurations, the path of which could be analyzed to glean insight into how the model solves the optimization problem. The generated configurations contain the model's explorations and would provide useful insight into the alignment between what the model response contains and what solution path it actually takes.

---

> ### Author Rebuttal · Authors · 2026-03-31
>
> Thank you for your time and constructive comments! Our detailed responses are as follows.
>
> > W1 & Q1 What CausalGame measures.
>
> We need to clarify that realistic causal thinking for scientific discovery naturally involves multiple capabilities, and we implement several operations to highlight the challenge of causal thinking:
> - In CausalGame, different settings differ mostly in the causal thinking difficulty, while other settings, such as the budget is identical. Hence, **the performance differences across different settings can be attributed to the capability differences in causal thinking**;
> - Our correlation analysis in Fig.6 also shows that **CausalGame measures a distinct capability**. If poor performance were primarily due to long-horizon decision-making or adaptive experimentation, we would expect strong positive correlations with benchmarks like tau-2 (multi-turn tool use) or Vending-Bench (long-horizon coherence). Instead, we observe only weak correlations, suggesting CausalGame captures an ability not measured by existing agentic benchmarks.
> - Our fine-grained failure mode analysis in Sec. 4.3 also shows that the lack of causal thinking capability is the main reason for the low scores of LLM agents in CausalGame.
>
> > W2.1 & Q2 Rubric judge and sensitivity.
>
> Following your suggestion, we have revised our manuscript to include samples of our rubric evaluation and full details of the rubrics.
> The rubrics we use can be found [here](https://anonymous.4open.science/r/CausalGame-Rebuttal-7CB5/rubric_scoring_criteria.md), from which we define and identify a set of failure modes as shown [here](https://anonymous.4open.science/r/CausalGame-Rebuttal-7CB5/failure_mode_def.md). We reran the fine-grained failure mode analysis and updated the results as in [Failure mode distribution](https://anonymous.4open.science/r/CausalGame-Rebuttal-7CB5/updated_Fig7.pdf) and [Rubric scores](https://anonymous.4open.science/r/CausalGame-Rebuttal-7CB5/updated_Fig8.pdf).
>
> Detailed examples can be found [here](https://anonymous.4open.science/r/CausalGame-Rebuttal-7CB5/rubric_scoring_example.md)
>
> In addition, we also examine the **agreement of different LLM judges**. We use three judge models (gemini-3-flash, grok-4-1-fast-reasoning, and qwen3-next-80b-a3b) to score the agent responses. We calculate ICC(2,3) [1] to assess consistency among these models.
>
> | Rubric | ICC(2,3) | Interpretation | Score Distribution (0.0/0.5/1.0) |
> |:------:|:--------:|:--------------:|:--------------------------------:|
> | ED1 | 0.8905 | Good | 29% / 28% / 43% |
> | RQ1 | 0.8784 | Good | 56% / 21% / 23% |
> | DU1 | 0.8530 | Good | 35% / 22% / 43% |
> | CR2 | 0.6406 | Moderate | 87% / 6% / 7% |
> | CR1 | 0.6297 | Moderate | 87% / 8% / 5% |
> | CR3 | 0.6063 | Moderate | 92% / 8% / 1% |
>
> The results show high inter-rater agreement across all evaluation criteria (Mean ICC = 0.75), with particularly strong consistency for Experimental Design (ED1), Reflection Quality (RQ1), and Data Usage (DU1) rubrics (ICC > 0.85). While the Causal Reasoning rubrics (CR1-3) showed moderate agreement (ICC ~0.61-0.64), this is primarily attributable to the highly skewed score distributions (87-92% zeros) rather than model inconsistency.
>
> [1] Intraclass correlations: uses in assessing rater reliability.
>
> > W2.2 Configuration-based failure modes.
>
> We conducted a judge-independent configuration path analysis on the Antenna Trap, directly examining the sequence of designs agents deployed rather than relying on self-reports. We quantify exploration via Euclidean distance between consecutive configurations and define failure modes through objective thresholds. Key findings across 62 sessions:
> - 54.8% exhibited component lock-in (≤2 unique values for some components), indicating insufficient exploration;
> - 17.7% partially recognized the antenna trend (testing down to antenna_def 6–10) but never reached the optimal range (≤5);
> - 9.7% completely fell into the antenna trap, never testing antenna_def ≤ 10 despite fluctuating survival rates;
> - Even models that discovered the correct direction often suffered from optimization drift, e.g., one GPT-5.2 session found antenna_def=3 with 73.3% survival but drifted back to suboptimal values in later steps
>
> This behavioral analysis confirms the rubric-based findings through an entirely independent lens: models fail not only in explanation but in their actual experimental choices.
>
>
> > Q3 Effective interventions progress on CausalGame.
>
> CausalGame's procedural generation enables constructing diverse environments for agentic RL that explicitly incentivize causal thinking beyond correlational shortcuts. Hence, we foresee interventions that leverage CausalGame to further train the agent as a promising direction.
> We consider meaningful progress along two axes: (1) survival rate improvements in CausalGame; and (2) higher causal reasoning rubric scores, ensuring progress reflects genuine causal understanding rather than better search heuristics.

---

> > ### Author Rebuttal · Reviewer_ZoF6 · 2026-04-04
> >
> > **W1 & Q1 What CausalGame measures.**
> >
> > Would you be able to provide additional details on how figure 6 was generated? Ex. Is it based on some relative ranking of model performances or computed based on raw performances? It would also strengthen the paper to add the answers to these questions in an appendix section, as it is difficult to fully rely on figure 6 without a full understanding of how the results are generated.
> >
> > *In CausalGame, different settings differ mostly in the causal thinking difficulty, while other settings, such as the budget is identical. Hence, the performance differences across different settings can be attributed to the capability differences in causal thinking*
> >
> > I believe that it's reasonable to claim that the difficulty of the causal problem is varied across the settings. However, while the sources of difficulty (selection bias, measurement error, and latent confounding) cover or partially cover many evaluations of causal reasoning, they are not exhaustive (e.g., counterfactual reasoning and instrumental variable reasoning). In light of this, the claim of Observation 1 seems to be too general and should be attenuated to the specific challenges and assumptions evaluated in this work.
> >
> > **W2.2 Configuration-based failure modes.**
> > Thank you for including this.  It adds context to the results and would be helpful to see in the camera-ready version. I understand that implementation details likely could not be provided in the intial rebuttal due to character limits, so would it be possible to elaborate on how the evaluation was conducted?

---

> > > ### Author Response · Authors · 2026-04-04
> > >
> > > Thank you for engaging in the discussion and for your further feedback. Please feel assured that we will incorporate all the discussion and implementation details in the final version. Below, we provide more clarification regarding your remaining concerns.
> > >
> > > > How Figure 6 was generated
> > >
> > > For each model, we collect its performance score on CausalGame and on each external benchmark (HLE, SWE-Coding, tau-2, Vending-Bench, AA-LCR, Non-Hallucination Rate). We then compute pairwise Spearman rank correlations across models between CausalGame scores and each benchmark score.
> > >
> > > > The claim of observation 1
> > >
> > > We agree, and we will revise the claim to be more specific: State-of-the-art frontier LLMs fail to identify and reason about hidden causal mechanisms in interactive settings involving challenges like hidden confounders, selection bias, and noisy measurements.
> > >
> > > > Implementation details for configuration-based failure modes.
> > >
> > > Specifically, based on all agentic sessions from the Antenna Trap, we extracted the full sequence of 7-dimensional design vectors (engine_def, wing_def, body_def, cockpit_def, antenna_def, camera_def, gun_def) across all deployment rounds. We quantified exploration via Euclidean distance between consecutive configuration changes. We then define six failure modes by objective thresholds applied directly to these configuration sequences:
> > >
> > > | Failure Mode | Criteria |
> > > |--------------|-------------------|
> > > | Antenna Trap | All `antenna_def` > 10 (never tested ≤10) |
> > > | High Antenna Bias | Min `antenna_def` ∈ [6, 10] (tested low but not optimal ≤5) |
> > > | Premature Convergence | Avg `change` < 5 after step 5 (only minor adjustments) |
> > > | Suboptimal Lock-in | >60% deployments with survival < 50% (stuck in poor configs) |
> > > | Ignoring Evidence| Best config not used in final 2 deployments (abandoned success) |
> > > | Component Lock-in | Any component has ≤2 unique values (insufficient exploration) |
> > >
> > > Based on the failure modes, we can then count and analyze the frequencies and derive the conclusions as well as case studies, as illustrated in our rebuttal response.
> > > Please feel free to let us know if you still have any other questions. We are happy to provide more clarification if needed. Thank you again for your time and effort in reviewing our work!

---

### Official Review · Reviewer_LrZZ · 2026-03-13

**Soundness:** 3
**Presentation:** 3
**Significance:** 4
**Originality:** 3
**Overall Recommendation:** 4
**Confidence:** 4

**Summary:**

In this paper, the authors introduce CausalGame, an interactive benchmark designed to evaluate how well LLM-based agents can actually handle causal reasoning and discovery. Instead of standard static datasets, they use Structural Causal Models (SCMs) as the underlying engine to generate 14 different game scenarios. What makes these scenarios interesting is that they intentionally bake in realistic observational traps, like selection bias, hidden confounding, and measurement error. The authors tested 16 recent LLMs using both single-turn prompting and agentic (ReAct) setups, and found that the models consistently struggle—they fail to reach optimal survival rates and generally can't articulate the correct causal mechanisms. Overall, the benchmark pairs outcome-based success metrics with an LLM-judged rubric to diagnose these failures, offering a much-needed testing ground for future AI scientist agents.

**Compliance With Llm Reviewing Policy:**

Affirmed.

**Key Questions For Authors:**

1. Could you include some non-LLM or hybrid baselines (e.g., standard causal active-learning heuristics or randomized control policies)? This would really help demonstrate that the tasks are solvable and give us a non-LLM ceiling to compare against.
2. Will you be providing confidence intervals and variance across seeds/trials in the final version? Also, to ensure the theoretical optimal win thresholds are reproducible, could you provide at least one fully specified SCM (equations, noise distributions) per family?
3. How exactly is the LLM-judge rubric validated? Do you have any human annotations or inter-rater reliability metrics to ensure the judge isn't drifting or just favoring certain LLM explanation styles?

**Limitations:**

The authors do a thorough job discussing the failure modes of the LLMs they evaluated. However, they don't really address the limitations of their own evaluation setup specifically the potential circularity of using an uncalibrated LLM to score the explanatory rubric.

**Strengths And Weaknesses:**

Soundness:
I really appreciate the "SCM-as-engine" approach, it’s a principled way to ensure that agents only succeed if they actually recover causal mechanisms, rather than just exploiting spurious correlations. The experimental setup is also quite broad, testing a wide range of models. However, my main concern is the lack of non-LLM or hybrid baselines. Without seeing how a standard causal discovery heuristic or a simple randomized ablation policy performs, it’s hard to calibrate the actual difficulty of the benchmark or confirm task solvability. Furthermore, the SCMs (functional forms, noise distributions) aren't fully specified, and relying so heavily on an uncalibrated LLM judge for the rubric introduces a risk of circularity.

Presentation:
The paper is generally very well-written and easy to follow. The motivation is clear, and the qualitative examples (like the Antenna Trap) make a lot of intuitive sense. The failure-mode taxonomy is also a nice touch. That said, I noticed a few minor data hygiene issues in the tables and figures (like duplicated model names and some odd correlation values). More importantly, the paper is missing a dedicated reproducibility section detailing the experimental knobs, like temperature, exploration budgets, and tool limits.

Significance:
The authors are tackling a highly relevant and under-measured problem. If we want to build reliable AI scientist agents, evaluating their causal competence under realistic pitfalls is critical. The negative results are a sobering and valuable reality check for the community. To maximize its impact, though, I think the authors should release a simple rule-based agent that actually succeeds on the benchmark, so it can be used constructively for progress rather than just as a demonstration of LLM failures.

Originality:
While there is certainly no shortage of LLM benchmarks these days, explicitly grounding text environments in complex SCMs to test for latent confounders and measurement error is a very creative and novel synthesis. I do think the framing could be improved by drawing clearer comparisons to recent interactive scientific-discovery benchmarks (like BoxingGym) and connecting the experimental design aspects more strongly to the classical causal active learning literature.

---

> ### Author Rebuttal · Authors · 2026-03-31
>
> Thank you for your time and constructive comments! Below we provide detailed responses to your questions.
>
> > W1 & Q1 Non-LLM or hybrid baselines.
>
> In CausalGame, we aim to provide a more realistic interactive environment using natural language and JSON data. In contrast, standard causal discovery or causal active-learning methods (even when combined with LLMs [1]) rely on structured tabular data. Therefore, we further consider exploring 4 randomized ablation policies:
>
> `Default` (submit the initial design unchanged), `Random` (uniformly sample each DEF value from [0, 50]),  `Uniform High` (set all components to DEF=50), and  `No-Explore LLM` (randomly perform 10 deploys and use the LLM to analyze observations and submit design):
>
> | Baseline | Type | Avg Survival Rate |
> |----------|------|:-------:|
> | Default | Rule-based | 49.0% |
> | Random | Rule-based | 52.0% |
> | Uniform High | Rule-based | 52.7% |
> | No-Explore LLM | Hybrid | 52–63% |
>
> Details can be found in the [anonymous repo](https://anonymous.4open.science/r/CausalGame-Rebuttal-7CB5/). Interestingly, those baselines can outperform several full-agent models on bias-heavy scenarios, suggesting the necessity of causal thinking.
>
> > W1.2&Q2.1 Confidence intervals and variance.
>
> Please refer to our response to Reviewer fihw (W2) and the [anonymous repo](https://anonymous.4open.science/r/CausalGame-Rebuttal-7CB5/) for the full significance analysis.
>
> > Q2.2 Fully Specified SCMs
>
> We have revised our manuscript to include fully specified SCMs with one representative scenario per category. The complete structural equations, noise distributions, and parameter specifications for all 14 scenarios are available in the anonymous repository [link] (https://anonymous.4open.science/r/CausalGame-Rebuttal-7CB5/scm_specification.md).
>
> > W1.3 & Q3 Reliability of LLM-as-judge
>
> Please refer to our responses to Reviewer ZoF6.
>
> > W2 Duplicated model names, odd correlation and reproducibility section.
>
> Thank you for careful reading. We have checked and removed duplicated model name typos.
> We also included the following reproducibility details in the revised manuscript:
> | Parameter | Value | Notes |
> |-----------|-------|-------|
> | **Trials per combination** | 3 | Independent sessions |
> | **Exploration budget** | 200 drones total | Agent decides per-call allocation |
> | **Evaluation fleet** | 1000 drones | Stage 2 final evaluation |
> | **Deploy calls limit** | 10 | Maximum interventions in Stage 1 |
> | **Temperature / Max tokens** | Default API values for all models | No custom tuning, ensuring fair comparison |
> Furthermore, we provide the API provider info in [this link] (https://anonymous.4open.science/r/CausalGame-Rebuttal-7CB5/model_access.md).
>
> > W3 Simple, rule-based agent that actually succeeds on the benchmark
>
> Since optimal experiment design involves non-trivial efforts[2], it is relatively challenging to find a single rule-based agent that succeeds across all scenarios. Nevertheless, we can construct simple baselines (such as the previous randomized policy) to obtain non-trivial win rate. For example, `uniform_high` baseline (all DEF=50) achieves 100% win rate on 4 of 6 DZ-categorical scenarios (~78% survival). Similarly, `default` achieves 100% on AT-local_optima by copying the near-optimal history design.
>
>
> > W4 Drawing clearer comparisons to the related works.
>
> Our benchmark occupies a distinctive niche at the intersection of interactive scientific-discovery benchmarks and causal active learning:
> - Recent interactive discovery benchmarks such as BoxingGym, DiscoveryWorld, NewtonBench, primarily evaluate whether agents can identify functional relationships from observed data, but none explicitly construct adversarial causal challenges where **naive statistical analysis yields systematically misleading conclusions**. CausalGame incorporates realistic challenges such as selection biases and hidden confounders that can not be solved by statistical analysis alone, mirroring well-documented obstacles in real scientific history.
> - Active causal discovery considers a simplified setting where all the causal variables are present and a perfect intervention outcome is accessible[1,2], which is usually not the case in realistic causal discovery[3]. CausalGame requires agents to cope with both hidden variables and imperfect observations affected by selection bias, **bridging the gap between idealized active learning and realistic discovery**.
>
> **References**
>
> [1] Can Large Language Models Help Experimental Design for Causal Discovery?
>
> [2] Experiment Selection for Causal Discovery
>
> [3] Discovery of the Hidden World with Large Language Models

---

> > ### Author Rebuttal · Reviewer_LrZZ · 2026-04-03
> >
> > i will keep my positive score

---

> > > ### Author Response · Authors · 2026-04-03
> > >
> > > Dear Reviewer LrZZ,
> > >
> > > Thank you for your engagement in the discussion and for maintaining your positive assessment of our work. We sincerely appreciate your constructive feedback.
> > >
> > > We noticed you selected option (c). Would you mind confirming whether this was intended? We are happy to provide further clarification if needed.
> > >
> > > Thank you again for your time and thoughtful comments on our work!
> > >
> > > Best regards,
> > > Authors

---

### Decision · Program_Chairs · 2026-04-30

**Decision:**

Accept (spotlight)

**Comment:**

In this paper, the authors introduce CausalGame, a new benchmark for evaluating the casual thinking capabilities of AI agents using interactive games. Designed for AI Scientist agents, the benchmark asks agents to design experimental protocol, gather observation data, and derive a solution with a report containing an explanation. CausalGame includes 14 game scenarios that incorporate a selection of bias, noisy measurements, and hidden confounders. The authors evaluate 16 AI agents using CausalGame and illustrate that agents AI agents consistently fail to reason about and recover the underlying causal relationships required for solving the tasks.

The reviewers have noted that the paper is well-written and easy to follow. They highlighted that the benchmark is well-constructed and original, and that the manuscript provides thorough experimental results to support the main claims in the paper. The reviewers have raised some concerns which were addressed by the authors during the rebuttal phase, leading to increases in score and confidence. I similarly believe that this is an important research contribution with compelling results, and therefore I’m pleased to recommend it for acceptance.